# A representation-learning game for classes of prediction tasks

## Abstract

We introduce a formulation for learning dimensionality-reducing representations of unlabeled feature vectors, when a prior knowledge on future prediction tasks is available. The formulation is based on a three-player game, in which the first player chooses a representation, the second player then adversarially chooses a prediction task, and the third player predicts the response based on the represented features. The first and third player aim is to minimize, and the second player to maximize, the *regret*: The minimal prediction loss using the representation compared to the same loss using the original features. Our first contribution is theoretical and addresses the mean squared error loss function, and the case in which the representation, the response to predict and the predictors are all linear functions. We establish the optimal representation in pure strategies, which shows the effectiveness of the prior knowledge, and the optimal regret in mixed strategies, which shows the usefulness of randomizing the representation. We prove that optimal randomization requires a precisely characterized finite number of representations, which is smaller than the dimension of the feature vector, and potentially much smaller. Our second contribution is an efficient gradient-based iterative algorithm that approximates the optimal mixed representation for a general loss function, and general classes of representations, response functions and predictors.

## 1 Introduction

A common practice in modern data-science is to collect as much data as possible, even without an exact knowledge of a subsequent prediction task it will be used for. The data collected is an unlabeled set of feature vectors $\{\boldsymbol{x}_i\} \subset \mathbb{R}^d$ . Then, when a specific prediction task becomes of interest, responses $\boldsymbol{y}_i \in \mathcal{Y}$ are collected, and a learning algorithm is trained on the pairs $\{(\boldsymbol{x}_i, \boldsymbol{y}_i)\}$. Modern sources, such as high-definition images or genomic sequences, have high dimension $d$, and this raises the question of *dimensionality-reduction*, either for a better generalization [1], for storage/communication savings [2–4], or for interpretability [5]. The goal is thus to find a *representation* $\boldsymbol{z} = R(\boldsymbol{x}) \in \mathbb{R}^r$, where $d \gg r$, that preserves the relevant part of the features, without a full knowledge of their utility for future prediction tasks. In this paper, we propose an unsupervised-learning game-theoretic framework for this goal, whose central aspect is an assumption of prior knowledge on the *class* of future prediction tasks. Our contributions are a theoretical solution in a fully linear setting, under the mean squared error (MSE) loss, and an algorithm for the general setting.

Popular approaches to dimensionality reduction are oblivious to prior knowledge on the prediction task. Most prominently, *principal component analysis* (PCA) [6–9], and non-linear extensions such as kernel PCA [10] and *auto-encoders* (AE) [11–13, 1], aim that the representation $\boldsymbol{z}$ will maximally preserve the *variation* in $\boldsymbol{x}$. Nonetheless, prior knowledge may indicate that the highly varying directions in the feature space are irrelevant for future prediction tasks. From the supervised learn-

ing perspective, it is well established that efficient representations are inherent to efficient learning [14, 15]. In this respect, the *information bottleneck* (IB) principle [16–19] was used to postulate that efficient supervised learning learns representations which are both low-complexity and relevant [20–24] (this spurred a debate, e.g., [25, 26]). The original IB formulation is based on the mutual information functional [27], which is difficult to estimate (especially in high dimensions), and ignores complexity constraints on the representation or prediction [28, 29]; see a review in Appendix B. Using the notion of *usable information*, introduced in [28], optimal representations for the *supervised* learning setting were explored in [29] via a *two-player game* between Alice, which selects a prediction problem of $y$ given $x$, and Bob, which then selects the representation $z$. Alice then uses an empirical risk minimizer with the standard goal of minimizing the expected risk. It was established in [29] that ideal generalization is obtained for representations that optimize the *decodable IB*.

In this paper, we build upon [29], and propose a *three-player game* for unsupervised representation-learning, chosen without a specific prediction problem (Section 2). First, the *representation player* reduce $x \in \mathbb{R}^d$ to a representation $z \in \mathbb{R}^r$, where $r < d$. Second, the *response function player* chooses a response (label) $y$ rule $f$ for $x$, from a given known class of (random) response functions $\mathcal{F}$. The choice of this class manifests the prior knowledge available on the type of prediction problems that the representations will be used for. Third, the *predictor player* optimally predicts $y$ from $z$. The value of the game is determined by the *regret*: The prediction loss based on the representation $z$ compared to prediction loss based on $x$. The first and last player cooperate in order to minimize the regret, whereas the response function player aims to maximize it. In other words, the representation is chosen to minimize the worst-case prediction loss for any response in $\mathcal{F}$. The output of this game is the representation chosen by the first player. In order to focus on the representation aspect we side-step the generalization problem, and assume that sufficient labeled data will be provided to the predictor later on in order to accurately estimate the prediction rule.

This formulation directly addresses the relevance of a "direction" in the feature space to the prediction tasks in $\mathcal{F}$, rather than its variability, as in standard unsupervised learning (e.g., PCA and AE). Compared to [29], the representation is chosen based only on the *class* of possible response functions, rather than a specific one. Such knowledge on $\mathcal{F}$ may stem from various considerations: Domain specific, imposed by privacy or fairness constraints, or stem from transfer or continual learning setting; see Appendix A for an extended discussion. Technically, the game in [29] replaces the order of the first (representation) and second (response) players. From a different perspective, our method is a *self-supervised learning* method, for which the prior knowledge on $\mathcal{F}$ serves as a "self-defined signal" for choosing an optimal representation, without any labeled data (see, e.g., [30] and [31] for recent surveys). In addition, our game formulation naturally leads to a *mixed strategies* solution [32], that is, allowing the representation player to randomized the representation rule, in order to mix up the adversarial response player. This randomization is an inherent aspect of the IB formulation, but its usage there is not rigorously justified. By contrast, for standard unsupervised learning, mixed representation does not improve the regret (see Proposition 14 in Appendix E.1 for the PCA setting). In Appendix B we provide a thorough discussion of related work.

## Contributions

• Theoretical: We address the fundamental setting in which the representation, the response, and the prediction are linear functions, under the MSE loss function (Section 3). The prior knowledge on $\mathcal{F}$ is represented by a symmetric matrix $S$ that determines the principal directions of the function in the feature space. We establish the optimal representation and regret in pure strategies, which shows the utility of the prior information, and in mixed strategies, which shows that randomizing the representation yields *strictly lower* regret. We prove that randomizing between merely $\ell^*$ different representation rules suffices, where $r + 1 \leq \ell^* \leq d$ is a precisely characterized *effective dimension*.

• Algorithmic: We develop an iterative gradients-based algorithm that approximates the optimal mixed representation (Section 4) for general representations/response/predictors and loss functions. The algorithm is greedy, and alternates between finding a new representation rule and an adversarial function. We empirically verify that the output mixed representation has close-to-optimal regret in the linear MSE setting. To optimize the weights of the representation, we essentially solve a minimax two-player games, and to this end, we utilize the classic multiplicative weights update (MWU) algorithm [33] (which is essentially a follow-the-regularized-leader [34, 35]).

## 2 Problem formulation

We use mostly conventional notation that is detailed in Appendix C. Specifically, the eigenvalues of a positive-semidefinite matrix $S$ are denoted as $\lambda_{\max}(S) \equiv \lambda_1(S) \geq \cdots \geq \lambda_d(S) = \lambda_{\min}(S)$ and $v_i(S)$ denotes an eigenvector corresponding to $\lambda_i(S)$ such that $V = V(S) := [v_1(S), v_2(S), \cdots, v_d(S)] \in \mathbb{R}^{d \times d}$ and $S = V(S)\Lambda(S)V^\top(S)$ is an eigenvalue decomposition. For a matrix $W \in \mathbb{R}^{d \times d}$ we let $W_{i:j} := [w_i, \ldots, w_j] \in \mathbb{R}^{(j-i+1) \times d}$ denote the matrix comprised of the columns indexed by $\{i, \ldots, j\}$. We denote the probability law of a random variable $\boldsymbol{x}$ as $\mathsf{L}(\boldsymbol{x})$.

Let $\boldsymbol{x} \in \mathcal{X}$ be a random feature vector, where $P_{\boldsymbol{x}} := \mathsf{L}(\boldsymbol{x})$ is known. Let $\boldsymbol{y} \in \mathcal{Y}$ be a corresponding response drawn according to a probability kernel $\boldsymbol{y} \sim f(\cdot \mid \boldsymbol{x} = x)$, where for brevity, we will refer to $f$ as the *response function*. We assume $f \in \mathcal{F}$ for some known class $\mathcal{F}$. Let $\boldsymbol{z} := R(\boldsymbol{x}) \in \mathbb{R}^r$ be an $r$-dimensional representation of $\boldsymbol{x}$ where $R: \mathcal{X} \to \mathbb{R}^r$ is chosen from a class $\mathcal{R}$ of representation functions, and let $Q: \mathbb{R}^r \to \mathcal{Y}$ be a prediction rule from a class $\mathcal{Q}$, with the loss function $\mathsf{loss}: \mathcal{Y} \times \mathcal{Y} \to \mathbb{R}_+$. The *regret* of the representation $R$ for the response function $f$ is

$$\mathsf{regret}(R, f \mid P_{\boldsymbol{x}}) := \min_{Q \in \mathcal{Q}} \mathbb{E}\left[\mathsf{loss}(\boldsymbol{y}, Q(R(\boldsymbol{x})))\right] - \min_{Q: \mathbb{R}^d \to \mathcal{Y}} \mathbb{E}\left[\mathsf{loss}(\boldsymbol{y}, Q(\boldsymbol{x}))\right]. \tag{1}$$

The *minimax regret in mixed strategies* is defined via the worst case response function in $\mathcal{F}$ as

$$\mathsf{regret}_{\mathsf{mix}}(\mathcal{R}, \mathcal{F} \mid P_{\boldsymbol{x}}) := \min_{\mathsf{L}(\boldsymbol{R}) \in \mathcal{P}(\mathcal{R})} \max_{f \in \mathcal{F}} \mathbb{E}\left[\mathsf{regret}(\boldsymbol{R}, f \mid P_{\boldsymbol{x}})\right], \tag{2}$$

where $\mathcal{P}(\mathcal{R})$ is a set of probability measures on the possible set of representations $\mathcal{R}$. The *minimax regret in pure strategies* restricts $\mathcal{P}(\mathcal{R})$ to degenerated measures (deterministic), and so the expectation in (2) is removed. Our main goal is to determine the optimal representation strategy, either in pure $R^* \in \mathcal{R}$ or mixed strategies $\mathsf{L}(\boldsymbol{R}^*) \in \mathcal{P}(\mathcal{R})$. To this end, we will also utilize the *maximin* version of (2). Specifically, let $\mathcal{P}(\mathcal{F})$ denote a set of probability measures supported on $\mathcal{F}$, and assume that for any $R \in \mathcal{R}$, there exists a measure in $\mathcal{P}(\mathcal{R})$ that puts all its mass on $R$. Then, the *minimax theorem* [32, Chapter 2.4] [36] implies that

$$\mathsf{regret}_{\mathsf{mix}}(\mathcal{R}, \mathcal{F} \mid P_{\boldsymbol{x}}) = \max_{\mathsf{L}(\boldsymbol{f}) \in \mathcal{P}(\mathcal{F})} \min_{R \in \mathcal{R}} \mathbb{E}\left[\mathsf{regret}(R, \boldsymbol{f} \mid P_{\boldsymbol{x}})\right]. \tag{3}$$

The right-hand side of (3) is the *maximin regret in mixed strategies,* and the maximizing probability law $\mathsf{L}(\boldsymbol{f}^*)$ is known as the *least favorable prior*. In general, $\mathsf{regret}_{\mathsf{mix}}(\mathcal{R}, \mathcal{F} \mid P_{\boldsymbol{x}}) \leq \mathsf{regret}_{\mathsf{pure}}(\mathcal{R}, \mathcal{F} \mid P_{\boldsymbol{x}})$, and the inequality can be strict. We mention that the use of expectation in the definition of the mixed regret over the randomized representation, implies that the empirical performance of a system based on this randomized representation achieves the mixed minimax regret value in the limit of large number of repeating representation games. The size of the dataset for each of these games should be large enough to allow for accurate learning of $\boldsymbol{f}$ to be used by the predictor. By contrast, the pure minimax regret guarantee is valid for a single representation, and thus more conservative from this aspect.

## 3 The linear setting under MSE loss

In this section, we focus on linear classes and the MSE loss function. The response function class is characterized by a quadratic constraint, to wit, the class $\mathcal{F}$ is specified by a matrix $S \in \mathbb{S}_{++}^d$ that represents the relative importance of each direction in the feature space in determining $\boldsymbol{y}$.

**Definition 1** (The linear MSE setting). Assume that $\mathcal{X} = \mathbb{R}^d$, that $\mathcal{Y} = \mathbb{R}$ and the loss function is the MSE, $\mathsf{loss}(y_1, y_2) = |y_1 - y_2|^2$. Assume that $\mathbb{E}[\boldsymbol{x}] = 0$ and let $\Sigma_{\boldsymbol{x}} := \mathbb{E}[\boldsymbol{x}\boldsymbol{x}^T] \in \mathbb{S}_{++}^d$ be its invertible covariance matrix. The classes of representations, response functions, and predictors are all linear, that is: (1) The representation is $z = R(x) = R^\top x$ for $R \in \mathcal{R} := \mathbb{R}^{d \times r}$ where $d > r$; (2) The response function is $F \in \mathcal{F} \subset \mathbb{R}^d$, and $\boldsymbol{y} = f^\top \boldsymbol{x} + \boldsymbol{n} \in \mathbb{R}$, where $\boldsymbol{n} \in \mathbb{R}$ is a heteroscedastic noise that satisfies $\mathbb{E}[\boldsymbol{n} \mid \boldsymbol{x}] = 0$, and given some specified $S \in \mathbb{S}_{++}^d$

$$f \in \mathcal{F}_S := \left\{ f \in \mathbb{R}^d : \|f\|_S^2 \leq 1 \right\}, \tag{4}$$

where $\|f\|_S := \|S^{-1/2}f\|_2 = (f^\top S^{-1} f)^{1/2}$ is the Mahalanobis norm; (3) The predictor is $Q(z) = q^\top z \in \mathbb{R}$ for $q \in \mathbb{R}^r$. Since the regret will depend on $P_{\boldsymbol{x}}$ only via $\Sigma_{\boldsymbol{x}}$, we will abbreviate the notation of the pure (resp. mixed) minimax regret to $\mathsf{regret}_{\mathsf{pure}}(\mathcal{F} \mid \Sigma_{\boldsymbol{x}})$ (resp. $\mathsf{regret}_{\mathsf{mix}}(\mathcal{F} \mid \Sigma_{\boldsymbol{x}})$).

In Appendix E.1 we show that standard PCA can be similarly formulated, by assuming that $\mathcal{F}$ is a singleton containing the noiseless identity function, so that $\boldsymbol{y} = \boldsymbol{x}$ surely holds, and $\hat{x} = Q(z) \in \mathbb{R}^d$. Proposition 14 therein shows that the pure and mixed minimax representations are both $R = V_{1:r}(\Sigma_{\boldsymbol{x}})$, and so randomization is not unnecessary. We begin with the pure minimax regret.

**Theorem 2.** *For the linear MSE setting (Definition 1)*

$$\mathsf{regret}_{\mathsf{pure}}(\mathcal{F}_S \mid \Sigma_{\boldsymbol{x}}) = \lambda_{r+1}\left(\Sigma_{\boldsymbol{x}}^{1/2} S \Sigma_{\boldsymbol{x}}^{1/2}\right). \tag{5}$$

*A minimax representation matrix is*

$$R^* := \Sigma_{\boldsymbol{x}}^{-1/2} \cdot V_{1:r}\left(\Sigma_{\boldsymbol{x}}^{1/2} S \Sigma_{\boldsymbol{x}}^{1/2}\right), \tag{6}$$

*and the worst case response function is*

$$f^* := S^{1/2} \cdot v_{r+1}\left(\Sigma_{\boldsymbol{x}}^{1/2} S \Sigma_{\boldsymbol{x}}^{1/2}\right). \tag{7}$$

The optimal representation thus whitens the feature vector $\boldsymbol{x}$, and then projects it on the top $r$ eigenvectors of the adjusted covariance matrix $\Sigma_{\boldsymbol{x}}^{1/2} S \Sigma_{\boldsymbol{x}}^{1/2}$, which reflects the prior knowledge that $f \in \mathcal{F}_S$. The proof is deferred to Appendix E.2, and its outline is as follows: Plugging the optimal predictor into the regret results a quadratic form in $f \in \mathbb{R}^d$, determined by a matrix which depends on the subspace spanned by the representation $R$. The worst-case $f$ is the determined via the *Rayleigh quotient theorem* [37, Theorem 4.2.2], and the optimal $R$ is found via the *Courant–Fischer variational characterization* [37, Theorem 4.2.6] (see Appendix D for a summary of useful mathematical results). We next consider the mixed minimax regret.

**Theorem 3.** *For the linear MSE setting (Definition 1)*

$$\mathsf{regret}_{\mathsf{mix}}(\mathcal{F}_S \mid \Sigma_{\boldsymbol{x}}) = \frac{\ell^* - r}{\sum_{i=1}^{\ell^*} \lambda_i^{-1}}, \tag{8}$$

*where $\lambda_i \equiv \lambda_i(S^{1/2}\Sigma_{\boldsymbol{x}}S^{1/2})$ and $\ell^*$ is any member of*

$$\left\{\ell \in [d]\setminus[r] : (\ell - r) \cdot \lambda_\ell^{-1} \leq \sum_{i=1}^{\ell} \lambda_i^{-1} \leq (\ell - r) \cdot \lambda_{\ell+1}^{-1}\right\} \tag{9}$$

*(with $\lambda_{d+1} \equiv 0$). Furthermore:*

- *The covariance matrix of the least favorable prior of $\boldsymbol{f}$: Let $\Lambda_\ell := \mathrm{diag}(\lambda_1, \ldots, \lambda_{\ell^*}, 0, \cdots, 0)$, and let $V \equiv V(S^{1/2}\Sigma_{\boldsymbol{x}}S^{1/2})$. Then, the covariance matrix of the least favorable prior of $\boldsymbol{f}$ is*

$$\Sigma_{\boldsymbol{f}}^* := \frac{V^\top \Lambda_{\ell^*}^{-1} V}{\sum_{i=1}^{\ell^*} \lambda_i^{-1}}. \tag{10}$$

- *The probability law of the minimax representation: Let $\overline{A} \in \{0,1\}^{\ell^* \times \binom{\ell^*}{r}}$ be a matrix whose columns are the members of the set*

$$\overline{\mathcal{A}} := \{\overline{a} \in \{0,1\}^{\ell^*} : \|\overline{a}\|_1 = \ell^* - r\} \tag{11}$$

*(in an arbitrary order). Let $\overline{b} = (b_1, \ldots, b_{\ell^*})^\top$ be such that*

$$b_i = (\ell^* - r) \cdot \frac{\lambda_i^{-1}}{\sum_{j=1}^{\ell^*} \lambda_j^{-1}}. \tag{12}$$

*Then, there exists a solution $p \in [0,1]^{\binom{\ell^*}{r}}$ with support size at most $\ell^* + 1$ to $\overline{A}p = \overline{b}$. For $j \in [\binom{\ell^*}{r}]$, let $\mathcal{I}_j := \{i \in [\ell^*] : \overline{A}_{ij} = 0\}$ be the zero indices on the $j$th column of $\overline{A}$, and let $V_{\mathcal{I}_j}$ denote the $r$ columns of $V$ whose index is in $\mathcal{I}_j$. A minimax representation is*

$$\boldsymbol{R}^* = \Sigma_{\boldsymbol{x}}^{-1/2} V_{\mathcal{I}_j} \tag{13}$$

*with probability $p_j$, for $j \in [\binom{\ell^*}{r}]$.*

Interestingly, while the eigenvalues $\lambda_i(\Sigma_{\boldsymbol{x}}^{1/2} S \Sigma_{\boldsymbol{x}}^{1/2}) = \lambda_i(S^{1/2} \Sigma_{\boldsymbol{x}} S^{1/2})$ are equal, the pure mini-max regret utilizes the eigenvectors of $\Sigma_{\boldsymbol{x}}^{1/2} S \Sigma_{\boldsymbol{x}}^{1/2}$ whereas the mixed minimax regret utilizes those of $S^{1/2} \Sigma_{\boldsymbol{x}} S^{1/2}$, which are possibly different. The proof of Theorem 3 is also in Appendix E.2, and is substantially more complicated and longer than for the pure regret. We use a two-step indirect approach, since it seems challenging to directly maximize over $\mathsf{L}(\boldsymbol{R})$. First, we solve the *maximin problem* (3), and find the least favorable prior $\mathsf{L}(\boldsymbol{f}^*)$. Second, we propose a probability law for the representation $\mathsf{L}(\boldsymbol{R})$, and show that its regret equals the maximin value, and thus also the minimax. With more detail, in the first step, we show that the regret only depends on $\mathsf{L}(\boldsymbol{f})$ via $\Sigma_{\boldsymbol{f}} = \mathbb{E}[\boldsymbol{f}\boldsymbol{f}^\top]$, and we explicitly construct a probability law that is both fully supported on $\mathcal{F}_S$ and has this co-variance matrix. This reduces the problem from optimizing $\mathsf{L}(\boldsymbol{f})$ to optimizing $\Sigma_{\boldsymbol{f}}$, whose solution (Lemma 16) leads to the least favorable $\Sigma_{\boldsymbol{f}}^*$, and then to the maximin value. In the second step, we explicitly construct a representation that achieves the maximin regret. Concretely, we construct representation matrices that use $r$ of the $\ell^*$ principal components of $\Sigma_{\boldsymbol{x}}^{1/2} S \Sigma_{\boldsymbol{x}}^{1/2}$, where $\ell^* > r$. The defining property of $\ell^*$ (9) established in the maximin solution is utilized to find weights on the $\binom{\ell^*}{r}$ possible representations, that achieves the maximin solution, and thus also the minimax. The proof uses Carathéodory's theorem (see Appendix D) which also establishes that the optimal $\{p_j\}$ is supported on at most $\ell^* + 1$ matrices, much less than $\binom{\ell^*}{r}$. We next make a few comments:

1. Computing the mixed minimax probability: This requires solving $\overline{A}^\top p = \overline{b}$ for a probability vector $p$, which is a linear-program feasibility problem that is routinely solved [38]. For illustration, if $r = 1$ then $\overline{A} \in \{0,1\}^{\ell^* \times \ell^*}$ is a square all ones matrix, except for a zero diagonal, and $p_j = 1 - (\ell^* - 1)\lambda_j^{-1}/(\sum_{i=1}^{\ell^*} \lambda_i^{-1})$ for $j \in [\ell^*]$. Similarly, the case $\ell^* = r + 1$ is solved by setting $p_j = (\lambda_j^{-1})/(\sum_{j'=1}^{\ell^*} \lambda_{j'}^{-1})$ on the $\ell^*$ standard basis vectors. Nonetheless, the dimension of $p$ is $\binom{\ell^*}{r}$ and thus increases fast as $\Theta((\ell^*)^r)$, and this approach may be intractable. However, in this case the algorithm we present in Section 4 can be used. As we empirically show, it approximately achieves the optimal regret, and the number of atoms is not much larger than $\ell^* + 1$.

2. Required randomness: The regret formulation (2) assumes that the actual realization of the rep-resentation rule is known to the predictor. Formally, this can be conveyed to the predictor using an small header of less than $\log_2(\ell^* + 1) \le \log(d + 1)$ bits. Practically, this is unnecessary and an efficient predictor can be learned from a labeled data set $(\boldsymbol{z}, \boldsymbol{y})$.

3. The rank of $\Sigma_{\boldsymbol{f}}^*$: The rank of the covariance matrix of the least favorable prior is an *effective dimension*, satisfying (see (8))

$$\ell^* = \arg\max_{\ell \in [d] \setminus [r]} \frac{1 - (r/\ell)}{\frac{1}{\ell} \sum_{i=1}^{\ell} \lambda_i^{-1}}. \tag{14}$$

By convention, $\{\lambda_i^{-1}\}_{i \in [d]}$ is a monotonic non-decreasing sequence, and so is the partial Cesàro mean $\psi(\ell) := \frac{1}{\ell} \sum_{i=1}^{\ell} \lambda_i^{-1}$. For example, if $\lambda_i = i^{-\alpha}$ with $\alpha > 0$ then $\psi(\ell) = \Theta(\ell^\alpha)$. If, e.g., $\psi(\ell) = \ell^\alpha$, then it is easily derived that $\ell^* \approx \min\{\frac{\alpha+1}{\alpha} r, d\}$. So, if $\alpha \ge \frac{r}{d-r}$ is large enough and the decay rate of $\{\lambda_i\}$ is fast enough then $\ell^* < d$, and otherwise $\ell^* = d$. As the decay rate of $\{\lambda_i\}$ becomes faster, the rank of $\Sigma_{\boldsymbol{f}}^*$ decreases to $r$. Importantly, $\ell^* \ge r + 1$ always holds, and so the optimal mixed representation is not deterministic even if $S^{1/2} \Sigma_{\boldsymbol{x}} S^{1/2}$ has less than $r$ significant eigenvalues (which can be represented by a single matrix $R \in \mathbb{R}^{d \times r}$). Hence, the mixed minimax regret is always *strictly lower* than the pure minimax regret. Thus, even when $S = I_d$, and no valuable prior knowledge is known on the response function, the mixed minimax representation is different from the standard PCA solution of top $r$ eigenvectors of $\Sigma_{\boldsymbol{x}}$.

4. Uniqueness of the optimal representation: Since one can always post-multiply $R^\top x$ by some invertible matrix, and then pre-multiply $z = R^\top x$ by its inverse, the following simple observation holds: When $\mathcal{R}$ and $\mathcal{Q}$ are not further restricted, then if $\boldsymbol{R}$ is a minimax representation, and $W(\boldsymbol{R}) \in \mathbb{R}^{r \times r}$ is an invertible matrix, then $\boldsymbol{R} \cdot W(\boldsymbol{R})$ is also a minimax representation.

5. Infinite-dimensional features: Theorems 2 and 3 assume a finite dimensional feature space, but as we show in Appendix F, the results can be easily generalized to an infinite dimensional Hilbert space $\mathcal{X}$, in the more restrictive setting that the noise $\boldsymbol{n}$ is statistically independent of $\boldsymbol{x}$.

**Example 4.** Assume $S = I_d$, and denote, for brevity, $V \equiv V(\Sigma_{\boldsymbol{x}}) := [v_1, \dots, v_d]$ and $\Lambda \equiv \Lambda(\Sigma_{\boldsymbol{x}}) := \mathrm{diag}(\lambda_1, \dots, \lambda_d)$. The optimal minimax representation in pure strategies (Theorem 2) is

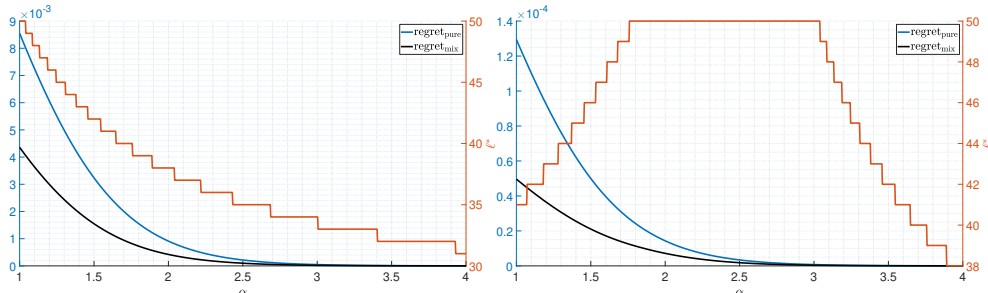

Figure 1: Left: Pure and mixed minimax regret and $\ell_*$ for Example 4, for $d = 50, r = 25$, with $\lambda_i = \sigma_i^2 \propto i^{-\alpha}$. Right: Pure and mixed minimax regret and $\ell_*$ for Example 5, for $d = 50, r = 25$, with $\sigma_i^2 \propto i^{-\alpha}$ and $s_i \propto i^2$. The trend of $\ell_*$ is reversed for $\alpha > 2$.

then

$$R^* = \Sigma_{\boldsymbol{x}}^{-1/2} \cdot V_{1:r} = V\Lambda_{\boldsymbol{x}}^{-1/2}V^\top V_{1:r} = V\Lambda_{\boldsymbol{x}}^{-1/2} \cdot [e_1, \ldots, e_r] = \left[\lambda_1^{-1/2} \cdot v_1, \ldots, \lambda_r^{-1/2} \cdot v_r\right], \tag{15}$$

which is comprised of the top $r$ eigenvectors of $\Sigma_{\boldsymbol{x}}$, scaled so that $v_i^\top \boldsymbol{x}$ has unit variance. By Comment 4 above, $V_{1:r}$ is also an optimal minimax representation. The worst case response is $f = v_{r+1}(\Sigma_{\boldsymbol{x}})$ and, as expected, since $R$ uses the first $r$ principal directions

$$\mathsf{regret}_{\mathsf{pure}}(\mathcal{F} \mid \Sigma_{\boldsymbol{x}}) = \lambda_{r+1}. \tag{16}$$

The minimax regret in mixed strategies (Theorem 3) is different, and given by

$$\mathsf{regret}_{\mathsf{mix}}(\mathcal{F} \mid \Sigma_{\boldsymbol{x}}) = \frac{\ell^* - r}{\sum_{i=1}^{\ell^*} \lambda_i^{-1}}, \tag{17}$$

where $\ell^*$ is determined by the decay rate of the eigenvalues of $\Sigma_{\boldsymbol{x}}$ (see (9)). The least favorable covariance matrix is given by (Theorem 3)

$$\Sigma_{\boldsymbol{f}}^* = \left[\sum_{i=1}^{\ell^*} \lambda_i^{-1}\right]^{-1} \cdot V \operatorname{diag}\left(\lambda_1^{-1}, \ldots, \lambda_{\ell^*}^{-1}, 0, \cdots, 0\right) \cdot V^\top. \tag{18}$$

Intuitively, the least favorable $\Sigma_{\boldsymbol{f}}^*$ equalizes the first $\ell^*$ eigenvalues of $\Sigma_{\boldsymbol{x}}\Sigma_{\boldsymbol{f}}^*$ (and nulls the other $d - \ell^*$) so that the representation is indifferent to these $\ell^*$ directions. As evident from the regret, the "equalization" of the $i$th eigenvalue adds a term of $\lambda_i^{-1}$ to the denominator, and if $\lambda_i$ is too small then $v_i$ is not chosen for the representation, as agrees with Comment 3 above (a fast decay of $\{\lambda_i\}$ reduces $\ell_*$ away from $d$). The mixed minimax representation sets

$$\boldsymbol{R}^* = \Sigma_{\boldsymbol{x}}^{-1/2} \cdot V_{\mathcal{I}_j} = \left[\lambda_{i_{j,1}}^{-1/2} \cdot v_{i_{j,1}}, \ldots, \lambda_{i_{j,r}}^{-1/2} \cdot v_{i_{j,r}}\right] \tag{19}$$

with probability $p_j$, where $\mathcal{I}_j \equiv \{i_{j,1}, \ldots, i_{j,r}\}$ (the derivation is similar to (15)). Thus, the optimal representation chooses a random subset of $r$ vectors from $\{v_1, \ldots, v_{\ell^*}\}$. See the left panel of Figure 1 for a numerical example.

**Example 5.** To demonstrate the effect of prior knowledge on the response function, we assume $\Sigma_{\boldsymbol{x}} = \operatorname{diag}(\sigma_1^2, \ldots, \sigma_d^2)$ and $S = \operatorname{diag}(s_1, \ldots, s_d)$, where $\sigma_1^2 \geq \sigma_2^2 \geq \cdots \geq \sigma_d^2$ (but $\{s_i\}_{i \in [d]}$ are not necessarily ordered). Letting $f = (f_1, \ldots, f_d)$, the class of response functions is $\mathcal{F}_S := \{f \in \mathbb{R}^d : \sum_{i=1}^d (f_i^2/s_i) \leq 1\}$, and so coordinates $i \in [d]$ with a large $s_i$ have large influence on the response. Let $(i_{(1)}, \ldots, i_{(d)})$ be a permutation of $[d]$ so that $\sigma_{i(j)}^2 s_{i(j)}$ it the $j$th largest value of $(\sigma_i^2 s_i)_{i \in [d]}$. The pure minimax regret is (Theorem 2)

$$\mathsf{regret}_{\mathsf{pure}}(\mathcal{F} \mid \Sigma_{\boldsymbol{x}}) = \sigma_{i_{r+1}}^2 s_{i_{r+1}}. \tag{20}$$

The optimal representation is $R = [e_{i_{(1)}}, e_{i_{(2)}}, \ldots, e_{i_{(r)}}]$, that is, uses the most influential coordinates, according to $\{s_i\}$, which may be different from the $r$ principal directions of $\Sigma_{\boldsymbol{x}}$. For the minimax regret in mixed strategies, Theorem 3 results

$$\mathsf{regret}_{\mathsf{mix}}(\mathcal{F} \mid \Sigma_{\boldsymbol{x}}) = \frac{\ell^* - r}{\sum_{j=1}^{\ell^*} (s_{i_j} \sigma_{i_j}^2)^{-1}} \tag{21}$$

for $\ell^* \in [d]\backslash[r]$ satisfying (9), and the covariance matrix of the least favorable prior is given by

$$\Sigma_{\boldsymbol{f}}^* = \frac{\sum_{j=1}^{\ell^*} \sigma_{i_j}^{-2} \cdot e_{i_j} e_{i_j}^\top}{\sum_{j=1}^{\ell^*} (s_{i_j} \sigma_{i_j}^2)^{-1}}. \tag{22}$$

That is, up to a scale factor $(\sum_{i=1}^{\ell^*} s_i^{-1} \sigma_i^{-2})^{-1}$, the matrix is diagonal so that the $k$th term on the diagonal is $\Sigma_{\boldsymbol{f}}^*(k,k) = \sigma_k^{-2}$ if $k = i_j$ for some $j \in [\ell^*]$ and $\Sigma_{\boldsymbol{f}}^*(k,k) = 0$ otherwise. As in Example 4, $\Sigma_{\boldsymbol{f}}^*$ equalizes the first $\ell^*$ eigenvalues of $\Sigma_{\boldsymbol{x}} \Sigma_{\boldsymbol{f}}$ (and nulls the other $d - \ell^*$). However, it does so in a manner that chooses them according to their influence on $\boldsymbol{f}^\top \boldsymbol{x}$. The random minimax representation in mixed strategies is

$$\boldsymbol{R}^* = \left[ \sigma_{i_{j,1}}^{-1} \cdot e_{i_{j,1}}, \ldots, \sigma_{i_{j,r}}^{-1} \cdot e_{i_{j,r}} \right] \tag{23}$$

with probability $p_j$. Again, all the first $\ell^*$ coordinates are used, and not just the top $r$. See the right panel of Figure 1 for a numerical example. We finally remark that, naturally, in the non-diagonal case, the minimax regret will also depend on the relative alignment between $S$ and $\Sigma_{\boldsymbol{x}}$.

# 4 An iterative algorithm for general classes and loss functions

In this section, we develop an iterative algorithm for finding the optimal representation in mixed strategies, i.e., solving (2) for general classes and loss functions. Since optimizing general probability measures over $\mathcal{R}$ is formidable, we restrict the optimization to finite mixed representations, i.e., assume that $\boldsymbol{R} = R^{(j)} \in \mathcal{R}$ with probability $p^{(j)}$, where $j \in [m]$ (which suffices for the linear MSE setting of Section 3, but possibly sub-optimal in general). Furthermore, the algorithm's operation will require randomization also for the response player, and so we set $\boldsymbol{f} = f^{(i)} \in \mathcal{F}$ with probability $o^{(i)}$ where $i \in [\overline{m}]$, and $\overline{m} = m_0 + m$ for some $m_0 \geq 0$. The resulting optimization problem then becomes

$$\min_{\{p^{(j)}, R^{(j)} \in \mathcal{R}\}} \max_{\{o^{(i)}, f^{(i)} \in \mathcal{F}\}} \min_{\{Q^{(j,i)} \in \mathcal{Q}\}} \sum_{j \in [m]} \sum_{i \in [\overline{m}]} p^{(j)} \cdot o^{(i)} \cdot \mathbb{E}\left[ \mathsf{loss}(f^{(i)}(\boldsymbol{x}), Q^{(j,i)}(R^{(j)}(\boldsymbol{x}))) \right], \tag{24}$$

under the constraints $p^{(j)} \geq 0$ and $\sum_j p^{(j)} = 1$, and $o^{(i)} \geq 0$ and $\sum_i o^{(i)} = 1$. Note that the prediction rule $Q^{(j,i)}$ is determined based on both $R^{(j)}$ and $f^{(i)}$, and that the ultimate goal of solving (24) is just to extract the optimal $\boldsymbol{R}$.

A high level description of the algorithm is to gradually add more representations to the support size of $\boldsymbol{R}$ up to $m$, where next $k$ will denote the current number of representations, $k \in [m]$. Initialization requires an representation $R^{(1)}$, as well as a *set* of functions $\{f^{(i)}\}_{i \in m_0}$, so that the final support size of $\boldsymbol{f}$ will be $\overline{m} = m_0 + m$. Finding this initial representation and the set of functions is based on the specific loss function and a possible set of representation/predictors. At iteration $k \in [m]$, the main loop of the algorithm has two phases. In the first phase, a new adversarial function is added to the set of functions, as the worse function for the current random representation. In the second phase, a new representation atom is added to the set of possible representations. This representation is determined based on the given set of functions. Concretely, the two phases operate as follows:

- Phase 1 – Given $k$ representations $\{R^{(j)}\}_{j \in (k)}$ with weights $\{p^{(j)}\}_{j \in [k]}$, the algorithm determines the function $f^{(m_0+k)}$ as the worst function for this random representation (optimal adversarial action of the response function player). Specifically,

$$\mathsf{reg}_k := \mathsf{regret}_{\mathsf{mix}}(\{R^{(j)}, p^{(j)}\}_{j \in [k]}, \mathcal{F} \mid P_{\boldsymbol{x}}) \tag{25}$$

$$:= \max_{f \in \mathcal{F}} \min_{\{Q^{(j)} \in \mathcal{Q}\}_{j \in [k]}} \sum_{j \in [k]} p^{(j)} \cdot \mathbb{E}\left[ \mathsf{loss}(f(\boldsymbol{x}), Q^{(j)}(R^{(j)}(\boldsymbol{x}))) \right] \tag{26}$$

is solved, and $f^{(m_0+k)}$ is set to be the maximizer. This simplifies (24) in the sense that $m$ is replaced by $k$, the random representation $\boldsymbol{R}$ is kept fixed, and $f \in \mathcal{F}$ is optimized as a pure strategy (the previous functions $\{f^{(i)}\}_{i \in [m_0+k-1]}$ are ignored).

- Phase 2 – Adding a representation atom: Given fixed $\{f^{(j)}\}_{j\in[m_0+k]}$ and $\{R^{(j)}\}_{j\in[k]}$, a new representation $R^{(k+1)}$ is found as the most incrementally valuable representation atom. Specifically,

$$\min_{R^{(k+1)}\in\mathcal{R}} \mathsf{regret}_{\mathsf{mix}}(\{R^{(j_1)}\}_{j_1\in[k+1]}, \{f^{(j_2)}\}_{j_2\in[m_0+k]} \mid P_{\boldsymbol{x}})$$

$$:= \min_{R^{(k+1)}\in\mathcal{R}} \min_{\{p^{(j_1)}\}_{j_1\in[k+1]}} \max_{\{o^{(j_2)}\}_{j_2\in[m_0+k]}} \min_{\{Q^{(j_1,j_2)}\in\mathcal{Q}\}_{j_1\in[k+1],j_2\in[m_0+k]}}$$

$$\sum_{j_1\in[k+1]}\sum_{j_2\in[m_0+k]} p^{(j_1)}\cdot o^{(j_2)}\cdot\mathbb{E}\left[\mathsf{loss}(f^{(j_1)}(\boldsymbol{x}), Q^{(j_1,j_2)}(R^{(j_1)}(\boldsymbol{x})))\right] \quad (27)$$

is solved, the solution $R^{(k+1)}$ is added to the set of representations, and the weights are updated to the optimal $\{p^{(j_1)}\}_{j_1\in[k+1]}$. Compared to (24), here the response functions and current $k$ representations are kept fixed, and only their weights $\{p^{(j_1)}\}$ $\{o^{(j_2)}\}$ are optimized, along with $R^{(k+1)}$.

The procedure is described in Algorithm 1, where, following the main loop, $m^* = \arg\min_{k\in[m]}\mathsf{reg}_k$ representation atoms are chosen and the output is $\{R^{(j)}, p^{(j)}\}_{j\in[m^*]}$. Algorithm 1 relies on solvers for the Phase 1 (26) and Phase 2 (27) problems. In Appendix G we propose two algorithms for these problems, which are based on gradient steps for updating the adversarial response and the new representation, and on the MWU algorithm [33] (*follow-the-regularized-leader* [35]) for updating the weights. In short, the Phase 1 algorithm updates the response function $f$ via a projected gradient step of the expected loss, and then adjusts the predictors $\{Q^{(j)}\}$ to the updated response function $f$ and the current representations $\{R^{(j)}\}_{j\in[k]}$. The Phase 2 algorithm only updates the new representation $R^{(k+1)}$ via projected gradient steps, while keeping $\{R^{(j)}\}_{j\in[k]}$ fixed. Given the representations $\{R^{(j)}\}_{j\in[k+1]}$ and the functions $\{f^{(i)}\}_{i\in[m_0+k]}$, a predictor $Q^{(j,i)}$ is then fitted to each representation-function pair, which also determines the loss for this pair. The weights $\{p^{(j)}\}_{j\in[k+1]}$ and $\{o^{(i)}\}_{i\in[m_0+k]}$ are updated towards the equilibrium of the two-player game determined by the loss of the predictors $\{Q^{(j,i)}\}_{j\in[k+1],i\in[m_0+k]}$ via the MWU algorithm.

---

**Algorithm 1** Solver of (24): An iterative algorithm for learning mixed representations.

---

1: **input** $P_{\boldsymbol{x}}, \mathcal{R}, \mathcal{F}, \mathcal{Q}, d, r, m, m_0$ ▷ Feature distribution, classes, dimensions and parameters
2: **input** $R^{(1)}, \{f^{(j)}\}_{j\in[m_0]}$ ▷ Initial representation and initial function (set)
3: **begin**
4: **for** $k = 1$ to $m$ **do**
5:     **phase 1:** $f^{(m_0+k)}$ is set by a solver of (26) and

$$\mathsf{reg}_k \leftarrow \mathsf{regret}_{\mathsf{mix}}(\{R^{(j)}, p^{(j)}\}_{j\in[k]}, \mathcal{F} \mid P_{\boldsymbol{x}}) \quad (28)$$

                                                 ▷ Solved using Algorithm 2
6:     **phase 2:** $R^{(k+1)}, \{p_k^{(j)}\}_{j\in[k+1]}$ is set by a solver of (27) ▷ Solved using Algorithm 3; step can be removed if $k = m$
7: **end for**
8: **set** $m^* = \arg\min_{k\in[m]}\mathsf{reg}_k$
9: **return** $\{R^{(j)}\}_{j\in[m^*]}$ **and** $p_{m_*} = \{p_k^{(j)}\}_{j\in[m^*]}$

---

We next outline two examples, where full details can be found in Appendix H.

**Example 6.** We validate that efficiency of Algorithm 1 in the linear MSE setting (Section 3), for which a closed-form solution exists. We ran Algorithm 1 on randomly drawn diagonal $\Sigma_{\boldsymbol{x}}$, and computed the ratio between the regret obtained by the algorithm to the theoretical value. The left panel of Figure 2 shows that the ratio is between $1.15 - 1.2$ in a wide range of $d$ values. We mention again that Algorithm 1 is useful even for this setting since finding an $(\ell^* + 1)$-sparse solution to $\overline{A}p = \overline{b}$ is computationally difficult when $\binom{\ell^*}{r}$ is very large. For example, in the largest dimension of the experiment, the potential number of representation matrices is $\binom{d}{r} = \binom{19}{5} = 11,628$.

Our next example pertains to a logistic regression setting, under the cross-entropy loss function.

**Definition 7** (The linear cross-entropy setting). Assume that $\mathcal{X} = \mathbb{R}^d$, that $\mathcal{Y} = \{\pm 1\}$ and that $\mathbb{E}[\boldsymbol{x}] = 0$. Assume that the class of representation is linear $z = R(x) = R^\top x$ for some $R \in \mathcal{R} :=$

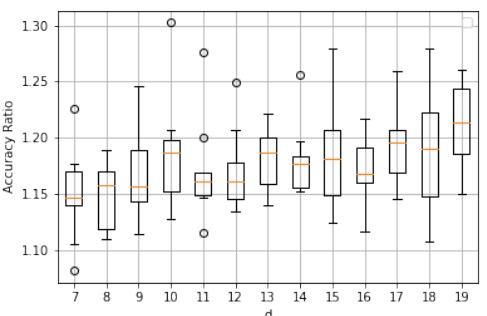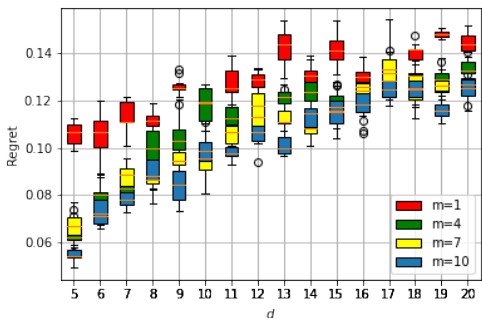

Figure 2: Results of Algorithm 1. Left: $r = 5$, varying $d$. The ratio between the regret achieved by Algorithm 1 and the theoretical regret in the linear MSE setting. Right: $r = 3$, varying $d$. The regret achieved by Algorithm 1 in the linear cross entropy setting, various $m$.

$\mathbb{R}^{d \times r}$ where $d > r$. Assume that a response function and a prediction rule determine the probability that $y = 1$ via logistic regression modeling, as $f(\boldsymbol{y} = \pm 1 \mid x) = 1/[1 + \exp(\mp f^\top x)]$. Assume the cross-entropy loss function, where given that the prediction that $\boldsymbol{y} = 1$ with probability $q$ results the loss $\mathsf{loss}(y, q) := -\frac{1}{2}(1 + y)\log q - \frac{1}{2}(1 - y)\log(1 - q)$. The set of predictor functions is $\mathcal{Q} := \{Q(z) = 1/[1 + \exp(-q^\top z)], \ q \in \mathbb{R}^r\}$. As for the linear case, we assume that $f \in \mathcal{F}_S$ for some $S \in \mathbb{S}^d_{++}$. It is not difficult to show that the regret is then given by the expected binary Kullback-Leibler (KL) divergence

$$\mathsf{regret}(R, f \mid P_{\boldsymbol{x}}) = \min_{q \in \mathbb{R}^r} \mathbb{E}\left[D_{\mathsf{KL}}\left([1 + \exp(-f^\top \boldsymbol{x})]^{-1} \;\|\; [1 + \exp(-q^\top R^\top \boldsymbol{x})]^{-1}\right)\right]. \quad (29)$$

**Example 8.** We ran Algorithm 1 on empirical distributions of features drawn from an isotropic normal distribution, in the linear cross-entropy setting. Algorithm 1 is suitable in this setting since gradients of the regret have closed-form (see Appendix H). The right panel of Figure 2 shows the reduced regret obtained by increasing the support size $m$ of the random representation, and thus the effectiveness of mixed representations.

We refer the reader to Appendix I for additional experiments with Algorithm 1.

## 5 Conclusion

We proposed a game-theoretic formulation for learning representations of unlabeled features when prior knowledge (or assumptions) on the class of future prediction tasks is available. We focused on the fundamental of linear MSE setting, and derived the optimal solution. Beyond the lower regret that is directly obtained from utilizing the prior knowledge, our results also revealed the importance of using randomized representations. We have then proposed an iterative algorithm suitable for general classes of functions and losses, and exemplified its effectiveness.

We next discuss *limitations* and potential future research: (1) We have focused on the elementary and simplified class $\mathcal{F}_S = \{f \colon \|f\|_S \leq 1\}$, mainly for theoretical investigations. A natural refinement to non-linear functions is the general class $\mathcal{F}_{S_x} := \mathbb{E}\left[\|\nabla_x f(\boldsymbol{x})\|^2_{S_{\boldsymbol{x}}}\right] \leq 1$, where $\{S_x\}_{x \in \mathbb{R}^d}$ is now locally specified (somewhat similarly to the regularization term used in contractive AE [39], though for different reasons). (2) Since the proposed iterative algorithm includes optimization over three players, it is of interest to develop version of the algorithm with lower computational optimization cost. (3) We have assumed that $\mathcal{F}_S$ is given in advance, and a natural follow-up goal is to efficiently learn $S$ from previous experience, e.g., improving $S$ from one episode to another in a meta-learning setup [40]. (4) It is interesting to evaluate the effectiveness of the learned representation in our formulation, as an initialization for further optimization when labeled data is collected. One may postulate that since our learned representation is *uniformly* good for all response functions in the class, it may serve as a universal initialization for such training.

## Broader impact

The research described in this paper is foundational, and does not aim for any specific application. Nonetheless, the learned representation is based on a prior assumption on the class of response functions, and the choice of this prior may have positive or negative impacts: For example, a risk of this choice of prior is that the represented features completely ignore a viable feature for making future predictions. A benefit that can stem from choosing a proper prior is that the representation will null the effect of features that lead to unfair advantages for some particular group, in future predictions. Anyhow, the results presented in the paper are indifferent to such future utilization, and any usage of these results should take into account the aforementioned possible implications.

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
