The outline of the supplementary material is as follows. In Appendix A we discuss in detail the origin of prior knowledge of classes of prediction problems. In Appendix B we review additional related work. In Appendix C we set our notation conventions. In Appendix D we summarize a few mathematical results that are used in later proofs. In Section E we show that PCA can be cast as a degenerate setting of our formulation, and provide the proofs of the main theorems in the paper (the linear MSE setting). In Appendix F we generalize these results to an infinite dimensional Hilbert space. In Appendix G we provide two algorithms for solving the Phase 1 and Phase 2 problems in Algorithm 1. In Appendix H we provide details on the examples for the experiments with the iterative algorithm. In Appendix I we describe an experiment in which the representation, response function and predictors are modeled as a neural network (NN).

## A    Classes of response functions

As said, our approach to optimal representation is based on the assumption that a class $\mathcal{F}$ of future prediction tasks is known. This assumption may represent prior knowledge or constraints on the response function, and can stem from various considerations. To begin, it might be hypothesized that some features are less relevant than others. As a simple intuitive example, the outer pixels in images are typically less relevant to the classification of photographed objects, regardless of their variability (which may stem from other affects, such as lighting conditions). Similarly, non-coding regions of the genotype are irrelevant for predicting phenotype. The prior knowledge may encode softer variations in relevance. Moreover, such prior assumption may be imposed on the learned function, e.g., it may be assumed that the response function respects the privacy of some features, or only weakly depends on features which provide an unfair advantage. In domain adaptation [**?** ], one may solve the prediction problem for feature distribution $P_{\boldsymbol{x}}$ obtaining a optimal response function $f_1$. Then, after a change of input distribution to $Q_{\boldsymbol{x}}$, the response function learned for this feature distribution $f_2$ may be assumed to belong to functions which are "compatible" with $f_1$. For example, if $P_{\boldsymbol{x}}$ and $Q_{\boldsymbol{x}}$ are supported on different subsets of $\mathbb{R}^d$, the learned response function $f_1(x)$ and $f_2(x)$ may be assumed to satisfy some type of continuity assumptions. Similar assumptions may hold for the more general setting of transfer learning [41]. Furthermore, such assumptions may hold in a *continual learning* setting [42–45], in which a sequence of response functions is learned one task at a time. Assuming that *catastrophic forgetting* is aimed to be avoided, then starting from the second task, the choice of representation may assume that the learned response function is accurate for all previously learned tasks.

## B    Additional related work

**The information bottleneck principle**    The IB principle is a prominent approach to feature relevance in the design of representations [16–19], and proposes to optimize the representation in order to maximize its relevance to the response $\boldsymbol{y}$. Letting $I(\boldsymbol{z};\boldsymbol{y})$ and $I(\boldsymbol{x};\boldsymbol{z})$ denote the corresponding mutual information terms [27], the IB principle aims to maximize the former while constraining the latter from above, and this is typically achieved via a Lagrangian formulation [46]. The resulting representation, however, is tailored to the joint distribution of $(\boldsymbol{x},\boldsymbol{y})$, i.e., to a specific prediction task. In practice, this is achieved using a labeled dataset (Generalization bounds were derived in [47]). As in our mixed representation approach, the use of randomized representation dictated by a probability kernel $P_{Z|X}$ is inherent to the IB principle. The IB principle was intensively utilized to hypothesize that prediction algorithms, e.g., deep neural networks (DNNs) [1] used for classification, must intrinsically include learning of efficient representations [20–24] (this spurred a debate, see, e.g., [25, 26]). However, this approach is inadequate in an unsupervised setting since the optimal representation depends on the response variable, and so labeled data should be provided when learning the representation. In addition, as explained in [29], while the resulting IB solution provides a fundamental limit for the problem, it also suffers from multiple theoretical and practical issues. The first main issue is that the mutual information terms are inherently difficult to estimate from finite samples [47–51], especially at high dimensions, and thus require resorting to approximations, e.g., variational bounds [52–55]. The resulting generalization bounds [47, 56] are still vacuous for modern settings [57]. The second main issue is that the IB formulation does not constrain the complexity of the representation and the prediction rule, which can be arbitrarily complex. These issues were addressed in [29] using the notion of *usable information*, introduced in [28]: The standard mutual information $I(\boldsymbol{z};\boldsymbol{y})$ can be described as the log-loss difference between a predictor for $\boldsymbol{z}$

which does not use or does use $y$ (or vice-versa, since mutual information is symmetric). Usable information, or $\mathcal{F}$-information $I_{\mathcal{F}}(z \to y)$, restricts the predictor to a class $\mathcal{F}$, which is computationally constrained. Several desirable properties were established in [28] for the $\mathcal{F}$-information, e.g., probably approximate correct (PAC) bounds via Rademacher-complexity based bounds [58] [59, Chapter 5][60, Chapters 26-28]. In [29], the authors used the notion of $\mathcal{F}$-information to define the *decodable IB* problem, with the goal of addressing the generalization capabilities of this IB problem. In order to explore this, the *two-player game* described in the introduction was proposed. Beyond those works, the IB framework has drawn a significant recent attention, and a remarkable number of extensions and ramifications have been proposed [61–70, 55, 71]. IB framework for self-supervised learning was recently discussed in [72].

**Randomization in representation learning**    Randomization is classically used in data representation, most notably, utilizing the seminal Johnson-Lindenstrauss Lemma [73] or more generally, *sketching* algorithms (e.g., [74–77]). Our use of randomization is different and is inspired by the classical Nash equilibrium [78]. Rather than using a single deterministic representation that was randomly chosen, we consider randomizing multiple representation rules. Training approaches based on mixed strategies were proposed, e.g., in the generative adversarial network (GAN) setting [79–81]. Specifically, inspired by the boosting technique [5], it was proposed in [81] to gradually add additional modes to the mix of generative models, and where the new mode added focuses on the distribution samples which are not adequately represented by the current set of modes. As mentioned in [81], this idea dates back to the use of boosting for density estimation [82]. Our proposed iterative algorithm follows this idea, and gradually adds representation rules, so that the new representation aims to cope with response functions that are not adequately fitted by the current set of representation rules.

**Game theoretic formulations in statistics and machine-learning**    The use of game theoretic formulations in statistics, between a player choosing a prediction algorithm and an adversary choosing a prediction problem (typically Nature), was established by Wald in his classical statistical decision theory [83] (see, e.g., [84, Chapter 12]). It is a common approach both in classic statistics and learning theory [85–88], as well as in modern high-dimensional statistics [59]. The effect of the representation (quantizer) on the consistency of learning algorithms when a surrogate convex loss function replaces the loss function of interest was studied in [3, 4, 86] (for binary and multiclass classification, respectively). A relation between information loss and minimal error probability was recently derived in [89].

Iterative algorithms for the solution of minimax games have drawn much attention in the last few years due to their importance in optimizing GANs [90, 91], adversarial training [92], and robust optimization [93]. The notion of convergence is rather delicate, even for the basic convex-concave two-player setting [94]. While the value output by the MWU algorithm [33], or improved versions [95, 96] converges to a no-regret solution, the actual strategies used by the players are, in fact, repelled away from the equilibrium point to the boundary of the probability simplex [97]. For general games, the gradient descent ascent (GDA) is a natural and practical choice, yet despite recent advances, its theory is still partial [98]. Various other algorithms have been proposed, e.g., [99–103]. According to the above description, and since our algorithm is both fairly general and involves two optimization phases, deriving theoretical bounds on its convergence seems to be elusive at this point. Nevertheless, the algorithm is also modular, and its two intermediate phases (see Appendix G) can be easily upgraded to more sophisticated optimization methods. Furthermore, each of the phases can be separately analyzed.

**Unsupervised pretraining**    From a broader perspective, our method is essentially an *unsupervised pretraining* method, similar to the methods which currently enable the recent success in natural language processing. Our model is much simplified compared to transformer architecture Vaswani et al. [104], but the unsupervised training aspect used for prediction tasks Devlin et al. [105] is common, and our results may shed light on these methods. For example, putting more weight on some words compared to others during training phase that uses the masked-token prediction objective.

## C  Notation conventions

For an integer $d$, $[d] := \{1, 2, \ldots, d\}$. For $p \geq 1$, $\|x\|_p := (\sum_{i=1}^d |x_i|^p)^{1/p}$ is the $\ell_p$ norm of $x \in \mathbb{R}^d$. The Frobenius norm of the matrix $A$ is denoted by $\|A\|_F = \sqrt{\mathrm{Tr}[A^T A]}$. The non-negative (resp. positive) definite cone of symmetric matrices is given by $\mathbb{S}_+^d$ (resp. $\mathbb{S}_{++}^d$). For a given positive-definite matrix $S \in \mathbb{S}_{++}^d$, the Mahalanobis norm of $x \in \mathbb{R}^d$ is given by $\|x\|_S := \|S^{-1/2}x\|_2 = (x^\top S^{-1}x)^{1/2}$, where $S^{1/2}$ is the symmetric square root of $S$. The matrix $W := [w_1, \ldots, w_r] \in \mathbb{R}^{d \times r}$ is comprised from the column vectors $\{w_i\}_{i \in [r]} \subset \mathbb{R}^d$. For a real symmetric matrix $S \in \mathbb{S}^d$, $\lambda_i(S)$ is the $i$th largest eigenvalue, so that $\lambda_{\max}(S) \equiv \lambda_1(S) \geq \lambda_2(S) \geq \cdots \geq \lambda_d(S) = \lambda_{\min}(S)$, and in accordance, $v_i(S)$ denote an eigenvector corresponding to $\lambda_i(S)$ (these are unique if there are no two equal eigenvalues, and otherwise arbitrarily chosen, while satisfying orthogonality $v_i^\top v_j = \langle v_i, v_j \rangle = \delta_{ij}$). Similarly, $\Lambda(S) := \mathrm{diag}(\lambda_1(S), \lambda_2(S), \cdots, \lambda_d(S))$ and $V(S) := [v_1(S), v_2(S), \cdots, v_d(S)]$, so that $S = V(S)\Lambda(S)V^\top(S)$ is an eigenvalue decomposition. For $j \geq i$, $V_{i:j} := [v_i, \ldots, v_j] \in \mathbb{R}^{(j-i+1) \times d}$ is the matrix comprised of the columns indexed by $\{i, \ldots, j\}$. The vector $e_i \in \mathbb{R}^d$ is the $i$th standard basis vector, that is, $e_i := [\underbrace{0, \ldots 0}_{i-1 \text{ terms}}, 1, \underbrace{0, \ldots 0}_{d-i \text{ terms}}]^\top$.

Random quantities (scalars, vectors, matrices, etc.) are denoted by boldface letters. For example, $\boldsymbol{x} \in \mathbb{R}^d$ is a random vector that takes values $x \in \mathbb{R}^d$ and $\boldsymbol{R} \in \mathbb{R}^{d \times r}$ is a random matrix. The probability law of a random element $\boldsymbol{x}$ is denoted by $\mathsf{L}(\boldsymbol{x})$. The probability of the event $\mathcal{E}$ in some given probability space is denoted by $\mathbb{P}[\mathcal{E}]$ (typically understood from context). The expectation operator is denoted by $\mathbb{E}[\cdot]$. The indicator function is denoted by $\mathbb{1}\{\cdot\}$, and the Kronecker delta is denoted by $\delta_{ij} := \mathbb{1}\{i = j\}$. We do not make a distinction between minimum and infimum (or maximum and supremum) as arbitrarily accurate approximation is sufficient for the description of the results in this paper. The binary KL divergence between $p_1, p_2 \in (0, 1)$ is denoted as

$$D_{\mathrm{KL}}(p_1 \| p_2) := p_1 \log \frac{p_1}{p_2} + (1 - p_1) \log \frac{1 - p_1}{1 - p_2}. \tag{30}$$

## D  Useful mathematical results

In this section we provide several simplified versions of mathematical results that are used in the proofs. The following well-known result is about the optimal low-rank approximation to a given matrix:

**Theorem 9** (*Eckart-Young-Mirsky* [59, Example 8.1] [106, Section 4.1.4]). *For a symmetric matrix* $S \in \mathbb{S}^d$

$$\|S_k - S\|_F \leq \min_{S' \in \mathbb{S}^d : \mathrm{rank}(S') \leq k} \|S - S'\|_F \tag{31}$$

*where*

$$S_k = \sum_{i \in [k]} \lambda_i(S) \cdot v_i(S) v_i^\top(S) \tag{32}$$

*(more generally, this is true for any unitarily invariant norm).*

We next review a simplified version of variational characterizations of eigenvalues of symmetric matrices:

**Theorem 10** (*Rayleigh quotient* [37, Theorem 4.2.2]). *For a symmetric matrix* $S \in \mathbb{S}^d$

$$\lambda_1(S) = \max_{x \neq 0} \frac{x^\top S x}{\|x\|_2^2}. \tag{33}$$

**Theorem 11** (*Courant–Fisher variational characterization* [37, Theorem 4.2.6]). *For a symmetric matrix* $S \in \mathbb{S}^d$, $k \in [d]$, *and a subspace* $T$ *of* $\mathbb{R}^d$

$$\lambda_k(S) = \min_{T : \dim(T) = k} \max_{x \in T \setminus \{0\}} \frac{x^\top S x}{\|x\|_2^2} = \max_{T : \dim(T) = d-k+1} \min_{x \in T \setminus \{0\}} \frac{x^\top S x}{\|x\|_2^2}. \tag{34}$$

**Theorem 12** (*Fan's variational characterization [37, Corollary 4.3.39.]*)*. For a symmetric matrix $S \in \mathbb{S}^d$ and $k \in [d]$*

$$\lambda_1(S) + \cdots + \lambda_k(S) = \min_{U \in \mathbb{R}^{d \times k} : U^\top U = I_k} \mathrm{Tr}[U^\top S U] \tag{35}$$

*and*

$$\lambda_{d-k+1}(S) + \cdots + \lambda_d(S) = \max_{U \in \mathbb{R}^{d \times k} : U^\top U = I_k} \mathrm{Tr}[U^\top S U]. \tag{36}$$

We will use the following celebrated result from convex analysis.

**Theorem 13** (*Carathéodory's theorem [107, Prop. 1.3.1]*)*. Let $\mathcal{A} \subset \mathbb{R}^d$ be non-empty. Then, any point $a$ in the convex hull of $\mathcal{A}$ can be written as a convex combination of at most $d + 1$ points from $\mathcal{A}$.*

# E    The linear MSE setting: additions and proofs

## E.1    The standard principal component setting

In order to highlight the formulation proposed in this paper, we show, as a starting point, that the well known PCA solution of representing $x \in \mathbb{R}^d$ with the top $r$ eigenvectors of the covariance matrix of $x$ can be obtained as a specific case of the regret formulation. In this setting, we take $\mathcal{F} = \{I_d\}$, and so $y = x$ with probability 1. In addition, the predictor class $\mathcal{Q}$ is a linear function from the representation dimension $r$ back to the features dimension $d$.

**Proposition 14.** *Consider the linear MSE setting, with the difference that the response is $y \in \mathbb{R}^d$, the loss function is the MSE $\mathsf{loss}(y_1, y_2) = \|y_1 - y_2\|^2$, and the predictor is $Q(z) = Q^\top z \in \mathbb{R}^d$ for $Q \in \mathbb{R}^{r \times d}$. Assume $\mathcal{F} = \{I_d\}$ so that $y = x$ with probability 1. Then,*

$$\mathsf{regret}_{\mathsf{pure}}(\mathcal{F} \mid \Sigma_x) = \mathsf{regret}_{\mathsf{mix}}(\mathcal{F} \mid \Sigma_x) = \sum_{i=r+1}^{d} \lambda_i(\Sigma_x), \tag{37}$$

*and an optimal representation is $R = V_{1:r}(\Sigma_x)$.*

The result of Proposition 14 verifies that the minimax and maximin formulations indeed generalize the standard PCA formulation. The proof is standard and follows from the *Eckart-Young-Mirsky theorem* (e.g.,[59, Example 8.1] [106, Section 4.1.4]), which determines the best rank $r$ approximation in the Frobenius norm.

*Proof of Proposition 14.* Since $\mathcal{F} = \{I_d\}$ is a singleton, there is no distinction between pure and mixed minimax regret. It holds that

$$\mathsf{regret}(R, f) = \mathbb{E}\left[\|x - Q^\top R^\top x\|^2\right] \tag{38}$$

where $A = Q^\top R^\top \in \mathbb{R}^{d \times d}$ is a rank $r$ matrix. For any $A \in \mathbb{R}^{d \times d}$

$$\mathbb{E}\left[\|x - Ax\|^2\right] = \mathbb{E}\left[x^\top x - x^\top A x - x^\top A^\top x + x^\top A^\top A x\right] \tag{39}$$

$$= \mathrm{Tr}\left[\Sigma_x - A\Sigma_x - A^\top \Sigma_x + A^\top A \Sigma_x\right] \tag{40}$$

$$= \left\|\Sigma_x^{1/2} - \Sigma_x^{1/2} A\right\|_F^2 \tag{41}$$

$$= \left\|\Sigma_x^{1/2} - B\right\|_F^2, \tag{42}$$

where $B := \Sigma_x^{1/2} A$. By the classic *Eckart-Young-Mirsky theorem* [59, Example 8.1] [106, Section 4.1.4] (see Appendix D), the best rank $r$ approximation in the Frobenius norm is obtained by setting

$$B^* = \sum_{i=1}^{r} \lambda_i(\Sigma_x^{1/2}) \cdot v_i v_i^\top = \sum_{i=1}^{r} \sqrt{\lambda_i(\Sigma_x)} \cdot v_i v_i^\top \tag{43}$$

where $v_i \equiv v_i(\Sigma_x^{1/2}) = v_i(\Sigma_x)$ is the $i$th eigenvector of $\Sigma_x^{1/2}$ (or $\Sigma_x$). Then, the optimal $A$ is

$$A^* = \sum_{i=1}^{r} \sqrt{\lambda_i(\Sigma_x)} \cdot \Sigma_x^{-1/2} v_i v_i^\top = \sum_{i=1}^{r} \sqrt{\lambda_i(\Sigma_x)} \cdot \Sigma_x^{-1/2} v_i v_i^\top = \sum_{i=1}^{r} v_i v_i^\top, \tag{44}$$

since $v_i$ is also an eigenvector of $\Sigma_{\boldsymbol{x}}^{-1/2}$. Letting $R = U(R)\Sigma(R)V^\top(R)$ and $Q = U(Q)\Sigma(Q)V^\top(Q)$ be the singular value decomposition of $R$ and $Q$, respectively, it holds that

$$Q^\top R^\top = V(Q)\Sigma^\top(Q)V(Q)V(R)\Sigma^\top(R)U^\top(R). \tag{45}$$

Setting $V(Q) = V(R) = I_r$, and $\Sigma^\top(Q) = \Sigma(R) \in \mathbb{R}^{d \times r}$ to have $r$ ones on the diagonal (and all other entries are zero), as well as $U(Q) = U(R)$ to be an orthogonal matrix whose first $r$ columns are $\{v_i\}_{i \in [r]}$ results that $Q^\top R^\top = A^*$, as required. $\qquad\square$

### E.2 Proofs of pure and mixed minimax representations

Before the proof of Theorem 2, we state a simple and useful lemma, which provides the pointwise value of the regret and the optimal linear predictor for a given representation and response.

**Lemma 15.** *Consider the representation $\boldsymbol{z} = R^\top \boldsymbol{x} \in \mathbb{R}^r$. It then holds that*

$$\min_{q \in \mathbb{R}^r} \mathbb{E}\left[\left(f^\top \boldsymbol{x} + \boldsymbol{n} - q^\top \boldsymbol{z}\right)^2\right] \tag{46}$$

$$= \mathbb{E}\left[\mathbb{E}[\boldsymbol{n}^2 \mid \boldsymbol{x}]\right] + f^\top \left(\Sigma_{\boldsymbol{x}} - \Sigma_{\boldsymbol{x}}R(R^\top\Sigma_{\boldsymbol{x}}R)^{-1}R^\top\Sigma_{\boldsymbol{x}}\right)f. \tag{47}$$

*Proof.* The orthogonality principle states that

$$\mathbb{E}\left[\left(f^\top \boldsymbol{x} + \boldsymbol{n} - q^\top \boldsymbol{z}\right) \cdot \boldsymbol{z}^\top\right] = 0 \tag{48}$$

must hold for the optimal linear estimator. Using $\boldsymbol{z} = R^\top \boldsymbol{x}$ and taking expectations leads to the standard least-squares (LS) solution

$$q_{\mathrm{LS}} = (R^\top\Sigma_{\boldsymbol{x}}R)^{-1}R^T\Sigma_{\boldsymbol{x}}f, \tag{49}$$

assuming that $R^\top\Sigma_{\boldsymbol{x}}R$ is invertible (which we indeed assume as if this is not the case, the representation can be reduced to a dimension lower than $r$ in a lossless manner). The resulting regret of $R$ is thus given by

$$\mathsf{regret}(R, f) = \mathbb{E}\left[\left(f^\top \boldsymbol{x} + \boldsymbol{n} - q_{\mathrm{LS}}^\top \boldsymbol{z}\right)^2\right] \tag{50}$$

$$\stackrel{(a)}{=} \mathbb{E}\left[\left(f^\top \boldsymbol{x} + \boldsymbol{n}\right)^\top \left(f^\top \boldsymbol{x} + \boldsymbol{n} - q_{\mathrm{LS}}^\top \boldsymbol{z}\right)\right] \tag{51}$$

$$= \mathbb{E}\left[\left(f^\top \boldsymbol{x} + \boldsymbol{n}\right)^2 - \left(f^\top \boldsymbol{x} + \boldsymbol{n}\right)^\top q_{\mathrm{LS}}^\top \boldsymbol{z}\right] \tag{52}$$

$$\stackrel{(b)}{=} \mathbb{E}\left[\mathbb{E}[\boldsymbol{n}^2 \mid \boldsymbol{x}]\right] + f^\top\Sigma_{\boldsymbol{x}}f - \mathbb{E}\left[\boldsymbol{x}^\top f q_{\mathrm{LS}}^\top R^\top \boldsymbol{x}\right] \tag{53}$$

$$= \mathbb{E}\left[\mathbb{E}[\boldsymbol{n}^2 \mid \boldsymbol{x}]\right] + f^\top\Sigma_{\boldsymbol{x}}f - \mathrm{Tr}\left[f q_{\mathrm{LS}}^\top R^\top \Sigma_{\boldsymbol{x}}\right] \tag{54}$$

$$= \mathbb{E}\left[\mathbb{E}[\boldsymbol{n}^2 \mid \boldsymbol{x}]\right] + f^\top\Sigma_{\boldsymbol{x}}f - q_{\mathrm{LS}}^\top R^\top \Sigma_{\boldsymbol{x}}f \tag{55}$$

$$\stackrel{(c)}{=} \mathbb{E}\left[\mathbb{E}[\boldsymbol{n}^2 \mid \boldsymbol{x}]\right] + f^\top \left(\Sigma_{\boldsymbol{x}} - \Sigma_{\boldsymbol{x}}R(R^\top\Sigma_{\boldsymbol{x}}R)^{-1}R^\top\Sigma_{\boldsymbol{x}}\right)f, \tag{56}$$

where $(a)$ follows from the orthogonality principle in (48), $(b)$ follows from the tower property of conditional expectation and since $\mathbb{E}[\boldsymbol{x}\boldsymbol{n}] = \mathbb{E}[\boldsymbol{x} \cdot \mathbb{E}[\boldsymbol{n} \mid \boldsymbol{x}]] = 0$, and $(c)$ follows by substituting $q_{\mathrm{LS}}$ from (49). $\qquad\square$

We may now prove Theorem 2.

*Proof of Theorem 2.* For any given $f$, the optimal predictor based on $x \in \mathbb{R}^d$ achieves average loss of

$$\min_{q \in \mathbb{R}^d} \mathbb{E}\left[\left(f^\top \boldsymbol{x} + \boldsymbol{n} - q^\top \boldsymbol{x}\right)^2\right] = \mathbb{E}\left[\mathbb{E}[\boldsymbol{n}^2 \mid \boldsymbol{x}]\right] \tag{57}$$

(specifically, this is obtained by setting $R = I_d$ in Lemma 15 so that $\boldsymbol{z} = \boldsymbol{x}$). Hence, the resulting regret of $R$ over an adversarial choice of $f \in \mathcal{F}_S$ is

$$\max_{f \in \mathcal{F}_S} \mathsf{regret}(R, f) = \max_{f \in \mathcal{F}} \mathbb{E}\left[\left|f^\top \boldsymbol{x} + \boldsymbol{n} - q_{\mathrm{LS}}^\top \boldsymbol{z}\right|^2\right] - \mathbb{E}\left[\mathbb{E}[\boldsymbol{n}^2 \mid \boldsymbol{x}]\right]$$

$$\overset{(a)}{=} \max_{f \in \mathcal{F}_S} f^\top \left( \Sigma_{\boldsymbol{x}} - \Sigma_{\boldsymbol{x}} R(R^\top \Sigma_{\boldsymbol{x}} R)^{-1} R^\top \Sigma_{\boldsymbol{x}} \right) f \tag{58}$$

$$\overset{(b)}{=} \max_{\tilde{f}: \|\tilde{f}\|_2^2 \leq 1} \tilde{f}^\top \left( S^{1/2} \Sigma_{\boldsymbol{x}} S^{1/2} - S^{1/2} \Sigma_{\boldsymbol{x}} R(R^\top \Sigma_{\boldsymbol{x}} R)^{-1} R^\top \Sigma_{\boldsymbol{x}} S^{1/2} \right) \tilde{f} \tag{59}$$

$$\overset{(c)}{=} \lambda_1 \left( S^{1/2} \Sigma_{\boldsymbol{x}} S^{1/2} - S^{1/2} \Sigma_{\boldsymbol{x}} R(R^\top \Sigma_{\boldsymbol{x}} R)^{-1} R^\top \Sigma_{\boldsymbol{x}} S^{1/2} \right) \tag{60}$$

$$= \lambda_1 \left[ S^{1/2} \Sigma_{\boldsymbol{x}}^{1/2} \left( I_d - \Sigma_{\boldsymbol{x}}^{1/2} R(R^\top \Sigma_{\boldsymbol{x}} R)^{-1} R^\top \Sigma_{\boldsymbol{x}}^{1/2} \right) \Sigma_{\boldsymbol{x}}^{1/2} S^{1/2} \right] \tag{61}$$

$$\overset{(d)}{=} \lambda_1 \left[ S^{1/2} \Sigma_{\boldsymbol{x}}^{1/2} \left( I_d - \tilde{R}(\tilde{R}^\top \tilde{R})^{-1} \tilde{R}^\top \right) \Sigma_{\boldsymbol{x}}^{1/2} S^{1/2} \right] \tag{62}$$

$$\overset{(e)}{=} \lambda_1 \left[ \left( I_d - \tilde{R}(\tilde{R}^\top \tilde{R})^{-1} \tilde{R}^\top \right) \Sigma_{\boldsymbol{x}}^{1/2} S \Sigma_{\boldsymbol{x}}^{1/2} \left( I_d - \tilde{R}(\tilde{R}^\top \tilde{R})^{-1} \tilde{R}^\top \right) \right], \tag{63}$$

where $(a)$ follows from Lemma 15, $(b)$ follows by letting $\tilde{f} := S^{-1/2} f$ and recalling that any $f \in \mathcal{F}$ must satisfy $\|f\|_S^2 \leq 1$, $(c)$ follows from the *Rayleigh quotient theorem* [37, Theorem 4.2.2] (see Appendix D), $(d)$ follows by letting $\tilde{R} := \Sigma_{\boldsymbol{x}}^{1/2} R$, and $(e)$ follows since $I_d - \tilde{R}(\tilde{R}^\top \tilde{R})^{-1} \tilde{R}^\top$ is an orthogonal projection (idempotent and symmetric matrix) of rank $d - r$.

Now, to find the minimizer of $\max_{f \in \mathcal{F}_S} \mathrm{regret}(R, f)$ over $R$, we note that

$$\lambda_1 \left[ \left( I_d - \tilde{R}(\tilde{R}^\top \tilde{R})^{-1} \tilde{R}^\top \right) \Sigma_{\boldsymbol{x}}^{1/2} S \Sigma_{\boldsymbol{x}}^{1/2} \left( I_d - \tilde{R}(\tilde{R}^\top \tilde{R})^{-1} \tilde{R}^\top \right) \right]$$

$$\overset{(a)}{=} \max_{u: \|u\|_2 = 1} u^\top \left( I_d - \tilde{R}(\tilde{R}^\top \tilde{R})^{-1} \tilde{R}^\top \right) \Sigma_{\boldsymbol{x}}^{1/2} S \Sigma_{\boldsymbol{x}}^{1/2} \left( I_d - \tilde{R}(\tilde{R}^\top \tilde{R})^{-1} \tilde{R}^\top \right) u \tag{64}$$

$$\overset{(b)}{=} \max_{u: \|u\|_2 = 1, \ \tilde{R}^\top u = 0} u^\top \Sigma_{\boldsymbol{x}}^{1/2} S \Sigma_{\boldsymbol{x}}^{1/2} u \tag{65}$$

$$\overset{(c)}{\geq} \min_{\mathcal{S}: \dim(\mathcal{S}) = d - r} \max_{u: \|u\|_2 = 1, \ u \in \mathcal{S}} u^\top \Sigma_{\boldsymbol{x}}^{1/2} S \Sigma_{\boldsymbol{x}}^{1/2} u \tag{66}$$

$$\overset{(d)}{=} \lambda_{r+1} \left( \Sigma_{\boldsymbol{x}}^{1/2} S \Sigma_{\boldsymbol{x}}^{1/2} \right), \tag{67}$$

where $(a)$ follows again from the *Rayleigh quotient theorem* [37, Theorem 4.2.2], $(b)$ follows since $I_d - \tilde{R}(\tilde{R}^\top \tilde{R})^{-1} \tilde{R}^\top$ is an orthogonal projection matrix, and so we may write $u = u_\perp + u_\parallel$ so that $\|u_\perp\|^2 + \|u_\parallel\|^2 = 1$ and $\tilde{R}^\top u_\perp = 0$; Hence replacing $u$ with $u_\perp$ only increases the value of the maximum, $(c)$ follows by setting $\mathcal{S}$ to be a $d - r$ dimensional subspace of $\mathbb{R}^d$, and $(d)$ follows by the *Courant–Fischer variational characterization* [37, Theorem 4.2.6] (see Appendix D). The lower bound in $(c)$ can be achieved by setting the $r$ columns of $\tilde{R} \in \mathbb{R}^{d \times r}$ to be the top eigenvectors $\{v_i(\Sigma_{\boldsymbol{x}}^{1/2} S \Sigma_{\boldsymbol{x}}^{1/2})\}_{i \in [r]}$. This leads to the minimax representation $\tilde{R}^*$. From (63), the worst case $\tilde{f}$ is the top eigenvector of

$$\left( I_d - \tilde{R}^*((\tilde{R})^{*\top} \tilde{R}^*)^{-1} (\tilde{R})^{*\top} \right) \Sigma_{\boldsymbol{x}}^{1/2} S \Sigma_{\boldsymbol{x}}^{1/2} \left( I_d - \tilde{R}^*((\tilde{R})^{*\top} \tilde{R}^*)^{-1} (\tilde{R})^{*\top} \right). \tag{68}$$

This is a symmetric matrix, whose top eigenvector is the $(r + 1)$th eigenvector $v_{r+1}(\Sigma_{\boldsymbol{x}}^{1/2} S \Sigma_{\boldsymbol{x}}^{1/2})$. $\qquad \square$

We next prove Theorem 3.

*Proof of Theorem 3.* We follow the proof strategy mentioned after the statement of the theorem. We assume that $\boldsymbol{n} \equiv 0$ with probability 1, since, as for the pure minimax regret, this unavoidable additive term of $\mathbb{E}\left[\mathbb{E}[\boldsymbol{n}^2 \mid \boldsymbol{x}]\right]$ to the loss does not affect the regret.

*The minimax problem – a direct computation:* As in the derivations leading to (63), the minimax regret in (2) is given by

$$\mathrm{regret}_{\mathrm{mix}}(\mathcal{F}_S \mid \Sigma_{\boldsymbol{x}})$$
$$= \min_{\mathsf{L}(\boldsymbol{R}) \in \mathcal{P}(\mathcal{R})} \max_{f \in \mathcal{F}_S} \mathbb{E}\left[\mathrm{regret}(\boldsymbol{R}, f \mid \Sigma_{\boldsymbol{x}})\right] \tag{69}$$

$$= \min_{\mathsf{L}(\boldsymbol{R})\in\mathcal{P}(\mathcal{R})}\ \max_{\tilde{f}:\|\tilde{f}\|_2^2\le 1}\ \mathbb{E}\left[\tilde{f}^\top\left(S^{1/2}\Sigma_{\boldsymbol{x}}S^{1/2}-S^{1/2}\Sigma_{\boldsymbol{x}}\boldsymbol{R}(\boldsymbol{R}^\top\Sigma_{\boldsymbol{x}}\boldsymbol{R})^{-1}\boldsymbol{R}^\top\Sigma_{\boldsymbol{x}}S^{1/2}\right)\tilde{f}\right] \tag{70}$$

$$= \min_{\mathsf{L}(\Sigma_{\boldsymbol{x}}^{-1/2}\tilde{\boldsymbol{R}})\in\mathcal{P}(\mathcal{R})}\ \max_{\tilde{f}:\|\tilde{f}\|_2^2\le 1}\ \tilde{f}^\top S^{1/2}\Sigma_{\boldsymbol{x}}^{1/2}\mathbb{E}\left[I_d-\tilde{\boldsymbol{R}}(\tilde{\boldsymbol{R}}^\top\tilde{\boldsymbol{R}})^{-1}\tilde{\boldsymbol{R}}^\top\right]\Sigma_{\boldsymbol{x}}^{1/2}S^{1/2}\tilde{f} \tag{71}$$

$$= \min_{\mathsf{L}(\Sigma_{\boldsymbol{x}}^{-1/2}\tilde{\boldsymbol{R}})\in\mathcal{P}(\mathcal{R})}\ \lambda_1\left(S^{1/2}\Sigma_{\boldsymbol{x}}^{1/2}\mathbb{E}\left[I_d-\tilde{\boldsymbol{R}}(\tilde{\boldsymbol{R}}^\top\tilde{\boldsymbol{R}})^{-1}\tilde{\boldsymbol{R}}^\top\right]\Sigma_{\boldsymbol{x}}^{1/2}S^{1/2}\right) \tag{72}$$

where $\tilde{\boldsymbol{R}}=\Sigma_{\boldsymbol{x}}^{1/2}\boldsymbol{R}$. Determining the optimal distribution of the representation directly from this expression seems to be intractable. We thus next solve the maximin problem, and then return to the maximin problem (72), set a specific random representation, and show that it achieves the maximin value. This, in turn, establishes the optimality of this choice.

*The maximin problem:* Let an arbitrary $\mathsf{L}(\boldsymbol{f})$ be given. Then, taking the expectation of the regret over the random choice of $\boldsymbol{f}$, for any given $R\in\mathcal{R}$,

$$\mathbb{E}\left[\mathsf{regret}(R,\boldsymbol{f})\right]\overset{(a)}{=}\mathbb{E}\left[\mathrm{Tr}\left[\left(S^{1/2}\Sigma_{\boldsymbol{x}}S^{1/2}-S^{1/2}\Sigma_{\boldsymbol{x}}R(R^\top\Sigma_{\boldsymbol{x}}R)^{-1}R^\top\Sigma_{\boldsymbol{x}}S^{1/2}\right)\tilde{\boldsymbol{f}}\tilde{\boldsymbol{f}}^\top\right]\right] \tag{73}$$

$$\overset{(b)}{=}\mathrm{Tr}\left[\left(S^{1/2}\Sigma_{\boldsymbol{x}}S^{1/2}-S^{1/2}\Sigma_{\boldsymbol{x}}R(R^\top\Sigma_{\boldsymbol{x}}R)^{-1}R^\top\Sigma_{\boldsymbol{x}}S^{1/2}\right)\tilde{\Sigma}_{\boldsymbol{f}}\right] \tag{74}$$

$$=\mathrm{Tr}\left[\tilde{\Sigma}_{\boldsymbol{f}}^{1/2}\left(S^{1/2}\Sigma_{\boldsymbol{x}}S^{1/2}-S^{1/2}\Sigma_{\boldsymbol{x}}R(R^\top\Sigma_{\boldsymbol{x}}R)^{-1}R^\top\Sigma_{\boldsymbol{x}}S^{1/2}\right)\tilde{\Sigma}_{\boldsymbol{f}}^{1/2}\right] \tag{75}$$

$$\overset{(c)}{=}\mathrm{Tr}\left[\tilde{\Sigma}_{\boldsymbol{f}}^{1/2}S^{1/2}\Sigma_{\boldsymbol{x}}^{1/2}\left(I_d-\tilde{R}(\tilde{R}^\top\tilde{R})^{-1}\tilde{R}^\top\right)\Sigma_{\boldsymbol{x}}^{1/2}S^{1/2}\tilde{\Sigma}_{\boldsymbol{f}}^{1/2}\right] \tag{76}$$

$$=\mathrm{Tr}\left[\left(I-\tilde{R}(\tilde{R}^\top\tilde{R})^{-1}\tilde{R}^\top\right)\Sigma_{\boldsymbol{x}}^{1/2}S^{1/2}\tilde{\Sigma}_{\boldsymbol{f}}S^{1/2}\Sigma_{\boldsymbol{x}}^{1/2}\right] \tag{77}$$

$$\overset{(d)}{=}\mathrm{Tr}\left[\left(I-\tilde{R}(\tilde{R}^\top\tilde{R})^{-1}\tilde{R}^\top\right)\Sigma_{\boldsymbol{x}}^{1/2}S^{1/2}\tilde{\Sigma}_{\boldsymbol{f}}S^{1/2}\Sigma_{\boldsymbol{x}}^{1/2}\left(I-\tilde{R}(\tilde{R}^\top\tilde{R})^{-1}\tilde{R}^\top\right)\right] \tag{78}$$

$$\overset{(e)}{\ge}\min_{W\in\mathbb{R}^{d\times(d-r)}:W^\top W=I_{d-r}}\mathrm{Tr}\left[W^\top\Sigma_{\boldsymbol{x}}^{1/2}S^{1/2}\tilde{\Sigma}_{\boldsymbol{f}}S^{1/2}\Sigma_{\boldsymbol{x}}^{1/2}W\right] \tag{79}$$

$$\overset{(f)}{=}\sum_{i=r+1}^{d}\lambda_i(\Sigma_{\boldsymbol{x}}^{1/2}S^{1/2}\tilde{\Sigma}_{\boldsymbol{f}}S^{1/2}\Sigma_{\boldsymbol{x}}^{1/2}) \tag{80}$$

$$=\sum_{i=r+1}^{d}\lambda_i\left(\tilde{\Sigma}_{\boldsymbol{f}}S^{1/2}\Sigma_{\boldsymbol{x}}S^{1/2}\right), \tag{81}$$

where $(a)$ follows from Lemma 15 and setting $\tilde{\boldsymbol{f}}:=S^{-1/2}\boldsymbol{f}$, $(b)$ follows by setting $\tilde{\Sigma}_{\boldsymbol{f}}\equiv\Sigma_{\tilde{\boldsymbol{f}}}=\mathbb{E}[\tilde{\boldsymbol{f}}\tilde{\boldsymbol{f}}^\top]$, $(c)$ follows by setting $\tilde{R}:=\Sigma_{\boldsymbol{x}}^{1/2}R$, $(d)$ follows since $I-\tilde{R}(\tilde{R}^\top\tilde{R})^{-1}\tilde{R}^\top$ is an orthogonal projection (idempotent and symmetric matrix) of rank $d-r$, $(e)$ follows since any orthogonal projection can be written as $WW^\top$ where $W\in\mathbb{R}^{d\times(d-r)}$ is an orthogonal matrix $W^\top W=I_{d-r}$, $(f)$ follows from *Fan's variational characterization* [108] [37, Corollary 4.3.39.] (see Appendix D). Equality in $(e)$ can be achieved by letting $\tilde{R}$ be the top $r$ eigenvectors of $\Sigma_{\boldsymbol{x}}^{1/2}S^{1/2}\tilde{\Sigma}_{\boldsymbol{f}}S^{1/2}\Sigma_{\boldsymbol{x}}^{1/2}$.

The next step of the derivation is to maximize the expected regret over the probability law of $\boldsymbol{f}$ (or $\tilde{\boldsymbol{f}}$). Evidently, $\mathbb{E}[\mathsf{regret}(R,\boldsymbol{f})]=\sum_{i=r+1}^{d}\lambda_i(\tilde{\Sigma}_{\boldsymbol{f}}S^{1/2}\Sigma_{\boldsymbol{x}}S^{1/2})$ only depends on the random function $\tilde{\boldsymbol{f}}$ via $\tilde{\Sigma}_{\boldsymbol{f}}$. The covariance matrix $\tilde{\Sigma}_{\boldsymbol{f}}$ is constrained as follows. Recall that $\boldsymbol{f}$ is supported on $\mathcal{F}_S:=\{f\in\mathbb{R}^d:\|f\|_S^2\le 1\}$ (see (4)), and let $\Sigma_{\boldsymbol{f}}=\mathbb{E}[\boldsymbol{f}\boldsymbol{f}^\top]$ be its covariance matrix. Then, it must hold that $\mathrm{Tr}[S^{-1}\Sigma_{\boldsymbol{f}}]\le 1$. Then, it also holds that

$$1\ge\mathrm{Tr}[S^{-1}\Sigma_{\boldsymbol{f}}]=\mathrm{Tr}\left[\mathbb{E}[S^{-1}\boldsymbol{f}\boldsymbol{f}^\top]\right] \tag{82}$$

$$=\mathbb{E}\left[\boldsymbol{f}^\top S^{-1}\boldsymbol{f}\right] \tag{83}$$

$$=\mathbb{E}\left[\tilde{\boldsymbol{f}}^\top\tilde{\boldsymbol{f}}\right] \tag{84}$$

$$=\mathrm{Tr}\left[\tilde{\Sigma}_{\boldsymbol{f}}\right] \tag{85}$$

where $\tilde{\Sigma}_{\boldsymbol{f}} = S^{-1/2}\Sigma_{\boldsymbol{f}}S^{-1/2}$. Conversely, given any covariance matrix $\tilde{\Sigma}_{\boldsymbol{f}} \in \mathbb{S}_{++}^d$ such that $\mathrm{Tr}[\tilde{\Sigma}_{\boldsymbol{f}}] \leq 1$ there exists a random vector $\boldsymbol{f}$ supported on $\mathcal{F}_S$ such that

$$\mathbb{E}[\boldsymbol{f}\boldsymbol{f}^\top] = S^{1/2}\tilde{\Sigma}_{\boldsymbol{f}}S^{-1/2}. \tag{86}$$

We show this by an explicit construction. Let $\tilde{\Sigma}_{\boldsymbol{f}} = \tilde{V}_{\boldsymbol{f}}\tilde{\Lambda}_{\boldsymbol{f}}\tilde{V}_{\boldsymbol{f}}^\top$ be the eigenvalue decomposition of $\tilde{\Sigma}_{\boldsymbol{f}}$, and, for brevity, denote by $\tilde{\lambda}_i \equiv \lambda_i(\tilde{\Sigma}_{\boldsymbol{f}})$ the diagonal elements of $\tilde{\Lambda}_{\boldsymbol{f}}$. Let $\{\boldsymbol{q}_i\}_{i\in[d]}$ be a set of independent and identically (IID) distributed random variables, so that $\boldsymbol{q}_i$ is Rademacher, that is $\mathbb{P}[\boldsymbol{q}_i = 1] = \mathbb{P}[\boldsymbol{q}_i = -1] = 1/2$. Define the random vector

$$\boldsymbol{g} := \left(\boldsymbol{q}_1 \cdot \sqrt{\tilde{\lambda}_1}, \cdots, \boldsymbol{q}_d \cdot \sqrt{\tilde{\lambda}_d}\right)^\top. \tag{87}$$

The constraint $\mathrm{Tr}[\tilde{\Sigma}_{\boldsymbol{f}}] \leq 1$ implies that $\sum \tilde{\lambda}_i \leq 1$ and so $\|\boldsymbol{g}\|^2 = \sum_{i=1}^d \tilde{\lambda}_i \leq 1$ with probability 1. Then, letting $\tilde{\boldsymbol{f}} = \tilde{V}_{\boldsymbol{f}}\boldsymbol{g}$ it also holds that $\|\tilde{\boldsymbol{f}}\|_2^2 = \|\boldsymbol{g}\|_2^2 \leq 1$ with probability 1, and furthermore,

$$\mathbb{E}\left[\tilde{\boldsymbol{f}}\tilde{\boldsymbol{f}}^\top\right] = \tilde{V}_{\boldsymbol{f}}\mathbb{E}\left[\boldsymbol{g}\boldsymbol{g}^\top\right]\tilde{V}_{\boldsymbol{f}}^\top = \tilde{V}_{\boldsymbol{f}}\tilde{\Lambda}_{\boldsymbol{f}}\tilde{V}_{\boldsymbol{f}}^\top = \tilde{\Sigma}_{\boldsymbol{f}}. \tag{88}$$

Consequently, letting $\boldsymbol{f} = S^{1/2}\boldsymbol{f}$ assures that $\|\boldsymbol{f}\|_S = \|\tilde{\boldsymbol{f}}\|_2 \leq 1$ and $\mathbb{E}[\boldsymbol{f}\boldsymbol{f}^\top] = S^{1/2}\tilde{\Sigma}_{\boldsymbol{f}}S^{-1/2}$, as was required to obtain. Therefore, instead of maximizing over probability laws on $\mathcal{P}(\mathcal{F}_S)$, we may equivalently maximize over $\tilde{\Sigma}_{\boldsymbol{f}} \in \mathbb{S}_{++}^d$ such that $\mathrm{Tr}[\tilde{\Sigma}_{\boldsymbol{f}}] \leq 1$, i.e., to solve

$$\mathrm{regret}_{\mathrm{mix}}(\mathcal{F}_S \mid \Sigma_{\boldsymbol{x}}) = \max_{\tilde{\Sigma}_{\boldsymbol{f}}:\mathrm{Tr}[\tilde{\Sigma}_{\boldsymbol{f}}]\leq 1} \sum_{i=r+1}^d \lambda_i(\tilde{\Sigma}_{\boldsymbol{f}}S^{1/2}\Sigma_{\boldsymbol{x}}S^{1/2}). \tag{89}$$

The optimization problem in (89) is solved in Lemma 16, and is provided after this proof. Setting $\Sigma = S^{1/2}\Sigma_{\boldsymbol{x}}S^{1/2}$ in Lemma 16, and letting $\lambda_i \equiv \lambda_i(S^{1/2}\Sigma_{\boldsymbol{x}}S^{1/2})$, the solution is given by

$$\frac{\ell^* - r}{\sum_{i=1}^{\ell^*} \frac{1}{\lambda_i}} \tag{90}$$

where $\ell^* \in [d]\setminus[r]$ satisfies

$$\frac{\ell^* - r}{\lambda_{\ell^*}} \leq \sum_{i=1}^{\ell^*} \frac{1}{\lambda_i} \leq \frac{\ell^* - r}{\lambda_{\ell^*+1}}. \tag{91}$$

Lemma 16 also directly implies that an optimal $\tilde{\Sigma}_{\boldsymbol{f}}$ is given as in (10). The value in (90) is exactly $\mathrm{regret}_{\mathrm{mix}}(\mathcal{F}_S \mid \Sigma_{\boldsymbol{x}})$ claimed by the theorem, and we next show it is indeed achievable by a properly constructed random representation.

*The minimax problem – a solution via the maximin certificate:* Given the value of the regret game in mixed strategies found in (90), we may also find a minimax representation in mixed strategies. To this end, we return to the minimax expression in (72), and propose a random representation which achieves the maximin value in (90). Note that for any given $\tilde{R}$, the matrix $I_d - \tilde{R}(\tilde{R}^\top\tilde{R})^{-1}\tilde{R}^\top$ is an orthogonal projection, that is, a symmetric matrix whose eigenvalues are all either 0 or 1, and it has at most $r$ eigenvalues equal to zero. We denote its eigenvalue decomposition by $I_d - \tilde{R}(\tilde{R}^\top\tilde{R})^{-1}\tilde{R}^\top = U\Omega U^\top$. Then, any probability law on $\tilde{\boldsymbol{R}}$ induces a probability law on $U$ and $\Omega$ (and vice-versa). To find the mixed minimax representation, we propose setting $U = V(\Sigma_{\boldsymbol{x}}^{1/2}S\Sigma_{\boldsymbol{x}}^{1/2}) \equiv V$ with probability 1, that is, to be deterministic, and thus only randomize $\Omega$. With this choice, and by denoting, for brevity, $\Lambda \equiv \Lambda(\Sigma_{\boldsymbol{x}}^{1/2}S\Sigma_{\boldsymbol{x}}^{1/2}) = \Lambda(S^{1/2}\Sigma_{\boldsymbol{x}}S^{1/2})$, the value of the objective function in (72) is given by

$$\lambda_1\left(S^{1/2}\Sigma_{\boldsymbol{x}}^{1/2}V \cdot \mathbb{E}[\Omega] \cdot V^\top\Sigma_{\boldsymbol{x}}^{1/2}S^{1/2}\right)$$

$$= \lambda_1\left(\mathbb{E}[\Omega] \cdot V^\top\Sigma_{\boldsymbol{x}}^{1/2}S\Sigma_{\boldsymbol{x}}^{1/2}V\right) \tag{92}$$

$$= \lambda_1\left(\mathbb{E}[\Omega] \cdot \Lambda\right). \tag{93}$$

Now, the distribution of $\boldsymbol{\Omega}$ is equivalent to a distribution on its diagonal, which is supported on the finite set $\mathcal{A} := \{a \in \{0,1\}^d : \|a\|_1 \geq d - r\}$. Our goal is thus to find a probability law on $\boldsymbol{a}$, supported on $\mathcal{A}$, which solves

$$\min_{\mathsf{L}(\boldsymbol{\Omega})} \max_{i \in [d]} \lambda_1 \left(\mathbb{E}[\boldsymbol{\Omega}] \cdot \Lambda\right) = \min_{\mathsf{L}(\boldsymbol{a})} \max_{i \in [d]} \mathbb{E}[\boldsymbol{a}_i]\lambda_i \tag{94}$$

where $\lambda_i \equiv \lambda_i(S^{1/2}\Sigma_{\boldsymbol{x}}S^{1/2})$ are the diagonal elements of $\Lambda$. Consider $\ell^*$, the optimal dimension of the maximin problem, which satisfies (91). We then set $\boldsymbol{a}_{\ell^*+1} = \cdots = \boldsymbol{a}_d = 1$ to hold with probability 1, and so it remains to determine the probability law of $\overline{\boldsymbol{a}} := (\boldsymbol{a}_1, \ldots, \boldsymbol{a}_{\ell^*})$, supported on $\tilde{\mathcal{A}} := \{a \in \{0,1\}^{\ell^*} : \|a\|_1 \geq \ell^* - r\}$. Clearly, reducing $\|a\|_1$ only reduces the objective function $\max_{i \in [d]} \mathbb{E}[\boldsymbol{a}_i\lambda_i]$, and so we may in fact assume that $\overline{\boldsymbol{a}}$ is supported on $\overline{\mathcal{A}} := \{\overline{a} \in \{0,1\}^{\ell^*} : \|\overline{a}\|_1 = \ell^* - r\}$, a finite subset of cardinality $\binom{\ell^*}{r}$. Suppose that we find a probability law $\mathsf{L}(\overline{\boldsymbol{a}})$ supported on $\overline{\mathcal{A}}$ such that

$$\mathbb{E}[\boldsymbol{a}_i] = (\ell^* - r) \cdot \frac{1/\lambda_i}{\sum_{i=1}^{\ell^*} 1/\lambda_i} := b_i, \tag{95}$$

for all $i \in [\ell^*]$. Then, since $\mathbb{E}[\boldsymbol{a}_i] = 1$ for $i \in [d]\backslash[\ell^*]$

$$\max_{i \in [d]} \mathbb{E}[\boldsymbol{a}_i]\lambda_i = \max\left\{ \frac{\ell^* - r}{\sum_{i=1}^{\ell^*} \frac{1}{\lambda_i}}, \lambda_{\ell^*+1}, \cdots, \lambda_d \right\} \tag{96}$$

$$= \max\left\{ \frac{\ell^* - r}{\sum_{i=1}^{\ell^*} \frac{1}{\lambda_i}}, \lambda_{\ell^*+1} \right\} \tag{97}$$

$$\stackrel{(*)}{=} \frac{\ell^* - r}{\sum_{i=1}^{\ell^*} \frac{1}{\lambda_i}}, \tag{98}$$

where $(*)$ follows from the condition on $\ell^*$ in the right inequality of (91). This proves that such probability law achieves the minimax regret in mixed strategies. This last term is $\mathsf{regret}_{\mathrm{mix}}(\mathcal{F}_S \mid \Sigma_{\boldsymbol{x}})$ claimed by the theorem. It remains to construct $\mathsf{L}(\overline{\boldsymbol{a}})$ which satisfies (95). To this end, note that the set

$$\mathcal{C} := \left\{ c \in [0,1]^{\ell^*} : \|c\|_1 = \ell^* - r \right\} \tag{99}$$

is convex and compact, and $\overline{\mathcal{A}}$ is the set of its *extreme points* ($\mathcal{C}$ is the convex hull of $\overline{\mathcal{A}}$). Letting $\overline{b} = (b_1, \ldots, b_{\ell^*})^\top$ as denoted in (95), it holds that $\overline{b}_i \geq 0$ and $\{\overline{b}_i\}_{i=1}^{\ell^*}$ is a non-decreasing sequence. Using the condition on $\ell^*$ in the left inequality of (91), it then holds that

$$\overline{b}_1 \leq \cdots \leq \overline{b}_{\ell^*} = (\ell^* - r) \cdot \frac{1/\lambda_{\ell^*}}{\sum_{i=1}^{\ell^*} 1/\lambda_i} \leq 1. \tag{100}$$

Hence, $\overline{b} \in \mathcal{C}$. By Carathéodory's theorem [107, Prop. 1.3.1] (see Appendix D), any point inside a convex compact set in $\mathbb{R}^{\ell^*}$ can be written as a convex combination of at most $\ell^* + 1$ extreme points. Thus, there exists $\{p_{\overline{a}}\}_{\overline{a} \in \overline{\mathcal{A}}}$ such that $p_{\overline{a}} \in [0,1]$ and $\sum_{\overline{a} \in \overline{\mathcal{A}}} p_{\overline{a}} = 1$ so that $\overline{b} = \sum_{\overline{a} \in \overline{\mathcal{A}}} p_{\overline{a}} \cdot \overline{a}$, and moreover the support of $p_{\overline{a}}$ has cardinality at most $\ell^* + 1$. Let $\overline{A} \in \{0,1\}^{\ell^* \times |\overline{\mathcal{A}}|}$ be such that its $j$th column is given by the $j$th member of $\overline{\mathcal{A}}$ (in an arbitrary order). Let $p \in [0,1]^{|\overline{\mathcal{A}}|}$ be a vector whose $j$th element corresponds to the $j$th member of $\overline{\mathcal{A}}$. Then, $p$ is the solution to $\overline{A}p = \overline{b}$, and as claimed above, such a solution with at most $\ell^* + 1$ nonzero entries always exists. Setting $\boldsymbol{a} = (\overline{a}, \underbrace{1 \ldots, 1}_{d - \ell^* \text{ terms}})$ with probability $p_{\overline{a}}$ then assures that (95) holds, as was required to be proved.

Given the above, we observe that setting $\tilde{R}$ as in the theorem induces a distribution on $\boldsymbol{\Omega}$ for which the random entries of its diagonal $\boldsymbol{a}$ satisfy (95), and thus achieve $\mathsf{regret}_{\mathrm{mix}}(\mathcal{F}_S \mid \Sigma_{\boldsymbol{x}})$. $\qquad\square$

We next turn to complete the proof of Theorem 3 by solving the optimization problem in (89). Assume that $\Sigma \in \mathbb{S}_{++}^d$ is a strictly positive covariance matrix $\Sigma \succ 0$, and consider the optimization problem

$$v_r^* = \max_{\tilde{\Sigma}_{\boldsymbol{f}} \in \mathbb{S}_+^d} \sum_{i=r+1}^d \lambda_i(\tilde{\Sigma}_{\boldsymbol{f}}\Sigma)$$

$$\text{subject to } \text{Tr}[\tilde{\Sigma}_{\boldsymbol{f}}] \leq 1 \tag{101}$$

for some $r \in [d-1]$. Note that the objective function refers to the maximization of the $d-r$ minimal eigenvalues of $\Sigma^{1/2}\tilde{\Sigma}_{\boldsymbol{f}}\Sigma^{1/2}$.

**Lemma 16.** *Let*

$$a_\ell := \frac{\ell - r}{\sum_{i=1}^{\ell} \frac{1}{\lambda_i(\Sigma)}}. \tag{102}$$

*The optimal value of (101) is* $v^* = \max_{[d]\setminus[r]} a_\ell$ *and* $\ell^* \in \arg\max_{[d]\setminus[r]} a_\ell$ *iff*

$$\frac{\ell^* - r}{\lambda_{\ell^*}(\Sigma)} \leq \sum_{i=1}^{\ell^*} \frac{1}{\lambda_i(\Sigma)} \leq \frac{\ell^* - r}{\lambda_{\ell^*+1}(\Sigma)}. \tag{103}$$

*An optimal solution is*

$$\tilde{\Sigma}_{\boldsymbol{f}}^* = \left[\sum_{i=1}^{\ell^*} \frac{1}{\lambda_i(\Sigma)}\right]^{-1} \cdot V(\Sigma) \operatorname{diag}\left(\frac{1}{\lambda_1(\Sigma)}, \cdots, \frac{1}{\lambda_{\ell^*}(\Sigma)}, 0, \cdots, 0\right) \cdot V(\Sigma)^\top. \tag{104}$$

*Proof.* Let $\overline{\Sigma}_{\boldsymbol{f}} = \Sigma^{1/2}\tilde{\Sigma}_{\boldsymbol{f}}\Sigma^{1/2}$, let $\overline{\Sigma}_{\boldsymbol{f}} = \overline{U}_{\boldsymbol{f}}\overline{\Lambda}_{\boldsymbol{f}}\overline{U}_{\boldsymbol{f}}^\top$ be its eigenvalue decomposition, and, for brevity, denote $\overline{\lambda}_i \equiv \lambda_i(\overline{\Sigma}_{\boldsymbol{f}})$. Then, the trace operation appearing in the constraint of (101) can be written as

$$\text{Tr}[\tilde{\Sigma}_{\boldsymbol{f}}] = \text{Tr}\left[\Sigma^{-1/2}\overline{\Sigma}_{\boldsymbol{f}}\Sigma^{-1/2}\right] \tag{105}$$

$$= \text{Tr}\left[\Sigma^{-1/2}\overline{U}_{\boldsymbol{f}}\overline{\Lambda}_{\boldsymbol{f}}\overline{U}_{\boldsymbol{f}}^\top\Sigma^{-1/2}\right] \tag{106}$$

$$= \text{Tr}\left[\Sigma^{-1/2}\left(\sum_{i=1}^{d}\lambda_i\overline{u}_i\overline{u}_i^\top\right)\Sigma^{-1/2}\right] \tag{107}$$

$$= \sum_{i=1}^{d}\overline{\lambda}_i \cdot \left(\overline{u}_i^\top\Sigma^{-1}\overline{u}_i\right) \tag{108}$$

$$= \sum_{i=1}^{d}c_i\overline{\lambda}_i, \tag{109}$$

where $\overline{u}_i = v_i(\overline{U}_{\boldsymbol{f}})$ (that is, the $i$th column of $\overline{U}_{\boldsymbol{f}}$), and $c_i := \overline{u}_i^\top\Sigma^{-1}\overline{u}_i$ (which satisfies $c_i > 0$). Thus, the optimization problem in (101) over $\tilde{\Sigma}_{\boldsymbol{f}}$ is equivalent to an optimization problem over $\{\overline{\lambda}_i, \overline{u}_i\}_{i\in[d]}$, given by

$$v_r^* = \max_{\{\overline{u}_i, \overline{\lambda}_i\}_{i\in[d]}} \sum_{i=r+1}^{d} \overline{\lambda}_i$$

$$\text{subject to } \sum_{i=1}^{d} c_i\overline{\lambda}_i \leq 1,$$

$$c_i = \overline{u}_i^\top\Sigma^{-1}\overline{u}_i,$$

$$\overline{u}_i^\top\overline{u}_j = \delta_{ij},$$

$$\overline{\lambda}_1 \geq \overline{\lambda}_2 \geq \cdots \geq \overline{\lambda}_d \geq 0. \tag{110}$$

To solve the optimization problem (110), let us fix feasible $\{\overline{u}_i\}_{i\in[d]}$, so that $\{c_i\}_{i\in[d]}$ are fixed too. This results the problem

$$v_r^*(\{\overline{u}_i\}) \equiv v_r^*(\{c_i\}) = \max_{\{\overline{\lambda}_i\}_{i\in[d]}} \sum_{i=r+1}^{d} \overline{\lambda}_i$$

$$\text{subject to } \sum_{i=1}^{d} c_i\overline{\lambda}_i \leq 1,$$

$$\overline{\lambda}_1 \geq \overline{\lambda}_2 \geq \cdots \geq \overline{\lambda}_d \geq 0. \tag{111}$$

The objective function of (111) is linear in $\{\overline{\lambda}_i\}_{i \in [d]}$ and its constraint set is a convex bounded polytope. So the solution to (111) must be obtained on the boundary of the constraint set. Clearly, the optimal value satisfies $v_r^*(\{c_i\}) \geq 0$, and thus the solution $\{\overline{\lambda}_i^*\}_{i \in [d]}$ must be obtained when the constraint $\sum_{i=1}^{d} c_i \overline{\lambda}_i \leq 1$ is satisfied with equality. Indeed, if this is not the case then one may scale all $\overline{\lambda}_i^*$ by a constant larger than 1, and obtain larger value of the objective, while still satisfying the constraint.

To find the optimal solution to (111), we consider feasible points for which $\ell := \max\{i \in [d] : \overline{\lambda}_i > 0\}$ is fixed. Let $\{\overline{\lambda}_i^*\}_{i \in [d]}$ be the optimal solution of (111), under the additional constraint that $\overline{\lambda}_{\ell+1} = \cdots = \overline{\lambda}_d = 0$. We next prove that $\overline{\lambda}_1^* = \cdots = \overline{\lambda}_\ell^*$ must hold. To this end, assume by contradiction that there exists $j \in [\ell]$ so that $\overline{\lambda}_{j-1}^* > \overline{\lambda}_j^* > 0$. There are two cases to consider, to wit, whether $j - 1 < r + 1$ and so only $\overline{\lambda}_j$ appears in the objective of (111), or, otherwise, $j - 1 \geq r + 1$ and then $\overline{\lambda}_{j-1} + \overline{\lambda}_j$ appears in the objective of (111). Assuming the first case, let $\alpha = \overline{\lambda}_{j-1}^* c_{j-1} + \overline{\lambda}_j^* c_j$ and consider the optimization problem

$$\max_{\hat{\lambda}_{j-1}, \hat{\lambda}_j} \hat{\lambda}_j$$
$$\text{subject to } \hat{\lambda}_{j-1} c_{j-1} + \hat{\lambda}_j c_j = \alpha,$$
$$\hat{\lambda}_{j-1} \geq \hat{\lambda}_j > 0. \tag{112}$$

It is easy to verify that the optimum of this problem is $\hat{\lambda}_{j-1}^* = \hat{\lambda}_j^* = \frac{\alpha}{c_{j-1}+c_j}$. Thus, if $\overline{\lambda}_{j-1}^* > \overline{\lambda}_j^*$ then one can replace this pair with $\overline{\lambda}_{j-1}^* = \overline{\lambda}_j^* = \hat{\lambda}_{j-1}^* = \hat{\lambda}_j^*$ so that the value of the constraint $\sum_{i=1}^{d} \overline{\lambda}_i c_i$ remains the same, and thus $(\overline{\lambda}_1^*, \cdots, \hat{\lambda}_{j-1}^*, \hat{\lambda}_j^*, \overline{\lambda}_{j+1}^*, \ldots \overline{\lambda}_d^*)$ is a feasible point, while the objective function value of (111) is smaller; a contradiction. Therefore, it must hold for the first case that $\overline{\lambda}_{j-1}^* = \overline{\lambda}_j^*$. For the second case, in a similar fashion, let now $\alpha = \overline{\lambda}_{j-1}^* c_{j-1} + \overline{\lambda}_j^* c_j$, and consider the optimization problem

$$\max_{\hat{\lambda}_{j-1}, \hat{\lambda}_j} \hat{\lambda}_j + \hat{\lambda}_{j-1}$$
$$\text{subject to } \hat{\lambda}_{j-1} c_{j-1} + \hat{\lambda}_j c_j = \alpha,$$
$$\hat{\lambda}_{j-1} \geq \hat{\lambda}_j > 0. \tag{113}$$

The solution for this optimization problem is at one of the two extreme points of the feasible interval for $\hat{\lambda}_j$. Since $\lambda_j^* > 0$ was assumed it therefore must hold that $\hat{\lambda}_{j-1}^* = \hat{\lambda}_j^*$, and hence also $\overline{\lambda}_{j-1}^* = \overline{\lambda}_j^*$. Thus, $\lambda_{j-1}^* < \lambda_j^*$ leads to a contradiction. From the above, we deduce that the optimal solution of (111) under the additional constraint that $\overline{\lambda}_{\ell+1} = \cdots = \overline{\lambda}_d = 0$ is

$$\overline{\lambda}_1^* = \cdots = \overline{\lambda}_\ell^* = \frac{1}{\sum_{i=1}^{\ell} c_i} \tag{114}$$

$$\overline{\lambda}_{\ell+1}^* = \cdots = \overline{\lambda}_d^* = 0, \tag{115}$$

and that the optimal value is $\frac{\ell-r}{\sum_{i=1}^{\ell} c_i}$. Since $\ell \in [d] \backslash [r]$ can be arbitrarily chosen, we deduce that the value of (111) is

$$v^*(\{c_i\}) = \max_{\ell \in [d] \backslash [r]} \frac{\ell - r}{\sum_{i=1}^{\ell} c_i}. \tag{116}$$

For any given $\ell \in [d] \backslash [r]$, we may now optimize over $\{\overline{u}_i\}$, which from (116) is equivalent to minimizing $\sum_{i=1}^{\ell} c_i$. It holds that

$$\min_{\{\overline{u}_i\}} \sum_{i=1}^{\ell} c_i = \min_{\{\overline{u}_i : \overline{u}_i^\top \overline{u}_j = \delta_{ij}\}} \sum_{i=1}^{\ell} \overline{u}_i^\top \Sigma^{-1} \overline{u}_i \tag{117}$$

$$= \min_{\{\overline{u}_i : \overline{u}_i^\top \overline{u}_j = \delta_{ij}\}} \text{Tr}\left[\Sigma^{-1} \sum_{i=1}^{\ell} \overline{u}_i \overline{u}_i^\top\right] \tag{118}$$

$$\overset{(a)}{=} \min_{\dot{U} \in \mathbb{R}^{d \times \ell} : \dot{U}^\top \dot{U} = I_\ell} \text{Tr} \left[ \Sigma^{-1} \dot{U} \dot{U}^\top \right] \tag{119}$$

$$= \min_{\dot{U} \in \mathbb{R}^{d \times \ell} : \dot{U}^\top \dot{U} = I_\ell} \text{Tr} \left[ \dot{U}^\top \Sigma^{-1} \dot{U} \right] \tag{120}$$

$$\overset{(b)}{=} \sum_{i=1}^{\ell} \frac{1}{\lambda_i(\Sigma)}, \tag{121}$$

where in $(a)$ $\dot{U} \in \mathbb{R}^{d \times \ell}$ whose $\ell$ columns are $\{\overline{u}_i\}_{i \in [\ell]}$ and $\dot{U}^\top \dot{U} = I_\ell$, and in $(b)$ we have used *Fan's variational characterization* [108] [37, Corollary 4.3.39.] (see Appendix D). Substituting back to (116) results that

$$v_r^* = \max_{\ell \in [d] \setminus [r]} \frac{\ell - r}{\sum_{i=1}^{\ell} \frac{1}{\lambda_i(\Sigma)}} = \max_{\ell \in [d] \setminus [r]} a_\ell. \tag{122}$$

Let us denote that maximizer index by $\ell^*$. Then, Fan's characterization is achieved by setting $\overline{U}_{\boldsymbol{f}} = V$ (so that the $\ell^*$ columns of $\dot{U}$ are the $\ell^*$ eigenvectors $v_i(\Sigma)$, corresponding to the $\ell^*$ largest eigenvalues of $\Sigma$), so that

$$\overline{\Sigma}_{\boldsymbol{f}}^* = \left[ \sum_{i=1}^{\ell^*} \frac{1}{\lambda_i(\Sigma)} \right]^{-1} \cdot V \cdot \text{diag} \left( \underbrace{1, \ldots, 1}_{\ell^* \text{ terms}}, 0, \cdots, 0 \right) \cdot V^\top, \tag{123}$$

and then

$$\tilde{\Sigma}_{\boldsymbol{f}}^* = \Sigma^{-1/2} \overline{\Sigma}_{\boldsymbol{f}}^* \Sigma^{-1/2} \tag{124}$$

$$= \left[ \sum_{i=1}^{\ell^*} \frac{1}{\lambda_i(\Sigma)} \right]^{-1} \cdot V \Lambda^{-1/2} V^\top V \cdot \text{diag} \left( 1, \ldots, 1, 0, \cdots, 0 \right) V^\top V \Lambda^{-1/2} V^\top \tag{125}$$

$$= \left[ \sum_{i=1}^{\ell^*} \frac{1}{\lambda_i(\Sigma)} \right]^{-1} \cdot V \cdot \text{diag} \left( \frac{1}{\lambda_1(\Sigma)}, \ldots, \frac{1}{\lambda_{\ell^*}(\Sigma)}, 0, \cdots, 0 \right) \cdot V^\top \tag{126}$$

as claimed in (104).

To complete the proof, it remains to characterize $\ell^*$, which belongs to the set possible indices maximizing $\{a_\ell\}_{\ell \in [d] \setminus [r]}$. Since $\ell^*$ maximizes $a_\ell$ it must be a local maximizer, that is, it must hold that $a_{\ell^*-1} \leq a_{\ell^*} \geq a_{\ell^*+1}$. By simple algebra, these conditions are equivalent to those in (103). It remains to show that any $\ell \in [d] \setminus [r]$ which satisfies (103) has the same value, and thus any local maxima is a global maxima. We will show this by proving that the sequence $\{a_\ell\}_{\ell=r}^{d}$ is *unimodal*, as follows. Let $\Delta_\ell := a_{\ell+1} - a_\ell$ be the discrete derivative of $\{a_\ell\}_{\ell \in [d]}$, and consider the sequence $\{\Delta_\ell\}_{\ell \in [d] \setminus [r]}$. We show that as $\ell$ increases from $r$ to $d$, $\{\Delta_\ell\}_{\ell \in [d] \setminus [r]}$ is only changing its sign at most once. To this end, we first note that

$$\Delta_\ell = \frac{\ell + 1 - r}{\sum_{i=1}^{\ell+1} \frac{1}{\lambda_i(\Sigma)}} - \frac{\ell - r}{\sum_{i=1}^{\ell} \frac{1}{\lambda_i(\Sigma)}} = \frac{\sum_{i=1}^{\ell} \frac{1}{\lambda_i(\Sigma)} - (\ell - r) \frac{1}{\lambda_{\ell+1}(\Sigma)}}{\left[ \sum_{i=1}^{\ell+1} \frac{1}{\lambda_i(\Sigma)} \right] \left[ \sum_{i=1}^{\ell} \frac{1}{\lambda_i(\Sigma)} \right]}. \tag{127}$$

Since the denominator of (127) is strictly positive, it suffices to prove that the sequence comprised of the numerator of (127), to wit $\{\zeta_\ell\}_{\ell \in [d] \setminus [r]}$ with

$$\zeta_\ell := \sum_{i=1}^{\ell} \frac{1}{\lambda_i(\Sigma)} - (\ell - r) \frac{1}{\lambda_{\ell+1}(\Sigma)}, \tag{128}$$

is only changing its sign at most once. Indeed, this claim is true because $\zeta_r = \sum_{i=1}^{\ell} \frac{1}{\lambda_i(\Sigma)} > 0$ and because $\{\zeta_\ell\}_{\ell \in [d] \setminus [r]}$ is a monotonic non-increasing sequence,

$$\zeta_\ell - \zeta_{\ell+1} = (\ell - r + 1) \left[ \frac{1}{\lambda_{\ell+2}(\Sigma)} - \frac{1}{\lambda_{\ell+1}(\Sigma)} \right] \geq 0. \tag{129}$$

Therefore, $\{\zeta_\ell\}_{\ell \in [d] \setminus [r]}$ has at most a single sign change (its has a positive value at $\ell = r$ and is monotonically non-increasing with $\ell$ up to $\ell = d$), and so is $\{\Delta_\ell\}_{\ell=r}^{d}$. The single sign change property of the finite difference $\{\Delta_\ell\}_{\ell=r}^{d}$ is equivalent to the fact that $\{a_\ell\}_{\ell=r}^{d}$ is *unimodal*. Thus, any local maximizer of $a_\ell$ is also a global maximizer. $\qquad \square$

## F  The Hilbert space MSE setting

In this section, we show that the regret expressions in Section 3 can be easily generalized to an infinite dimensional Hilbert space, for responses with noise that is statistically independent of the features. We still assume the MSE loss function ($\mathcal{Y} = \mathbb{R}$ , and $\mathsf{loss}(y_1, y_2) = (y_1 - y_2)^2$), and that the predictor is a linear function. However, we allow the the representation and response function to be functions in a Hilbert space. As will be evident, the resulting regret is not very different from the finite-dimensional case. Formally, this is defined as follows:

**Definition 17** (The Hilbert space MSE setting). Assume that $\boldsymbol{x} \sim P_{\boldsymbol{x}}$ is supported on a compact subset $\mathcal{X} \subset \mathbb{R}^d$, and let $L_2(P_{\boldsymbol{x}})$ be the Hilbert space of functions from $\mathcal{X} \to \mathbb{R}$ such that $\mathbb{E}[f^2(\boldsymbol{x})] = \int_{\mathcal{X}} f^2(\boldsymbol{x}) \cdot \mathrm{d}P_{\boldsymbol{x}} < \infty$, with the inner product,

$$\langle f, g \rangle := \int_{\mathcal{X}} f(\boldsymbol{x}) g(\boldsymbol{x}) \cdot \mathrm{d}P_{\boldsymbol{x}} \tag{130}$$

for $f, g \in L_2(P_{\boldsymbol{x}})$. Let $\{\phi_j(x)\}_{j=1}^{\infty}$ be an orthonormal basis for $L_2(P_{\boldsymbol{x}})$.

A representation is comprised of a set of functions $\{\psi_i\}_{i \in [r]} \subset L_2(P_{\boldsymbol{x}})$, $\psi_i \colon \mathcal{X} \to \mathbb{R}$, so that

$$\mathcal{R} := \{R(x) = (\psi_1(x), \dots, \psi_r(x))^{\top} \in \mathbb{R}^r\}. \tag{131}$$

Let $\{\lambda_j\}_{j \in \mathbb{N}}$ be a positive monotonic non-increasing sequence for which $\lambda_j \downarrow 0$ as $j \to \infty$, and let $\mathcal{F}$ be the set of functions from $\mathcal{X} \to \mathbb{R}$ such that given $f \in \mathcal{F}$, the response is given by

$$\boldsymbol{y} = f(\boldsymbol{x}) + \boldsymbol{n} \in \mathbb{R} \tag{132}$$

where

$$f \in \mathcal{F}_{\{\lambda_j\}} := \left\{ f(x) = \sum_{j=1}^{\infty} f_j \phi_j(x) \colon \{f_j\}_{j \in \mathbb{N}} \in \ell_2(\mathbb{N}), \quad \sum_{j=1}^{\infty} \frac{f_j^2}{\lambda_j} \leq 1 \right\}, \tag{133}$$

where $\boldsymbol{n} \in \mathbb{R}$ is a homoscedastic noise that is statistically independent of $\boldsymbol{x}$ and satisfies $\mathbb{E}[\boldsymbol{n}] = 0$. Infinite-dimensional ellipsoids such as $\mathcal{F}_{\{\lambda_j\}}$ naturally arise in reproducing kernel Hilbert spaces (RKHS) [59, Chapter 12] [60, Chapter 16], in which $\{\lambda_j\}$ is the eigenvalues of the kernel. In this case, the set $\mathcal{F}_{\{\lambda_i\}} = \{f \colon \|f\|_{\mathcal{H}} \leq 1\}$ where $\|\cdot\|_{\mathcal{H}}$ is the norm of the RKHS $\mathcal{H}$. For example, $\mathcal{H}$ could be the first-order Sobolev space of functions with finite first derivative energy.

Let the set of predictor functions be the set of linear functions from $\mathbb{R}^d \to \mathbb{R}$, that is

$$\mathcal{Q} := \{Q(z) = q^{\top}z = \sum_{i=1}^{r} q_i \cdot \psi_i(x), \ q \in \mathbb{R}^r\}. \tag{134}$$

We denote the pure (resp. mixed) minimax regret as $\mathsf{regret}_{\mathsf{pure}}(\mathcal{R}, \mathcal{F}_{\{\lambda_j\}} \mid P_{\boldsymbol{x}})$ (resp. $\mathsf{regret}_{\mathsf{mix}}(\mathcal{R}, \mathcal{F}_{\{\lambda_j\}} \mid P_{\boldsymbol{x}})$). We begin with pure strategies.

**Theorem 18.** *For the Hilbert space MSE setting (Definition 17)*

$$\mathsf{regret}_{\mathsf{pure}}(\mathcal{R}, \mathcal{F}_{\{\lambda_j\}} \mid P_{\boldsymbol{x}}) = \lambda_{r+1}. \tag{135}$$

*A minimax representation is*

$$R^*(x) = (\phi_1(x), \dots, \phi_r(x))^{\top}, \tag{136}$$

*and the worst case response function is* $f^* = \sqrt{\lambda_{r+1}} \cdot \phi_{r+1}$.

We now turn to the minimax representation in mixed strategies.

**Theorem 19.** *For the Hilbert space MSE setting (Definition 17)*

$$\mathsf{regret}_{\mathsf{mix}}(\mathcal{R}, \mathcal{F}_{\{\lambda_j\}} \mid P_{\boldsymbol{x}}) = \frac{\ell^* - r}{\sum_{i=1}^{\ell^*} \frac{1}{\lambda_i}}, \tag{137}$$

*where $\ell^*$ is defined as (9) of Theorem 3 (with the replacement $d \to \mathbb{N}_+$). Let $\{\boldsymbol{b}_j\}_{j=1}^{\infty}$ be an IID sequence of Rademacher random variables, $\mathbb{P}[\boldsymbol{b}_i = 1] = \mathbb{P}[\boldsymbol{b}_i = -1] = 1/2$. Then, a least favorable prior $\boldsymbol{f}^*$ is*

$$\boldsymbol{f}_i^* = \begin{cases} \boldsymbol{b}_i \cdot \frac{1}{\sqrt{\sum_{i=1}^{\ell^*} \frac{1}{\lambda_i}}}, & 1 \leq i \leq \ell_* \\ 0, & i \geq \ell_* + 1 \end{cases}, \tag{138}$$

and a law of minimax representation is to choose

$$\boldsymbol{R}^*(x) = \{\phi_{\mathcal{I}_j}(x)\}_{j=1}^r \tag{139}$$

with probability $p_j$ , $j \in [\binom{\ell^*}{r}]$, defined as in Theorem 3.

**Discussion** Despite having countably infinite possible number of representations, the optimal representation only utilizes a *finite* set of orthogonal functions, as determined by the radius of $\mathcal{F}_{\{s_i\}}$. The proof of Theorems 18 and 19 is obtained by reducing the infinite dimensional problem to a $d$-dimensional problem via an approximation argument, then showing the the finite dimensional case is similar to the problem of Section 3, and then taking limit $d \uparrow \infty$.

## F.1  Proofs

Let us denote the $d$-dimensional *slice* of $\mathcal{F}_{\{\lambda_j\}}$ by

$$\mathcal{F}^{(d)}_{\{\lambda_j\}} := \left\{ f(x) \in \mathcal{F}_{\{\lambda_j\}} : f_j = 0 \text{ for all } j \geq d+1 \right\}. \tag{140}$$

Further, let us consider the restricted representation class, in which the representation functions $\psi_i(t)$ belong to the span of the first $d$ basis functions, that is

$$\mathcal{R}^{(d)} := \{R(x) \in \mathcal{R} := \psi_i(x) \in \mathrm{span}(\{\phi_i\}_{i \in [d]}) \text{ for all } i \in [r]\}. \tag{141}$$

The following proposition implies that the regret in the infinite-dimensional Hilbert space is obtained as the limit of finite-dimensional regrets, as the one characterized in Section 3:

**Proposition 20.** *It holds that*

$$\mathsf{regret}_{\mathsf{pure}}(\mathcal{R}, \mathcal{F}_{\{\lambda_j\}} \mid P_{\boldsymbol{x}}) = \lim_{d \uparrow \infty} \mathsf{regret}_{\mathsf{pure}}(\mathcal{R}^{(d)}, \mathcal{F}^{(d)}_{\{\lambda_j\}} \mid P_{\boldsymbol{x}}) \tag{142}$$

*and*

$$\mathsf{regret}_{\mathsf{mix}}(\mathcal{R}, \mathcal{F}_{\{\lambda_j\}} \mid P_{\boldsymbol{x}}) = \lim_{d \uparrow \infty} \mathsf{regret}_{\mathsf{mix}}(\mathcal{R}^{(d)}, \mathcal{F}^{(d)}_{\{\lambda_j\}} \mid P_{\boldsymbol{x}}). \tag{143}$$

*Proof.* Let $\{c_{ij}\}_{j \in \mathbb{N}}$ be the coefficients of the orthogonal expansion of $\psi_i$, $i \in [r]$, that is, $\psi_i = \sum_{j=1}^{\infty} c_{ij}\phi_j$. With a slight abuse of notation, we also let $c_i := (c_{i1}, c_{i2} \ldots) \in \ell_2(\mathbb{N})$. We use a sandwich argument. On one hand,

$$\mathsf{regret}_{\mathsf{pure}}(\mathcal{R}, \mathcal{F}_{\{\lambda_j\}} \mid P_{\boldsymbol{x}}) = \min_{R \in \mathcal{R}} \max_{f \in \mathcal{F}_{\{\lambda_j\}}} \mathsf{regret}(R, f) \tag{144}$$

$$\geq \min_{R \in \mathcal{R}} \max_{f \in \mathcal{F}^{(d)}_{\{\lambda_j\}}} \mathsf{regret}(R, f) \tag{145}$$

$$\overset{(*)}{=} \min_{R \in \mathcal{R}^{(d)}} \max_{f \in \mathcal{F}^{(d)}_{\{\lambda_j\}}} \mathsf{regret}(R, f) \tag{146}$$

$$= \mathsf{regret}_{\mathsf{pure}}(\mathcal{R}^{(d)}, \mathcal{F}^{(d)}_{\{\lambda_j\}} \mid P_{\boldsymbol{x}}), \tag{147}$$

where $(*)$ follows from the following reasoning: For any $(R \in \mathcal{R}, f \in \mathcal{F}^{(d)}_{\{\lambda_j\}})$,

$$\mathsf{regret}(R, f) = \min_{q \in \mathbb{R}^r} \mathbb{E}\left[ \left( \sum_{j=1}^d f_j \phi_j(\boldsymbol{x}) + \boldsymbol{n} - \sum_{j=1}^{\infty} \sum_{i=1}^r q_i c_{ij} \phi_j(\boldsymbol{x}) \right)^2 \right] - \mathbb{E}\left[ \boldsymbol{n}^2 \right] \tag{148}$$

$$\overset{(a)}{=} \min_{q \in \mathbb{R}^r} \mathbb{E}\left[ \left( \sum_{j=1}^d f_j \phi_j(\boldsymbol{x}) - \sum_{j=1}^{\infty} \sum_{i=1}^r q_i c_{ij} \phi_j(\boldsymbol{x}) \right) \right] \tag{149}$$

$$\overset{(b)}{=} \min_{q \in \mathbb{R}^r} \sum_{j=1}^d \left( f_j - \sum_{i=1}^r q_i c_{ij} \right)^2 + \sum_{j=d+1}^{\infty} \left( \sum_{i=1}^r q_i c_{ij} \right)^2, \tag{150}$$

where here $(a)$ follows since the noise $\boldsymbol{n}$ is independent of $\boldsymbol{x}$, and since, similarly to the finite-dimensional case (Section 3), the prediction loss based on the features $x \in \mathcal{X}$ is $\mathbb{E}[\boldsymbol{n}^2]$, for any given $f \in \mathcal{F}$, $(b)$ follows from Parseval's identity and the orthonormality of $\{\phi_j\}_{j\in\mathbb{N}}$. So,

$$
\min_{R\in\mathcal{R}} \max_{f\in\mathcal{F}^{(d)}_{\{\lambda_j\}}} \mathsf{regret}(R, f)
$$

$$
= \min_{\{c_{ij}\}_{i\in[r], j\in\mathbb{N}}} \max_{f\in\mathcal{F}^{(d)}_{\{\lambda_j\}}} \min_{q\in\mathbb{R}^r} \sum_{j=1}^{d} \left( f_j - \sum_{i=1}^{r} q_i c_{ij} \right)^2 + \sum_{j=d+1}^{\infty} \left( \sum_{i=1}^{r} q_i c_{ij} \right)^2. \tag{151}
$$

Evidently, since $\sum_{j=d+1}^{\infty} (\sum_{i=1}^{r} q_i c_{ij})^2 \geq 0$, an optimal representation may satisfy that $c_{ij} = 0$ for all $j \geq d+1$. Thus, the optimal representation belongs to $\mathcal{R}^{(d)}$.

On the other hand,

$$
\mathsf{regret}_{\mathsf{pure}}(\mathcal{R}, \mathcal{F}_{\{\lambda_j\}} \mid P_{\boldsymbol{x}}) = \min_{R\in\mathcal{R}} \max_{f\in\mathcal{F}_{\{\lambda_j\}}} \mathsf{regret}(R, f) \tag{152}
$$

$$
\leq \min_{R\in\mathcal{R}^{(d)}} \max_{f\in\mathcal{F}_{\{\lambda_j\}}} \mathsf{regret}(R, f) \tag{153}
$$

$$
\overset{(*)}{\leq} \min_{R\in\mathcal{R}^{(d)}} \max_{f\in\mathcal{F}^{(d)}_{\{\lambda_j\}}} \mathsf{regret}(R, f) + \lambda_{d+1} \tag{154}
$$

$$
= \mathsf{regret}_{\mathsf{pure}}(\mathcal{R}^{(d)}, \mathcal{F}^{(d)}_{\{\lambda_j\}} \mid P_{\boldsymbol{x}}) + \lambda_{d+1}, \tag{155}
$$

where $(*)$ follows from the following reasoning: For any $(R \in \mathcal{R}^{(d)}, f \in \mathcal{F}_{\{\lambda_j\}})$,

$$
\mathsf{regret}(R, f) = \min_{q\in\mathbb{R}^r} \mathbb{E}\left[ \left( \sum_{j=1}^{\infty} f_j \phi_j(\boldsymbol{x}) + \boldsymbol{n} - \sum_{j=1}^{\infty}\sum_{i=1}^{r} q_i c_{ij} \phi_j(\boldsymbol{x}) \right)^2 \right] - \mathbb{E}[\boldsymbol{n}^2] \tag{156}
$$

$$
\overset{(a)}{=} \min_{q\in\mathbb{R}^r} \sum_{j=1}^{d} \left( f_j - \sum_{i=1}^{r} q_i c_{ij} \right)^2 + \sum_{j=d+1}^{\infty} f_j^2 \tag{157}
$$

$$
\overset{(b)}{\leq} \min_{q\in\mathbb{R}^r} \sum_{j=1}^{d} \left( f_j - \sum_{i=1}^{r} q_i c_{ij} \right)^2 + \lambda_{d+1}, \tag{158}
$$

where $(a)$ follows similarly to the analysis made in the previous step, and $(b)$ follows since for any $f \in \mathcal{F}_{\{\lambda_j\}}$ it holds that

$$
\sum_{j=d+1}^{\infty} f_j^2 \leq \lambda_{d+1} \sum_{j=d+1}^{\infty} \frac{f_j^2}{\lambda_j} \leq \lambda_{d+1} \sum_{j=1}^{\infty} \frac{f_j^2}{\lambda_j} \leq \lambda_{d+1}. \tag{159}
$$

Combining (147) and (155) and using $\lambda_{d+1} \downarrow 0$ completes the proof for the pure minimax regret. The proof for the mixed minimax is analogous and thus is omitted. $\qquad\square$

We also use the following simple and technical lemma.

**Lemma 21.** *For $R \in \mathcal{R}^{(d)}$ and $f \in \mathcal{F}^{(d)}$[1]*

$$
\mathsf{regret}(R, f) = f^\top \left( I_d - R^\top (RR^\top)^{-1} R \right) f, \tag{160}
$$

*where $R \in \mathbb{R}^{r\times d}$ is the matrix of coefficients of the orthogonal expansion of $\psi_i = \sum_{j=1}^{d} c_{ij}\phi_j$ for $i \in [r]$, so that $R(i, j) = c_{ij}$.*

---

[1]Note that any $f \in \mathcal{F}^{(d)}$ may be uniquely identified with a $d$-dimensional vector $f \in \mathbb{R}^d$. With a slight abuse of notation we do not distinguish between the two.

*Proof.* It holds that

$$
\text{regret}(R, f) = \min_{q \in \mathbb{R}^r} \mathbb{E}\left[\left(\sum_{j=1}^d f_j \phi_j(\boldsymbol{x}) + \boldsymbol{n} - \sum_{i=1}^r q_i \sum_{j=1}^d c_{ij} \phi_j(\boldsymbol{x})\right)^2\right] - \mathbb{E}\left[\boldsymbol{n}^2\right] \tag{161}
$$

$$
= \min_{q \in \mathbb{R}^r} \mathbb{E}\left[\sum_{j=1}^d \left(f_j - \sum_{i=1}^r q_i c_{ij}\right) \phi_j(\boldsymbol{x})\right] \tag{162}
$$

$$
= \min_{q \in \mathbb{R}^r} \sum_{j=1}^d \left(f_j - \sum_{i=1}^r q_i c_{ij}\right)^2 \tag{163}
$$

$$
= \min_{q \in \mathbb{R}^r} \sum_{j=1}^d \left[f_j^2 - 2f_j \sum_{i=1}^r q_i c_{ij} + \sum_{i_1=1}^r \sum_{i_2=1}^r q_{i_1} c_{i_1 j} q_{i_2} c_{i_2 j}\right] \tag{164}
$$

$$
= \min_{q \in \mathbb{R}^r} f^\top f - 2q^\top R f + q^\top R R^\top q \tag{165}
$$

$$
= f^\top \left(I_d - R^\top (R R^\top)^{-1} R\right) f, \tag{166}
$$

where the last equality is obtained by the minimizer $q^* = (RR^\top)^{-1}Rf$. □

*Proof of Theorems 18 and 19.* By Proposition 20, we may first consider the finite dimensional case, and then take the limit $d \uparrow \infty$. By Lemma 21, in the $d$-dimensional case (for both the representation and the response function), the regret is formally as in the linear setting under the MSE of Theorem 2, by setting therein $\Sigma_{\boldsymbol{x}} = I_d$, and $S = \text{diag}(\lambda_1, \ldots, \lambda_d)$ (c.f. Lemma 15). The claim of the Theorem 18 then follows by taking $d \uparrow \infty$ and noting that $\lambda_{d+1} \downarrow 0$. The proof of Theorem 19 is analogous and thus omitted. □

# G  Iterative algorithms for the Phase 1 and Phase 2 problems

In this section we describe our proposed algorithms for the solution Phase 1 and Phase 2 problems of Algorithm 1. Those algorithms are general, and only require providing gradients of the regret function (1) and an initial representation and a set of adversarial functions. These are individually determined for each setting. See Section H for the way these are determined in Examples 6 and 8.

## G.1  Phase 1: finding a new adversarial function

We propose an algorithm to solve the Phase 1 problem (26), which is again based on an iterative algorithm. We denote the function's value at the $t$th iteration by $f_{(t)}$. The proposed Algorithm 2 operates as follows. At initialization, the function $f_{(1)} \in \mathcal{F}$ is arbitrarily initialized (say at random), and then the optimal predictor $Q^{(j)}$ is found for each of the $k$ possible representations $R^{(j)}$, $j \in [k]$. Then, the algorithm iteratively repeats the following steps, starting with $t = 2$: (1) Updating the function from $f_{(t-1)}$ to $f_{(t)}$ based on a gradient step of

$$
\sum_{j \in [k]} p^{(j)} \cdot \mathbb{E}\left[\text{loss}(f_{(t-1)}(\boldsymbol{x}), Q^{(j)}(R^{(j)}(\boldsymbol{x})))\right], \tag{167}
$$

that is, the weighted loss function of the previous iteration function, which is then followed by a projection to the feasible class of functions $\mathcal{F}$, denoted as $\Pi_{\mathcal{F}}(\cdot)$ (2) Finding the optimal predictor $Q^{(j)}$ for the current function $f_{(t)}$ and the given representations $\{R^{(j)}\}_{j \in [k]}$, and computing the respective loss for each representation,

$$
L^{(j)} := \mathbb{E}\left[\text{loss}(f_{(t)}(\boldsymbol{x}), Q^{(j)}(R^{(j)}(\boldsymbol{x})))\right]. \tag{168}
$$

This loop iterates for $T_f$ iterations, or until convergence.

---

**Algorithm 2** A procedure for finding a new function via the solution of (26)

---

 1: **procedure** PHASE 1 SOLVER($\{R^{(j)}, p^{(j)}\}_{j \in [k]}, \mathcal{F}, \mathcal{Q}, d, r, P_{\boldsymbol{x}}$)
 2:     **begin**
 3:     Initialize $T_f$                                                    $\triangleright$ Number of iterations parameters
 4:     Initialize $\eta_f$                                                           $\triangleright$ Step size parameter
 5:     Initialize $f_{(1)} \in \mathcal{F}$                           $\triangleright$ Function initialization, e.g., at random
 6:     **for** $j = 1$ to $k$ **do**
 7:         set $Q^{(j)} \leftarrow \arg\min_{Q \in \mathcal{Q}} \mathbb{E}\left[\mathsf{loss}(f_{(1)}(\boldsymbol{x}), Q(R^{(j)}(\boldsymbol{x})))\right]$
 8:     **end for**
 9:     **for** $t = 2$ to $T_f$ **do**
10:         **update** $f_{(t-1/2)} = f_{(t-1)} + \eta_f \cdot \sum_{j \in [k]} p^{(j)}_{(t-1)} \cdot \nabla_f \mathbb{E}\left[\mathsf{loss}(f_{(t-1)}(\boldsymbol{x}), Q^{(j)}(R^{(j)}(\boldsymbol{x})))\right]$
     $\triangleright$ A gradient update of the function
11:         **project** $f_{(t)} = \Pi_{\mathcal{F}}(f_{(t-1/2)})$                        $\triangleright$ Projection on the class $\mathcal{F}$
12:         **for** $j = 1$ to $k$ **do**
13:             set $Q^{(j)} \leftarrow \arg\min_{Q \in \mathcal{Q}} \mathbb{E}\left[\mathsf{loss}(f_{(t)}(\boldsymbol{x}), Q(R^{(j)}(\boldsymbol{x})))\right]$    $\triangleright$ Update of predictors
14:             set $L^{(j)} \leftarrow \mathbb{E}\left[\mathsf{loss}(f_{(t)}(\boldsymbol{x}), Q^{(j)}(R^{(j)}(\boldsymbol{x})))\right]$ $\triangleright$ Compute loss of each representation
15:         **end for**
16:     **end for**
17:     **return** $f_{(T)}$**, and the regret** $\sum_{j \in [k]} p^{(j)} \cdot L^{(j)}$
18: **end procedure**

---

**Design choices and possible variants of the basic algorithm** At initialization, we have chosen a simple random initialization for $f_{(1)}$, but it may also be initialized based on some prior knowledge of the adversarial function. For the update of the predictors, we have specified a full computation of the optimal predictor, which can be achieved in practice by running another iterative algorithm such as stochastic gradient descent (SGD) until convergence. If this is too computationally expensive, the number of gradient steps may be limited. The update of the function is done via projected SGD with a constant step size $\eta_f$, yet it is also possible to modify the step size with the iteration, e.g., the common choice $\eta_f/\sqrt{t}$ at step $t$ Hazan [35]. Accelerated algorithms, e.g., moment-based may also be deployed.

**Convergence analysis** A theoretical analysis of the convergence properties of the algorithm appears to be challenging. Evidently, this is a minimax game between the response player and the predictor player, but not a concave-convex game. As described in Appendix B, even concave-convex games are not well understood at this point. We thus opt to validate this algorithm numerically.

### G.2 Phase 2: finding a new representation

We propose an iterative algorithm to solve the Phase 2 problem (27), and thus finding a new representation $R^{(k+1)}$. To this end, we first note that the objective function in (27) can be separated into a part that depends on existing representations and a part that depends on the new one, specifically, as

$$
\sum_{j_1 \in [k]} \sum_{j_2 \in [m_0 + k]} p^{(j_1)} \cdot o^{(j_2)} \cdot \mathbb{E}\left[\mathsf{loss}(f^{(j_2)}(\boldsymbol{x}), Q^{(j_1, j_2)}(R^{(j_1)}(\boldsymbol{x})))\right]
$$
$$
+ \sum_{j_2 \in [m_0 + k]} p^{(k+1)} \cdot o^{(j_2)} \cdot \mathbb{E}\left[\mathsf{loss}(f^{(j_2)}(\boldsymbol{x}), Q^{(k+1, j_2)}(R^{(k+1)}(\boldsymbol{x})))\right]
$$
$$
= \sum_{j_1 \in [k]} \sum_{j_2 \in [m_0 + k]} p^{(j_1)} \cdot o^{(j_2)} \cdot L^{(j_1, j_2)}
$$
$$
+ \sum_{j_2 \in [m_0 + k]} p^{(k+1)} \cdot o^{(j_2)} \cdot \mathbb{E}\left[\mathsf{loss}(f^{(j_2)}(\boldsymbol{x}), Q^{(k+1, j_2)}(R^{(k+1)}(\boldsymbol{x})))\right], \tag{169}
$$

where

$$
L^{(j_1, j_2)} := \mathbb{E}\left[\mathsf{loss}(f^{(j_2)}(\boldsymbol{x}), Q^{(j_1, j_2)}(R^{(j_2)}(\boldsymbol{x})))\right], \tag{170}
$$

and the predictors $\{Q^{(j_1,j_2)}\}_{j_1\in[k],j_2\in[m_0+k]}$ can be optimized independently of the new representation $R^{(k+1)}$. We propose an iterative algorithm for this problem, and denote the new representation at the $t$th iteration of the algorithm by $R_{(t)}^{(k+1)}$. The algorithm's input is a set of $m_0 + k$ adversarial functions $\{f^{(i)}\}_{i\in[m_0+k]}$, and the current set of representations $\{R^{(j)}\}_{j\in[k]}$. Based on these, the algorithm may find the optimal predictor for $f^{(j_2)}$ based on the representation $R^{(j_1)}$, and thus compute the loss

$$L_*^{(j_1,j_2)} := \min_{Q\in\mathcal{Q}} \mathbb{E}\left[\mathsf{loss}(f^{(j_2)}(\boldsymbol{x}), Q(R^{(j_1)}(\boldsymbol{x})))\right] \tag{171}$$

for $j_1 \in [k]$ and $j_2 \in [m_0 + k]$. In addition, the new representation is arbitrarily initialized (say, at random) as $R_{(1)}^{(k+1)}$, and the predictors $\{Q_{(1)}^{(k+1,j_2)}\}_{j_2\in[m_0+k]}$ are initialized as the optimal predictors for $f^{(j_2)}$ given the representation $R_{(1)}^{(k+1)}$. The algorithm keeps track of weights for the representations (including the new one), which are initialized uniformly, i.e., $p_{(1)}^{(j_1)} = \frac{1}{k+1}$ for $j_1 \in [k+1]$ (including a weight for the new representation). The algorithm also keeps track of weights for the functions, which are also initialized uniformly as $o_{(1)}^{(j_2)} = \frac{1}{m_0+k}$ for $j_2 \in [m_0 + k]$. Then, the algorithm iteratively repeats the following steps, starting with $t = 2$: (1) Updating the new representation from $R_{(t-1)}^{(k+1)}$ to $R_{(t)}^{(k+1)}$ based on a gradient step of the objective function (27) as a function of $R^{(k+1)}$. Based on the decomposition in (169) the term of the objective which depends on $R^{(k+1)}$ is

$$p_{(t-1)}^{(k+1)} \sum_{j_2\in[m_0+k]} o_{(t-1)}^{(j_2)} \cdot \mathbb{E}\left[\mathsf{loss}(f^{(j_2)}(\boldsymbol{x}), Q^{(k+1,j_2)}(R^{(k+1)}(\boldsymbol{x})))\right], \tag{172}$$

that is, the loss function of the previous iteration new representation, weighted according to the current function weights $o_{(t-1)}^{(j_2)}$. Since the multiplicative factor $p_{(t-1)}^{(k+1)}$ is common to all terms, it is removed from the gradient computation (this aids in the choice of the gradient step). This gradient step is then possibly followed by normalization or projection, which we denote by the operator $\Pi_{\mathcal{R}}(\cdot)$. For example, in the linear case, it make sense to normalize $R^{(k+1)}$ to have unity norm (in some matrix norm of choice). After updating the new representation to $R_{(t)}^{(k+1)}$, optimal predictors are found for each function, the loss is computed

$$L_{(t)}^{(k+1,j_2)} := \min_{Q\in\mathcal{Q}} \mathbb{E}\left[\mathsf{loss}(f^{(j_2)}(\boldsymbol{x}), Q(R_{(t)}^{(k+1)}(\boldsymbol{x})))\right] \tag{173}$$

for all $j_2 \in [m_0 + k]$, and the optimal predictor is updated to $\{Q_{(t)}^{(k+1,j_2)}\}_{j_2\in[m_0+k]}$ based on this solution. (2) Given the current new representation $R_{(t)}^{(k+1)}$, the loss matrix

$$\{L_{(t)}^{(j_1,j_2)}\}_{j_1\in[k],j_2\in[m_0+k]} \tag{174}$$

is constructed where for $j_1 \in [k]$ it holds that $L_{(t)}^{(j_1,j_2)} = L^{(j_1,j_2)}$ for all $t$ (i.e., the loss of previous representations and functions is kept fixed). This is considered to be the loss matrix of a two-player zero-sum game between the representation player and the function player, where the representation player has $k + 1$ possible strategies and the function player has $m_0 + k$ strategies. The weights $\{p_{(t)}^{(j_1)}\}_{j_1\in[k+1]}$ and $\{o_{(t)}^{(j_2)}\}_{j_2\in[m_0+k]}$ are then updated according to the MWU rule. Specifically, for an *inverse temperature parameter* $\beta$ (or a *regularization parameter*), the update is given by

$$p_{(t)}^{(j)} = \frac{p_{(t-1)}^{(j)} \cdot \beta^{L^{(j)}}}{\sum_{\tilde{j}\in[k]} p_{(t-1)}^{(\tilde{j})} \cdot \beta^{L^{(\tilde{j})}}} \tag{175}$$

for the representation weights and, analogously, by

$$o_{(t)}^{(j)} = \frac{o_{(t-1)}^{(j)} \cdot \beta^{-L^{(j)}}}{\sum_{\tilde{j}\in[k]} o_{(t-1)}^{(\tilde{j})} \cdot \beta^{-L^{(\tilde{j})}}} \tag{176}$$

for the function weights (as the function player aims to maximize the loss). This can be considered as a regularized gradient step on the probability simplex, or more accurately, a *follow-the-regularized-leader* [35]. The main reasoning of this algorithm is that at each iteration the weights $\{p^{(j)}\}_{j\in[k+1]}$

and $\{o^{(j)}\}_{j\in[m_0+k]}$ are updated towards the solution of the two-player zero-sum game with payoff matrix $\{-L_{(t)}^{(j_1,j_2)}\}_{j_1\in[k+1],j_2\in[m_0+k]}$. In turn, based only on the function weights $\{o^{(j)}\}_{j\in[m_0+k]}$, the new representation is updated to $R_{(t)}^{(k+1)}$, which then changes the pay-off matrix at the next iteration. It is well known that the MWU solved two-player zero-sum game [33], in which the representation player can choose the weights and the function player can choose the function.

This loop iterates for $T_{\text{stop}}$ iterations, and then the optimal weights are given by the average over the last $T_{\text{avg}}$ iterations [33], i.e.,

$$p_*^{(j)} = \frac{1}{T_{\text{avg}}} \sum_{t=T_{\text{stop}}-T_{\text{avg}}+1}^{T_{\text{stop}}} p_{(t)}^{(j)}, \tag{177}$$

and

$$o_*^{(j)} = \frac{1}{T_{\text{avg}}} \sum_{t=T_{\text{stop}}-T_{\text{avg}}+1}^{T_{\text{stop}}} o_{(t)}^{(j)}. \tag{178}$$

In the last $T_R - T_{\text{stop}}$ iterations, only the representation $R_{(t)}^{(k+1)}$ and the predictors are updated. The algorithm then outputs $R_{(T)}^{(k+1)}$ as the new representation and the weights $\{p_*^{(j)}\}_{j\in[k+1]}$.

**Design choices and possible variants of the basic algorithm** At initialization, we have chosen a simple random initialization for $R_{(1)}^{(k+1)}$, but it may also be initialized based on some prior knowledge of the desired new representation. The initial predictors $\{Q_{(1)}^{(k+1,j_2)}\}_{j_2\in[m_0+k]}$ will then be initialized as the optimal predictors for $R_{(1)}^{(k+1)}$ and $\{f^{(j_2)}\}_{j_2\in[m_0+k]}$. We have initialized the representation and function weights uniformly. A possibly improved initialization for the function weights is to put more mass on the more recent functions, that is, for large values of $j_2$, or to use the minimax strategy of the function player in the two-player zero-sum game with payoff matrix $\{-L_{(t)}^{(j_1,j_2)}\}_{j_1\in[k],j_2\in[m_0+k]}$ (that is, a game which does not include the new representation). As in the Phase 1 algorithm, the gradient update of the new representation can be replaced by a more sophisticated algorithm, the computation of the optimal predictors can be replaced with (multiple) update steps, and the step size may also be adjusted. For the MWU update, we use the proposed scaling in [33]

$$\beta = \frac{1}{1 + \sqrt{\frac{c \ln m}{T}}} \tag{179}$$

for some constant $c$. It is well known that using the last iteration of a MWU algorithm may fail [97], while averaging the weights value of all iterations provides the optimal value of a two-player zero-sum games [33]. For improved accuracy, we compute the average weights over the last $T_{\text{avg}}$ iterations (thus disregarding the initial iterations). We then halt the weights update and let the function and predictor update to run for $T - T_{\text{stop}}$ iterations in order to improve the convergence of $R^{(k+1)}$. Finally, the scheduling of the steps may be more complex, e.g., it is possible that running multiple gradient steps follows by multiple MWU steps may improve the result.

## H Details for the examples of Algorithm 1

As mentioned, the solvers of the Phase 1 and Phase 2 problems of Algorithm 1 require the gradients of the regret (1) as inputs, as well as initial representation and set of adversarial functions. We next provide these details for the examples in Section 4. The code for the experiments was written in `Python 3.6` the code is available at this link. The optimization of hyperparameters was done using the `Optuna` library. The hardware used is standard and detailed appear in Table 1.

### H.1 Details for Example 6: the linear MSE setting

In this setting, the expectation over the feature distribution can be carried out analytically, and the regret is given by

$$\text{regret}(R, f \mid \Sigma_{\boldsymbol{x}}) = \mathbb{E}\left[\left(f^\top \boldsymbol{x} - q^\top R^\top \boldsymbol{x}\right)^2\right] \tag{180}$$

---

**Algorithm 3** A procedure for finding a new representation $R^{(k+1)}$ via the solution of (27)

---

1: **procedure** PHASE 2 SOLVER($\{R^{(j_1)}\}_{j\in[k]}, \{f^{(j_2)}\}_{j_2\in[m_0+k]}, \mathcal{R}, \mathcal{F}, \mathcal{Q}, d, r, P_{\boldsymbol{x}}$)

2:     **begin**

3:         Initialize $T_R, T_{\text{stop}}, T_{\text{avg}}$                                  ▷ Number of iterations parameters

4:         Initialize $\eta_R$                                                         ▷ Step size parameter

5:         Initialize $\beta \in (0,1)$                                            ▷ Inverse temperature parameter

6:         Initialize $f_{(1)} \in \mathcal{F}$                                   ▷ Function initialization, e.g., at random

7:         Initialize $p_{(1)}^{(j)} \leftarrow 0$ for $j \in [k]$ and $p_{(1)}^{(k+1)} \leftarrow 0$       ▷ A uniform weight initialization for the
    representations

8:         Initialize $o_{(1)}^{(j_2)} \leftarrow \frac{1}{m_0+k}$ for $j_2 \in [k]$        ▷ A uniform weight initialization for the functions

9:         **for** $j_1 = 1$ to $k$ **do**

10:             **for** $j_2 = 1$ to $m_0 + k$ **do**

11:                 **Set** $Q^{(j_1,j_2)} \leftarrow \arg\min_{Q\in\mathcal{Q}} \mathbb{E}\left[\text{loss}(f^{(j_2)}(\boldsymbol{x}), Q(R^{(j_1)}(\boldsymbol{x})))\right]$
    ▷ Optimal predictors for existing representations and input functions

12:                 **Set** $L^{(j_1,j_2)} \leftarrow \min_{Q\in\mathcal{Q}} \mathbb{E}\left[\text{loss}(f^{(j_2)}(\boldsymbol{x}), Q^{(j_1,j_2)}(R^{(j_1)}(\boldsymbol{x})))\right]$   ▷ The minimal loss

13:             **end for**

14:         **end for**

15:         **for** $j_2 = 1$ to $m_0 + k$ **do**

16:             **Initialize** $R_{(1)}^{(k+1)}$                                 ▷ Arbitrarily, e.g., at random

17:             **Set** $Q_{(1)}^{(k+1,j_2)} \leftarrow \arg\min_{Q\in\mathcal{Q}} \mathbb{E}\left[\text{loss}(f^{(j_2)}(\boldsymbol{x}), Q(R^{(k+1)}(\boldsymbol{x})))\right]$ for $j_2 \in [m_0 + k]$
    ▷ Optimal predictors for new representation and input functions

18:         **end for**

19:         **for** $t = 2$ to $T_R$ **do**

20:             **update**                                          ▷ A gradient update of the new representation

$$R_{(t-1/2)}^{(k+1)} = R_{(t-1)}^{(k+1)} + \eta_R \cdot \sum_{j_2\in[m_0+k]} o_{(t-1)}^{(j_2)} \cdot \nabla_{R^{(k+1)}} \mathbb{E}\left[\text{loss}(f^{(j_2)}(\boldsymbol{x}), Q^{(k+1,j_2)}(R_{(t-1)}^{(k+1)}(\boldsymbol{x})))\right]$$

21:             **projection** $R_{(t)}^{(k+1)} = \Pi_{\mathcal{R}}(R_{(t-1/2)}^{(k+1)})$                ▷ Standardization based on the class $\mathcal{R}$

22:             **for** $j = 1$ to $k$ **do**

23:                 **Set** $Q^{(k+1,j_2)} \leftarrow \arg\min_{Q\in\mathcal{Q}} \mathbb{E}\left[\text{loss}(f^{(j_2)}(\boldsymbol{x}), Q(R_{(t)}^{(k+1)}(\boldsymbol{x})))\right]$
    ▷ Update of predictors for the new representation

24:                 $L_{(t)}^{(k+1,j_2)} \leftarrow \mathbb{E}\left[\text{loss}((f^{(j_2)}(\boldsymbol{x}), Q^{(k+1,j_2)}(R_{(t)}^{(k+1)}(\boldsymbol{x})))\right]$                ▷ Compute loss

25:             **end for**

26:             **Set** $L_{(t)}^{(j_1,j_2)} \leftarrow L^{(j_1,j_2)}$ for $j_1 \in [k]$ and $j_2 \in [m_0 + k]$

27:             **if** $t < T_{\text{stop}}$ **then**

28:                 **update** $p_{(t)}^{(j)} \leftarrow \frac{p_{(t-1)}^{(j)} \cdot \beta^{L^{(j)}}}{\sum_{\tilde{j}\in[k]} p_{(t-1)}^{(\tilde{j})} \cdot \beta^{L^{(\tilde{j})}}}$ for $j \in [k]$                       ▷ A MWU

29:                 **update** $o_{(t)}^{(j)} \leftarrow \frac{o_{(t-1)}^{(j)} \cdot \beta^{-L^{(j)}}}{\sum_{\tilde{j}\in[m_0+k]} o_{(t-1)}^{(\tilde{j})} \cdot \beta^{-L^{(\tilde{j})}}}$ for $j \in [m_0 + k]$             ▷ A MWU

30:             **else if** $t = T_{\text{stop}}$ **then**

31:                 **update** $p_{(t)}^{(j)} = p_{(t)}^{(j)} \leftarrow \frac{1}{T_{\text{avg}}} \sum_{t=T_{\text{stop}}-T_{\text{avg}}+1}^{T_{\text{stop}}} p_{(t)}^{(j)}$ for $j \in [k]$
    ▷ Optimal weights by averaging last $T_{\text{avg}}$ iterations

32:                 **update** $o_{(t)}^{(j)} \leftarrow \frac{1}{T_{\text{avg}}} \sum_{t=T_{\text{stop}}-T_{\text{avg}}+1}^{T_{\text{stop}}} o_{(t)}^{(j)}$ for $j \in [m_0 + k]$
    ▷ Optimal weights by averaging last $T_{\text{avg}}$ iterations

33:             **else**

34:                 **update** $p_{(t)}^{(j)} \leftarrow p_{(t-1)}^{(j)}$ for $j \in [k]$              ▷ No update for the last $T - T_{\text{stop}}$ iterations

35:                 **update** $o_{(t)}^{(j)} \leftarrow o_{(t-1)}^{(j)}$ for $j \in [m_0 + k]$ ▷ No update for the last $T - T_{\text{stop}}$ iterations

36:             **end if**

37:             **return** $R_{(T)}^{(k+1)}$ **and** $\{p_{(T_R)}^{(j)}\}_{j\in[k+1]}$

38:         **end for**

39: **end procedure**

---

Table 1: Hardware details

| CPU | RAM | GPU |
|---|---|---|
| Intel i9 13900k | 64GB | RTX 3090 Ti |

$$= f^\top \Sigma_{\boldsymbol{x}} f - 2q^\top R^\top \Sigma_{\boldsymbol{x}} f + q^\top R^\top R q. \tag{181}$$

The regret only depends on the feature distribution $P_{\boldsymbol{x}}$ via $\Sigma_{\boldsymbol{x}}$. For each run of the algorithm, the co-variance matrix $\Sigma_{\boldsymbol{x}}$ was chosen to be diagonal with elements drawn from a log-normal distribution, with parameters $(0, \sigma_0)$, and $S = I_d$.

**Regret gradients**    The gradient of the regret w.r.t. the function $f$ is given by

$$\nabla_f \mathbb{E}\left[\left(f^\top \boldsymbol{x} - q^\top R^\top \boldsymbol{x}\right)^2\right] = 2f^\top \Sigma_{\boldsymbol{x}} - 2q^\top R^\top \Sigma_{\boldsymbol{x}} \tag{182}$$

and the projection on $\mathcal{F}_S$ is

$$\Pi_{\mathcal{F}}(f) = \begin{cases} \frac{f}{\|f\|_S}, & \|f\|_S \geq 1 \\ f, & \|f\|_S < 1 \end{cases}. \tag{183}$$

However, we may choose to normalize by $\frac{f}{\|f\|_S}$ even if $\|f\|_S \leq 1$ since in this case the regret is always larger if $f$ is replaced by $\frac{f}{\|f\|_S}$ (in other words, the worst case function is obtained on the boundary of $\mathcal{F}_S$). The gradient w.r.t. the predictor $q$ is given by

$$\nabla_q \mathbb{E}\left[\left(f^\top \boldsymbol{x} - q^\top R^\top \boldsymbol{x}\right)^2\right] = \left[-2f^\top \Sigma_{\boldsymbol{x}} R + 2q^\top R^\top \Sigma_{\boldsymbol{x}} R\right]. \tag{184}$$

Finally, to derive the gradient w.r.t. $R$, let us denote $R := [R_1, R_2, \ldots, R_r] \in \mathbb{R}^{d \times r}$ where $R_i \in \mathbb{R}^d$ is the $i$th column ($i \in [r]$), and $q^\top = (q_1, q_2, \ldots, q_r)$. Then, $q^\top R^\top \boldsymbol{x} = \sum_{i \in [d]} q_i R_i^\top \boldsymbol{x}$ and the loss function is

$$\mathbb{E}\left[\left(f^\top \boldsymbol{x} - q^\top R^\top \boldsymbol{x}\right)^2\right] = \mathbb{E}\left[\left(f^\top \boldsymbol{x} - \sum_{i \in [d]} q_i \boldsymbol{x}^\top R_i\right)^2\right] \tag{185}$$

$$= f^\top \Sigma_{\boldsymbol{x}} f - 2q^\top R^\top \Sigma_{\boldsymbol{x}} f + q^\top R^\top \Sigma_{\boldsymbol{x}} R q. \tag{186}$$

The gradient of the regret w.r.t. $R_k$ is then given by

$$\nabla_{R_k} \left\{\mathbb{E}\left[\left(f^\top \boldsymbol{x} - q^\top R^\top \boldsymbol{x}\right)^2\right]\right\} = -2\mathbb{E}\left[\left(f^\top \boldsymbol{x} - q^\top R^\top \boldsymbol{x}\right) \cdot q_k \boldsymbol{x}^\top\right] \tag{187}$$

$$= -2q_k \left(f^\top \Sigma_{\boldsymbol{x}} - q^\top R^\top \Sigma_{\boldsymbol{x}}\right), \tag{188}$$

hence, more succinctly, the gradient w.r.t. $R$ is

$$\nabla_R \left\{\mathbb{E}\left[\left(f^\top \boldsymbol{x} - q^\top R^\top \boldsymbol{x}\right)^2\right]\right\} = -2q \left(f^\top \Sigma_{\boldsymbol{x}} - q^\top R^\top \Sigma_{\boldsymbol{x}}\right). \tag{189}$$

We remark that in the algorithm these gradients are multiplied by weights. We omit this term whenever the weight is common to all terms in order to keep the effective step size constant.

**Initialization**    Algorithm 1 requires an initial representation $R^{(1)}$ and an initial set of functions $\{f^{(j)}\}_{j \in [m_0]}$. In the MSE setting, each function $f \in \mathbb{R}^d$ is also a single column of a representation matrix $R \in \mathbb{R}^{d \times r}$. A plausible initialization matrix $R^{(1)} \in \mathbb{R}^{d \times r}$ is therefore the worst $r$ functions. These, in turn, can be found by running Algorithm (1) to obtain $\tilde{m} = r$ functions, by setting $\tilde{r} = 1$. A proper initialization for this run is simply an all-zero representation $\tilde{R}^{(1)} = 0 \in \mathbb{R}^{d \times 1}$. The resulting output is then $\{\tilde{R}_{(T)}^{(j)}\}_{j \in [r]}$ which can be placed as the $r$ columns of $R^{(1)}$. This initialization is then used for Algorithm 1.

**Algorithm parameters**    The algorithm parameters used for Example 6 are shown in Table 2. The parameters were optimally tuned for $\sigma_0 = 1$.

Table 2: Parameters for linear MSE setting example

| Parameter | $\beta_r$ | $\beta_f$ | $\eta_r$ | $\eta_f$ |
|-----------|-----------|-----------|----------|----------|
| Value | 0.94 | 0.653 | 0.713 | 0.944 |
| Parameter | $T_R$ | $T_f$ | $T_{\text{avg}}$ | $T_{\text{stop}}$ |
| Value | 100 | until convergence | 10 | 80 |

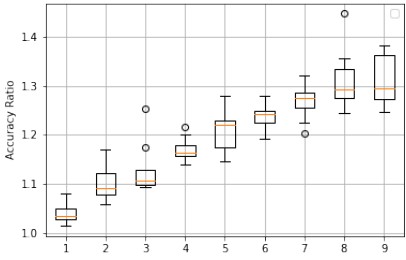 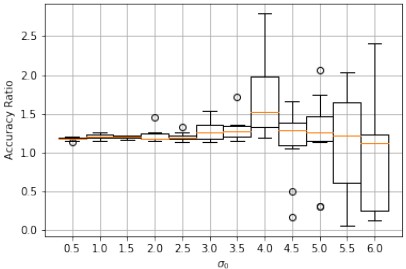

Figure 3: The ratio between the regret achieved by Algorithm 1 and the theoretical regret in the linear MSE setting. Left: $d = 20$, $\sigma_0 = 1$, varying $r$. Right: $r = 5$, $d = 20$, varying $\sigma_0$.

**Additional results** Additional results of the accuracy of the Algorithm 1 in the linear MSE setting are displayed in Figure 3. The left panel of Figure 3 shows that the algorithm output is accurate for small values of $r$, but deteriorates as $r$ increases. This is because when $r$ increases then so is $\ell^*$ and so is the required number of matrices in the support of the representation rule (denoted by $m$). Since the algorithm gradually adds representation matrices to the support, an inaccurate convergence at an early iteration significantly affects later iterations. One possible way to remedy this is to run each iteration multiple times, and choose the best one, before moving on to the next one. Another reason is that given large number of matrices in the support (large $m$), it becomes increasingly difficult for the the MWU to accurately converge. Since the iterations of the MWU do not converge to the equilibrium point, but rather their average (see discussion in Appendix B) this can only be remedied by allowing more iterations for convergence (in advance) for large values of $m$. The right panel of Figure 3 shows that the algorithm output is accurate for a wide range of the condition number of the covariance matrix. This condition number is determined by the choice of $\sigma_0$, where low values typically result covariance matrices with condition number that is close to 1, while high values will typically result large condition number. The right panel shows that while the hyperparameters were tuned for $\sigma_0 = 1$, the result is fairly accurate for a wide range of $\sigma_0$ values, up to $\sigma_0 \approx 5$. Since for $Z \sim N(0, 1)$ (standard normal) it holds that $\mathbb{P}[-2 < Z < 2] \approx 95\%$, the typical condition number of a covariance matrix drawn with $\sigma_0 = 5$ is roughly $\frac{e^{2\sigma_0}}{e^{-2\sigma_0}} \approx 4.85 \cdot 10^8$, which is a fairly large range.

### H.2  Details for Example 8: the linear cross-entropy setting

In this setting,

$$\text{regret}(R, f \mid P_{\boldsymbol{x}}) = \min_{q \in \mathbb{R}^r} \mathbb{E}\left[D_{\text{KL}}\left([1 + \exp(-f^\top \boldsymbol{x})]^{-1} \mid\mid [1 + \exp(-q^\top R^\top \boldsymbol{x})]^{-1}\right)\right], \quad (190)$$

and the expectation over the feature distribution typically cannot be carried out analytically. We thus tested Algorithm 1 on empirical distributions of samples drawn from a high-dimensional normal distribution. Specifically, for each run, $B = 1000$ feature vectors were drawn from an isotropic normal distribution of dimension $d = 15$. The expectations of the regret and the corresponding gradients were then computed with respect to (w.r.t.) the resulting empirical distributions.

**Regret gradients** We use the facts that

$$\frac{\partial}{\partial p_1} D_{\text{KL}}(p_1 \mid\mid p_2) = \log \frac{p_1(1 - p_2)}{p_2(1 - p_1)} \quad (191)$$

and

$$\frac{\partial}{\partial p_2} D_{\mathrm{KL}}(p_1 \| p_2) = \frac{p_2 - p_1}{p_2(1 - p_2)}. \tag{192}$$

For brevity, let us next denote

$$p_1 := \frac{1}{1 + \exp(-f^\top \boldsymbol{x})} \tag{193}$$

and

$$p_2 := \frac{1}{1 + \exp(-q^\top R^\top \boldsymbol{x})}. \tag{194}$$

We next repeatedly use the chain rule for differentiation. First,

$$\nabla_f p_1 = \nabla_f \left[ \frac{1}{1 + \exp(-f^\top \boldsymbol{x})} \right] = \frac{\exp(-f^\top \boldsymbol{x}) \cdot \boldsymbol{x}}{[1 + \exp(-f^\top \boldsymbol{x})]^2} = p_1(1 - p_1) \cdot \boldsymbol{x}. \tag{195}$$

and

$$\nabla_q p_2 = \nabla_q \left[ \frac{1}{1 + \exp(-q^\top R^\top \boldsymbol{x})} \right] = \frac{\exp(-q^\top R^\top \boldsymbol{x}) \cdot R^\top \boldsymbol{x}}{[1 + \exp(-q^\top R^\top \boldsymbol{x})]^2} = p_2(1 - p_2) \cdot R^\top \boldsymbol{x}. \tag{196}$$

So, assuming that $P_{\boldsymbol{x}}$ is such that the order of differentiation and expectation may be interchanged (this can be guaranteed using dominated/monotone convergence theorems), the gradient of the regret w.r.t. $f$ is

$$\nabla_f \mathsf{regret}(R, f \mid P_{\boldsymbol{x}}) = \mathbb{E}\left[ \frac{\partial}{\partial p_1} D_{\mathrm{KL}}(p_1 \| p_2) \times \nabla_f p_1 \right] \tag{197}$$

$$= \mathbb{E}\left[ \log\left( \frac{p_1(1 - p_2)}{p_2(1 - p_1)} \right) \cdot p_1(1 - p_1) \cdot \boldsymbol{x} \right] \tag{198}$$

$$= \mathbb{E}\left[ (f^\top - q^\top R^\top)\boldsymbol{x} \frac{\exp(-f^\top \boldsymbol{x})}{[1 + \exp(-f^\top \boldsymbol{x})]^2} \cdot \boldsymbol{x} \right] \tag{199}$$

$$= \mathbb{E}\left[ \frac{\exp(-f^\top \boldsymbol{x})}{[1 + \exp(-f^\top \boldsymbol{x})]^2} \cdot \boldsymbol{x}^\top (f - Rq)\boldsymbol{x} \right]. \tag{200}$$

Next, under similar assumptions, the gradient of the regret w.r.t. the predictor $q$ is

$$\nabla_q \mathsf{regret}(R, f \mid P_{\boldsymbol{x}}) = \mathbb{E}\left[ \frac{\partial}{\partial p_2} D_{\mathrm{KL}}(p_1 \| p_2) \times \nabla_q p_2 \right] \tag{201}$$

$$= \mathbb{E}\left[ \left( \frac{1}{1 + \exp(-q^\top R^\top \boldsymbol{x})} - \frac{1}{1 + \exp(-f^\top \boldsymbol{x})} \right) \cdot R^\top \boldsymbol{x} \right]. \tag{202}$$

Finally, as for the MSE case, to derive the gradient w.r.t. $R$, we denote $R := [R_1, R_2, \ldots, R_r] \in \mathbb{R}^{d \times r}$ where $R_i \in \mathbb{R}^d$ is the $i$th column ($i \in [r]$), and $q^\top = (q_1, q_2, \ldots, q_r)$. Then, $q^\top R^\top \boldsymbol{x} = \sum_{i \in [d]} q_i R_i^\top \boldsymbol{x}$ and

$$p_2 = \frac{1}{1 + \exp(-\sum_{i \in [d]} q_i R_i^\top \boldsymbol{x})}. \tag{203}$$

Then, the gradient of $p_2$ w.r.t. $R_k$ is then given by

$$\nabla_{R_k} p_2 = p_2(1 - p_2) \cdot q_i \boldsymbol{x}, \tag{204}$$

hence, more succinctly, the gradient w.r.t. $R$ is

$$\nabla_R p_2 = p_2(1 - p_2) \cdot \boldsymbol{x} q^\top. \tag{205}$$

Hence,

$$\nabla_R \mathsf{regret}(R, f \mid P_{\boldsymbol{x}}) = \mathbb{E}\left[ \frac{\partial}{\partial p_2} D_{\mathrm{KL}}(p_1 \| p_2) \times \nabla_R p_2 \right] \tag{206}$$

$$= \mathbb{E}\left[ (p_2 - p_1) \cdot \boldsymbol{x} q^\top \right] \tag{207}$$

$$= \mathbb{E}\left[ \left( \frac{1}{1 + \exp(-q^\top R^\top \boldsymbol{x})} - \frac{1}{1 + \exp(-f^\top \boldsymbol{x})} \right) \cdot \boldsymbol{x} q^\top \right]. \tag{208}$$

Table 3: Parameters for linear cross entropy setting example

| Parameter | $\beta_r$ | $\beta_f$ | $\eta_r$ | $\eta_f$ |
|-----------|-----------|-----------|-----------|-----------|
| Value | 0.9 | 0.9 | $10^{-3}$ | $10^{-1}$ |
| Parameter | $T_R$ | $T_f$ | $T_{\text{avg}}$ | $T_{\text{stop}}$ |
| Value | 100 | 1000 | 25 | 50 |

**Initialization** Here the initialization is similar to the linear MSE setting, except that since a column of the representation cannot ideally capture even a single adversarial function, the initialization algorithm only searches for a single adversarial function ($\tilde{m} = 1$). This single function is then used to produce $R^{(1)}$ as the initialization of Algorithm 1.

**Algorithm parameters** The algorithm parameters used for Example 8 are shown in Table 3.

# I An experiment with a NN architecture

In the analysis and the experiments above we have considered basic linear functions. As mentioned, since the operation of Algorithm 1 only depends on the gradients of the loss function, it can be easily generalized to representations, response functions and predictors for which such gradients (or sub-gradients) can be provided. In this section, we exemplify this idea with a simple NN architecture. For $x \in \mathbb{R}^d$, we let the rectifier linear unit (ReLU) be denoted as $(x)_+$.

**Definition 22** (The NN setting). Assume the same setting as in Definitions 1 and 7, except that the class of representation, response and predictors are NN with $c$ hidden layers of sizes $h_R, h_f, h_q \in \mathbb{N}_+$, respectively, instead of linear functions. Specifically: (1) The representation is

$$R(x) = R_c^\top \left( \cdots \left( R_1^\top (R_0^\top x)_+ \right)_+ \right)_+ \tag{209}$$

for some $(R_0, R_1, \cdots R_c) \in \mathcal{R} := \{\mathbb{R}^{d \times h_R} \times \mathbb{R}^{h_R \times h_R} \cdots \mathbb{R}^{h_R \times h_R} \times \mathbb{R}^{h_R \times r}\}$ where $d > r$. (2) The response is determined by

$$f(x) = f_c^\top \left( \cdots \left( F_1^\top (F_0^\top x)_+ \right)_+ \right)_+ \tag{210}$$

where $(F_0, F_1, \ldots, f_c) \in \mathcal{F} := \{\mathbb{R}^{d \times h_f} \times \mathbb{R}^{h_f \times h_f} \cdots \mathbb{R}^{h_f \times h_f} \times \mathbb{R}^{h_f}\}$. (3) The predictor is determined by for some

$$q(z) = q_c^\top \left( \cdots \left( Q_1^\top (Q_0^\top z)_+ \right)_+ \right)_+ \tag{211}$$

where $(Q_0, Q_1, \ldots, q_c) \in \mathcal{Q} := \{\mathbb{R}^{r \times h_q} \times \mathbb{R}^{h_q \times h_q} \cdots \mathbb{R}^{h_q \times h_q} \times \mathbb{R}^{h_q}\}$.

**Regret gradients** Gradients were computed using `PyTorch` with standard gradients computation using backpropagation for an SGD optimizer.

**Initialization** The initialization algorithm is similar to the initialization algorithm used in the linear cross-entropy setting.

**Algorithm parameters** The algorithm parameters used for the example are shown in Table 4.

**Results** For a single hidden layer, Figure 4 shows the reduction of the regret with the iteration for the cross-entropy loss.

Table 4: Parameters for the NN cross-entropy setting.

| Parameter | $c$ | $h_R$ | $h_f$ | $h_q$ | |
|-----------|-----|-------|-------|-------|---|
| Value | 1 | $d$ | $d$ | $d$ | |
| Parameter | $\beta_r$ | $\beta_f$ | $\eta_r$ | $\eta_f$ | $\eta_q$ |
| Value | 0.9 | 0.9 | $10^{-3}$ | $10^{-1}$ | $10^{-1}$ |
| Parameter | $T_R$ | $T_f$ | $T_Q$ | $T_{\text{avg}}$ | $T_{\text{stop}}$ |
| Value | 100 | 1000 | 100 | 10 | 80 |

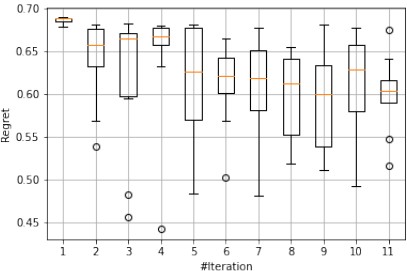

Figure 4: The regret achieved by Algorithm 1 in the NN cross-entropy setting as a function of the iteration $m$.