# OpenReview forum: "A representation-learning game for classes of prediction tasks"
_NeurIPS.cc/2023/Conference — Submitted to NeurIPS 2023_

### Official Review · Reviewer_WSXs · 2023-07-05

**Soundness:** 2 fair
**Presentation:** 3 good
**Contribution:** 2 fair
**Rating:** 6
**Confidence:** 4

**Summary:**

This paper extends and formulates learning representation procedure as a three-player game when the downstream task is known. Unlike conventional unsupervised learning algorithms, the proposed framework helps the learning process in the feature space be directed. They set up a general and basic mathematic form (linear-MSE) to study, which proves the utility of prior information. And, they extend results via studying mixed strategies, which shows the importance to use randomized representations. They propose gradient-based iterative algorithms to optimize objectives and validate the effectiveness of their framework with examples (i.e., linear-MSE and linear-CE).


**Strengths:**

This paper presents several strengths, including a clear and well-written introduction that guides the reader to understand the problem and the motivations behind it. The findings from this paper is interesting: it shows the benefits to use randomized representation. Besides, the authors provide concrete steps to introduce iterative algorithm for their general cases to establish it, although the takeaway conclusion is not always straightforward to see, such as in the figure for Example 5.

**Weaknesses:**

Here are some potential improvements to the original content:

1. The paper could benefit from a more concrete approach to evaluating the effectiveness of the proposed framework. While using regret as a metric is a relative approach, it may be more informative to show performance using exact MSE. For example, the paper could compare results between a conventional representation (without prior knowledge about downstream task) and a representation found by the proposed framework, with prior knowledge about the downstream task. This could help demonstrate any advantages of the proposed framework more clearly.

2. To strengthen the evidence presented in the paper, it would be beneficial to include experiments with real-world datasets. Providing results with tabular datasets could help establish stronger findings.

**Questions:**

1. It would be beneficial to observe the learning curve of three players during algorithm optimization of the regret value in order to better understand their learning behaviors.
2. The authors introduce the concept of "self-defined signal" as another perspective, but further clarification is needed to fully comprehend this idea.
3. Although recent language models do not require prior knowledge of future prediction tasks, they have demonstrated impressive generalization performance. It would be interesting to explore the potential of incorporating the proposed framework into the training procedure.

**Limitations:**

The paper presents a foundational approach that can be applied to various scenarios. The limitations and wider implications discussed in the paper are well-defined. Finding proper prior knowledge to use and further compute the representation should be studied and explored.

---

> ### Author Rebuttal · Authors · 2023-08-08
>
> # Response to the strengths
>
> To clarify Figure 1 for Example 5: It shows that mixed
> representation is more important whenever $\{s_{i}\sigma_{i}^{2}\}$
> are more spread ($\alpha$ is closer to $0$). Both regrets are decreasing
> as a function of $\alpha$, as can also be seen from the analytical
> expression. The value of $\ell_{*}$ is also shown, which is small
> for extreme values of $\alpha$ (large or small) since in this case
> the values $s_{i}\sigma_{i}^{2}$ are again more spread.
>
> # Response to the weaknesses
>
> We have conducted the requested experiment on synthetic
> data, though we believe that it exemplifies well the algorithm's merit, and
> can also be easily extended to real-world datasets. The dataset
> is a set of pictures, each has 4 shapes selected randomly from a dictionary
> of 6 shapes. In accordance, we consider a
> class with 6 different response functions, given by the 6 binary classification
> functions indicating whether a specific shape appears in the picture
> or not. The representation should be chosen so that these functions
> are predicted with minimal regret. Then, the phase 1 step in our proposed algorithm
> is simple, since the adversarial function can be found by a simple
> maximization over the 6 functions. The phase 2 step simply finds
> for each of the 6 classification functions the best representation-predictor
> $R,q$ using gradient descent, where $R$ is a linear representation,
> and $q$ is logistic regression. The algorithm then computes the loss of the obtained representation.
> For comparison, we also find the standard principal component analysis (PCA) representation and compare its obtained loss to that obtained by the algorithm.
> The result is shown in the additional PDF file of the review (Figure 3).
> It shows that the loss of PCA is much larger, not only for the worst-case
> function (blue), but also for the average-case function (orange).
> From a different point of view, in order to achieve the loss of the
> optimized representation with $r=1$ more
> than $r=15$ dimensions are required for PCA.
>
> This experiment can be easily extended to MNIST digits, and furthermore, an experiment with real-world
> data may replace synthetic shapes/digits with photos
> of faces from a large group of people. We will conduct such an experiment
> and report the results in the revised version. Nonetheless, since our example captures the core idea, we envision
> that the results will be similar to the ones obtained for the synthetic case.
>
> # Answers to the questions
>
> 1. An indirect way to observe the learning curve can is through the
> left panel of Figure 2 in the paper (Page 9). It can be observed there
> that the regret typically decreases as the number of representation
> matrices increases from $1$ to $4,7,10$. We did not include an explicit
> learning curve due to limited space, but we will add an explanation
> in the revised paper. In addition, we have plotted in the additional PDF file of the
> review (Figure 1) the learning curve in a linear-MSE
> setting, showing explicitly how the regret decreases as an additional
> matrix is added to the set of representations. We will add this figure
> to the appendix of the revised paper.
>
> 2. The term we use "self-defined signal" is borrowed from self-supervised
> learning (SSL). We refer the reviewer to the recent interesting survey
> [Shwartz-Ziv, LeCun, To Compress or Not to Compress - Self-Supervised
> Learning and Information Theory: A Review]. Similarly to our setting,
> in SSL, the learner only uses unlabeled features data, and is required
> to learn a representation. In the words of Shwartz-Ziv and LeCun "_...SSL
> employs self-defined signals to establish a proxy objective between
> the input and the signal. The model is initially trained using this
> proxy objective and subsequently fine-tuned on the target task._''
> For example, contrastive methods generate fictitious binary labels
> to the feature vectors, and the representation is optimized to minimize
> the distance between positive samples and maximizing the distance
> between negative samples. These binary labels are the aforementioned
> ``self-defined signal''. Similarly, our representation learning
> scheme is optimized from unlabeled features, and the optimization
> algorithm generates fictitious responses based on the adversarial
> function $f_{(t)}$ in the current iteration. The responses $f_{(t)}(X)$
> can be considered the self-defined signals that the learning algorithm
> generated to optimize its representation.
>
> 3. This is a very interesting future direction, though naturally large
> language models (LLMs) are beyond the scope of this paper (which is
> more theoretically oriented). Indeed, suppose that an LLM is first
> pre-trained in an unsupervised manner, and then fine-tuned for a specific
> prediction task. Suppose that the resulting prediction accuracy is
> low, and so the prediction task is, in some sense, "adversarial".
> What can be gained by a re-train of the LLM in order to better predict
> this "adversarial" prediction task? Can an additional "head"
> improve accuracy? How should they be trained? This is specifically
> of interest given the high cost associated with re-training.
>
> We thank the reviewer for the comments. We agree with the
> importance of experiments on real-world datasets, and aim to extensively
> experiment with our proposed algorithm in more practical scenarios in the future.
> However, our main motivation in this paper is theoretical
> and foundational. We hope that the additional numerical example that
> we provided already demonstrate the merit of our approach. We would kindly appreciate a re-assessment of
> the paper based on the above answers.

---

> > ### Comment · Reviewer_WSXs · 2023-08-15
> > **Thanks for detailed clarification!**
> >
> > Dear authors,
> >
> > Thanks for your clarification and further experimental results. Figure 1 in the attachment has addressed my concerns and offered great evidence, validating the effectiveness of proposed framework. Besides, thanks for writing details for self-defined signal, I believe authors' revision can enhance writing. At last, I agree with LLM's perspective with authors and acknowledge that this paper's scope aims to study the framework from theoretical perspective. I have raised my score by 1.

---

### Official Review · Reviewer_3ghR · 2023-07-06

**Soundness:** 3 good
**Presentation:** 3 good
**Contribution:** 2 fair
**Rating:** 6
**Confidence:** 3

**Summary:**

This work studies the problem of dimensionality reduction given prior information about downstream tasks via game theoretic optimization point of view. They define a three player game.  One player chooses the parameters of the representation function that reduces the dimensionality of the data. One player chooses the downstream task, constrained to respect a task distribution prior. The last player chooses the parameters of the model learned based on the dimensionality reduced inputs and the task. The authors theoretically analyze the case of linear dimensionality reduction and linear prediction functions under the MSE loss function. They then propose a greedy algorithm for the general case, which they evaluate empirically.

**Strengths:**

The framework and proposed solution are clearly explained. To the best of my knowledge, while adversarial learning between two or more agents is a well explored tool for improving generalization and robustness, the specific framework proposed by the authors is novel. In addition, even in the seemingly easy case of linear models and MSE loss, Theorem 3 is quite non trivial. The iterative greedy algorithm for the general case may also be of interest to practitioners.

**Weaknesses:**

 I would have liked to see more motivation for the specific framework proposed. More specifically:

 * It is not clear why the game is not a a two player game instead of a three player game. I do not understand why we need two separate players for controlling $R$ and $Q$. In fact the paper mostly ignores the $Q$ player by focusing on the regret of the $R$ player, implicitly assuming that $Q$ best responds by finding the optimal model. Is there a specific motivation for having the $Q$ player?
*  If we were to collapse the $R$ and $Q$ player, the setting is closer to adversarial learning between two players, the one that chooses the model and the one that (with some constraints) is allowed to change a given training dataset in an adversarial way. The main difference is that in this work the adversary controls a response function $f$ instead of manipulating a training set. It is not clear to me if/when this difference is significant, especially in practical settings where the input feature and response function distributions are discrete.
* It is not very clear to me if/how the randomized representation function $R$ in the mixed strategy setting can be used in practice. While it makes sense from a game theoretic point of view to extend to mixed strategies, I think connecting this back to the representation learning use case will help make the paper more compelling.

**Questions:**

I would suggest to the authors to clarify the points mentioned in the Weaknesses section.

**Limitations:**

Limitations have been adequately addressed.

---

> ### Author Rebuttal · Authors · 2023-08-06
>
> # Motivation
>
> While the concept of "representation learning" is rich, the main
> motivation of our paper is unsupervised dimensionality reduction (compression),
> in which the representation is restricted in its dimensionality. Our
> approach is mainly adequate whenever the compressed representation
> cannot capture all the information required for future predictions,
> because of the restricted dimensionality. In this case, the mixing
> (randomization) is utilized to cope with the future uncertainty.
> We further elaborate below.
>
> # Two-player vs. three-player game
>
> We have proposed a three player game in order to create a clear distinction
> between our work and previous works such as [Dubois et al, "Learning
> optimal representations with the decodable information bottleneck].
> This is however, just a semantic issue, since as mentioned in the
> paper, the first player (representation) and the third player (predictor)
> cooperate in order to minimize the regret, and they can actually be
> thought of as a single player. The fundamental difference is that
> between supervised ([Dubois et al]) vs. unsupervised learning
> (our paper). Mathematically, this difference affects the the order
> of actions allowed by the player, as follows:
>
> *  __Supervised learning__: The intention is to show that an efficient
> predictor from features $X$ to response $Y=f(X)$ can actually be
> decomposed into (i) Learning a representation $Z=R(X)$ of $X$ (2)
> Learning a predictor from $Z$ to $Y=Q(Z)=Q(R(X)$. Loosely speaking,
> various papers explain that the remarkable generalization of deep neural
> networks (DNNs) is obtained because they learn a very powerful representation,
> which allows their predictor $Q$ to be simple. In
> a DNN, the layers up to the last one are considered the representation,
> and the last one as the logistic regression predictor. To explore
> this idea [Dubois et al] considered the following two player game.
> Alice chooses a supervised prediction problem $X$ and $Y=f(X)$,
> and then Bob chooses a representation. In our notation the resulting
> problem can be written as $\max_{f}\min_{R}\min_{Q}$. In words, the
> representation $R$ is chosen after the supervised learning problem
> $f$ is chosen, and so the representation is tailored to the specific
> supervised learning at hand. This formulation thus fits supervised
> learning problems.
>
> * __Unsupervised learning (our paper)__: The player which takes the
> first action is the representation player, because it needs to choose
> the representation only on the basis of the unlabeled features (and
> prior knowledge about the class of future prediction tasks, but not
> the specific prediction task). Only afterwards, the function player
> chooses a response function $f$ that results a prediction problem
> of $Y=f(X)$ from $R(X)$. Then, given that the representation player
> now knows $(Z,Y)=(R(X),f(X))$ pairs, it can optimized its predictor
> $Q$. In our notation the resulting problem can be written as $\min_{R}\max_{f}\min_{Q}$,
> and it is important to note the difference in the order of the first
> and second maximum and minimum operations in this formulation (our
> unsupervised learning problem) compared to previous ones (supervised learning).
>
>
> The general ambition of these two problems is also different. In supervised
> learning, the goal is to establish that efficient predictors must
> have an internal efficient representation. We do not have such goal
> in our unsupervised learning problem, but rather dimensionality reduction.
> Our work can also be motivated by the timely pre-training approach:
> What is the optimal way to create a representation of the data in
> case there is no concrete prediction task (i.e., no labeled data)
> at this time, but this representation will be used in the future to
> design predictors ? We believe that such a formulation is of importance
> since pre-training is at the heart of recent advances in AI. The adversarial
> learning is somewhat different. Note that in our paper we focus on
> the representation and set aside generalization aspects. So our results
> are stated in terms of known or empirical feature distribution. In
> adversarial settings, there is a true unknown underlying distribution
> $(X,Y)$, and a corrupted data set $(\{X_{i},C(Y_{i})\}_{i=1}^{n})$
> where $C$ is some corruption operation. As said, in our case, there
> is no distinction between the dataset and the true distribution.
>
> # Mixed strategies
>
> In general, the expected pay-off of mixed strategies is relevant when
> the players are engaged in multiple, repeating games. When both players
> use their optimal mixed strategies, the expected payoff over multiple
> games cannot be improved by any unilateral action of the players.
> So, returning to our setting, using the optimal mixed representation
> strategy is relevant when there are multiple representation games.
> That is, the learner observers multiple datasets and need to compress
> each of them. A mixed strategy would mean to choosing the representation
> rule at random, and so different datasets would be represented by
> different rules. This fits practical situations in which datasets
> are routinely collected (say a camera/robot/space-probe that
> captures a set of photos each day, the collection of medical or genomic
> data from patients, etc.), and there is uncertainty regarding the future inference
> task. See another example in our response to reviewer xuhY. Intuitively, it make sense to randomize the representation,
> in order to avoid the unlucky situation in which the response of interest
> is always poorly predicted from the represented features. We have
> discussed this in line 114 in the paper.
>
> We thank you for your comments. We hope that our answers have clarified
> the points you have raised, and that this improves the overall quality of the
> paper.

---

> > ### Comment · Reviewer_3ghR · 2023-08-15
> > **Response very helpful**
> >
> > I would like to thank the authors for their very helpful response. I think I understand their motivation better now.
> >
> > I think the paper would be significantly improved if the authors highlight the aspects of the framework that make it unsupervised or more closely related to pretraining.
> >
> > When I think of pretraining for example, the distribution of tasks I observe during pretraining are not the same as the ones during training or inference. Same thing for unsupervised settings: I can devise tasks like clustering or training encoder-decoder networks for dimensionality reduction. But in the end my downstream task may be different e.g. classification.
> >
> > The paper should be more explicit about the relationship between the a priori known tasks and the tasks that need to be solved at inference time. Adding the discussion to related work on generalization bounds would also be useful given the paper targets unsupervised settings.
> >
> > I am satisfied with the response if the authors so I maintain my score and my acceptance recommendation.

---

### Official Review · Reviewer_xuhY · 2023-07-06

**Soundness:** 2 fair
**Presentation:** 3 good
**Contribution:** 2 fair
**Rating:** 4
**Confidence:** 3

**Summary:**

This paper studies the important problem of representation learning, where the goal is to learn a representation that works well for all problems in a known problem class. It presents theoretical results for the special case where both the representation function and prediction function class are linear, and also provide an extension of the algorithm to the general function class setting.

**Strengths:**

The problem of representation learning is obviously important, and the paper did a good job describing the problem setting, motivation and results.

**Weaknesses:**

1. The discussion regarding mixed representation in the paragraph on line 61 seems not practically relevant. In most cases, it makes sense to assume that there exists a representation z that captures all necessary information from x that predict any function f in F, which implies that there is a pure-strategy Nash Equilibrium.

2. An important piece of prior work is missing. [3] studies the exact same problem setting that is studied in this paper. Their theoretical result seems to subsume the theoretical result in this paper, as they allow arbitrary nonlinear representation and prediction functions. In addition, there is another line of work that assumes the tasks come in an i.i.d fashion, and uses uniform convergence analysis to bound the regret on a new task [1,2]. At the moment, it is unclear how the current paper compare with these prior works mentioned above. In particular, what are the differences between Theorem 2 in the current paper v.s. Theorem 2 and 3 in [3] when specialized to the linear setting.

[1] Jonathan Baxter. A model of inductive bias learning. Journal of artificial intelligence research, 12:
149–198, 2000.

[2] Andreas Maurer, Massimiliano Pontil, and Bernardino Romera-Paredes. The benefit of multitask
representation learning. The Journal of Machine Learning Research, 2016.

[3] Tripuraneni, Nilesh, Michael Jordan, and Chi Jin. "On the theory of transfer learning: The importance of task diversity." Advances in neural information processing systems， 2020.

**Questions:**

See above.

---

> ### Author Rebuttal · Authors · 2023-08-06
>
> # Answers to questions
> 1. The viability of mixed representations is for the _unsupervised_ learning problem we consider in the paper, whose goal is dimensionality reduction (our setting is indeed an unsupervised problem because only unlabeled features are provided to the learner, even though the representation is optimized for a future inference task). The case mentioned by the reviewer, in which a single representation is useful for many inference tasks occurs in other situations, e.g., when training neural networks with large datasets in supervised learning. This is reflected by the common practice to use an efficient neural network for one task and only train its final hidden layers in order to adapt it to a different task. So there a single representation is capable to capture all that is needed. However, this representation is typically unrestricted, as the network is either deep or wide. By contrast, our main motivation is dimensionality reduction (compression), in which the representation is restricted to a given dimension $r$. In this case, a mixed strategy is a practical option, as demonstrated by the following illustrative scenario: Suppose that datasets of photos are routinely collected by some device for future prediction tasks, but need to be substantially compressed. Suppose that there are two potential compression (representation) rules: One that smooths the photo and one that mainly keeps its edges. Using a pure representation strategy that is based only on smoothing will prohibit any future prediction task that hinges on sharp edges from obtaining low loss, and vice-versa. We propose to use a mixed representation, which randomizes between the two representation rules, and thus maximizes the worst-case prediction performance (in expectation, over multiple representation games). In general, mixed representation is a viable option whenever the compressed representation cannot capture all the information required for future predictions, and the mixing (randomization) is utilized to cope with future uncertainty. As evident, this situation is very different from cases in which the representation is not restricted, and labeled data is given. This explains the practicality, and perhaps even necessity,  of mixed strategies. We nonetheless remark that our results also address the optimal representation in pure strategies (Theorem 2), and that the algorithm we propose outputs a pure minimax representation if it is run on a single representation rule.
>
> 2. Thank you for referring us to these interesting papers. We will add them to the reference list of the paper, along with a discussion. We have carefully read [3] and a comparison of our paper with [3] is as follows. The common to both papers is that a representation is sought that is useful for a future prediction task. However, the our paper and [3] diverge from this point onward:
>
>    * __The learning setting is different (unsupervised vs. supervised)__: In our work the learner observes only unlabeled data and aims to optimize the representation, whereas in [3] the learner first observes $t$ labeled datasets of different tasks to optimize the representation, and then another labeled dataset for the new prediction task. In [3] the problem is realizable, that is, the labeling functions actually come from a common representation, but we do not make any such assumption.
>
>    * __The goal is different (approximation error vs. generalization error)__: We focus on explicitly finding the regret minimizing representation (either analytically or algorithmically), while [3] assumes two-step empirical risk minimization (ERM) and studies generalization bounds, stated in its Theorems 2 and 3. In fact, in our paper we side-step finite-sample issues and focus on the optimal representation for a given feature distribution, which could be either the true distribution or the empirical distribution of the data ([3] of course assumes that the joint distribution is unknown, as otherwise the problem is trivial). One simple consequence of this difference is that, in our case, task diversity (rich $\mathcal F$) leads to large regret, whereas in [3] task diversity leads to low generalization bound (as emphasized in the title of [3]). Therefore, loosely speaking, our paper studies a performance merit in the spirit of "approximation error", whereas [3] focuses on "estimation error". Therefore, our work actually complements the approach and results of [1,2,3], rather than subsumed by them. We also add that we have theoretical results for non-linear representations in a Hilbert space (Appendix F).
>
> We hope that these clarifications and the distinction from previous works raise the merit and contribution of our paper in the eyes of the reviewer, and would very much appreciate an updated assessment.

---

> > ### Comment · Reviewer_xuhY · 2023-08-10
> > **Misunderstanding regarding the disinfection between unsupervised v.s. supervised learning**
> >
> > It seems from the response that the authors have a misunderstanding regarding the disinfection between unsupervised v.s. supervised learning. The representation learning stage studied in this paper is definitely supervised learning, because you have access to the class of response function $\mathcal{F}$, meaning that you can generate **infinite** amount of labeled data on **any** task. This is an even stronger assumption than the setting studied [1,2,3], where only a finite set of labeled data is given for a subset of tasks sampled from $\mathcal{F}$.

---

> > > ### Author Response · Authors · 2023-08-17
> > > **A clarification of the misunderstanding**
> > >
> > > We accept your comment, and kindly ask to offer the following clarification.
> > >
> > > ## A clarification of the setting
> > > Our assumption that the response function belongs to a class ${\\cal F}$ covers an entire spectrum of possibilities between:
> > >
> > > 1. __Regular supervised learning__: ${\\cal F}=\\{f^{*}\\}$ is a singleton.
> > > 2. __No supervision__: When ${\\cal F}$ is the class of _all_ possible response functions, then this is essentially an unsupervised learning problem, since the learner does not have any valuable information on the response function. Indeed, when we considered the class $\\mathcal{F}\_S = \\{ \\|f\\|\_S\leq1 \\} $, we have shown that if $S$ is the identity matrix $I_{d}$ then the standard unsupervised PCA solution is recovered (Proposition 14, Appendix E).
> > >
> > > The case of a finite set of $t$ functions of [3] falls in between these two extremes.
> > >
> > > ## Why we (inaccurately) described our setting as unsupervised?
> > >
> > > 1. Our goal of dimensionality reduction is typically associated with unsupervised learning.
> > > 2. The recovery of the PCA solution for $S=I_d$ mentioned above.
> > > 3. As said by the reviewer, for a class of finite number of $t$ functions, a learner can easily generate labeled data, so this is supervised learning. However, for _continuous_ classes of functions, this may be computationally difficult. Indeed, the first part of the paper focuses on the class $\\mathcal{F}\_S = \\{ \\|f\\|\_S\leq1 \\} $, for which the effective number of functions scales exponentially with the dimension $d$. We thus referred to this setting as unsupervised, because unlike supervised learning, it is difficult for the learner to generate a response to all functions in the class (not even to a covering set of the class). Our paper addresses this computational challenge, either theoretically (for $\\mathcal{F}\_S$), or via an algorithm that optimizes the representation by efficiently generating response functions.
> > >
> > >
> > > Following your comment, we agree that just referring to our setting as unsupervised is inaccurate. in the revised paper, we will describe the setting more accurately, as a spectrum of possibilities between unsupervised and supervised learning, and place our results (“approximation”) in context to [1,2,3] (“generalization”). We believe that this will further improve both the presentation and contextualization of our results, and hope that this clarifies the misunderstanding.

---

### Official Review · Reviewer_rwLL · 2023-07-06

**Soundness:** 3 good
**Presentation:** 3 good
**Contribution:** 2 fair
**Rating:** 4
**Confidence:** 3

**Summary:**

This paper studies the representation learning from the multi-layer gaming perceptive. The authors propose an min-max optimization problem for learning a good representation with the prior knowledge of the latter prediction tasks. The authors provide some theoretical analyses and examples in the linear case. For the general tasks, some iterative algorithms are proposed and the numerical results are provided to demonstrate the performance.

**Strengths:**

- This paper is well written and easy to follow. The theoretical results are provided with several examples to demonstrate the intuition behind the theories.
- Numerical results demonstrate the effectiveness of the proposed result.

**Weaknesses:**

- The connection between the linear case and the general, three-player game case is not enough. In the linear case setting, it's easy to show a close form solution for this min-max problem. However, in the general setting, the authors does not fully demonstrate why the iterative based algorithm can convergence.
- I wonder the time complexity of the proposed general case algorithm since it need to call Algorithm 2 and 3 in the nested loops.

**Questions:**

Please address my concerns in the weakness part, if possible.

---

> ### Author Rebuttal · Authors · 2023-08-06
>
> # Response to the weaknesses
>
> (1) We believe that proving the closed-form solution for the min-max problem in the linear-MSE caseis a major contribution of the paper since the proof is neither trivial or simple: It spans 9 pages and requires a variety of techniques. We also mention that this solution is also generalized (Appendix F) to the a Hilbert space setting, which allows for _non-linear_ representations.
>
> Regarding the convergence analysis: Unfortunately, the SOTA knowledge in the analysis of such algorithms does not allow to make any meaningful claims on the convergence of our algorithm. In fact, iterative algorithms for
> min-max problems are not even well understood for basic settings,
> such as convex-concave min-max games. Their convergence is a rather delicate matter.
> One illustration to this claim is the fact that the only the _average_ iteration converges for concave-convex
> two-player games. This is a classic result from [Freund and Schapire, Adaptive game playing using multiplicative
> weights]. However, various authors have noted that the final iteration might actually diverge! Games which are non-convex/non-concave like ours are naturally much more complicated, and far less understood. We have discussed min-max games in detail in Appendix B, starting from line 686, and discussed this difficulty in the context of our algorithm in Appendix G.2, starting from line 1052. On a positive note, any future advances in min-max algorithms would naturally elucidate the convergence properties of our algorithm, so there is something to hope for.
>
> (2) Complexity of Algorithms 2 and 3: Algorithm 2 runs for a fixed
> number of iterations $T_{f}$, accepts $k$ representations, and thus makes
> $kT_{f}$ updates. Each update is comprised from a gradient step for the adversarial function (cost $C_{1})$, and optimization of
> the predictor (cost $C_{2})$. So the total computation complexity
> is $kT_{f}\cdot(C_{1}+C_{2})$. The most expensive part is the optimization
> of the predictor $C_{2}$, and this can be significantly reduced by
> running a few gradient steps for the predictor instead of a full optimization.
> If such $g$ gradient steps are taken then $C_{2}$ is replaced by $C_{1}g$,
> and the total computational cost is $kT_{f}(g+1)C_{1}$. Algorithm
> 3 is more complicated, but the computational complexity analysis is
> similar. It runs for $T_{R}$ iterations and the total cost is roughly
> on the order of $T_{R}k^{2}gC_{1}$ (taking $g$ gradient steps for
> the predictor optimization; $k$ is the number of representations,
> and is controlled by the learner; $C_{1}$ is determined by the computer,
> and $g$ should be large enough to assure quality results). We also
> refer the reviewer to the Appendices G.1 and G.2 for the two paragraphs
> entitled "Design choices and possible variants of the basic algorithm"
> which discuss various practical trade-offs between computational complexity
> and accuracy. That being said, as the numerical examples show, our
> algorithm can run on reasonably modest computational resources, and
> can be accelerated given larger computational resources.
>
> # Outlook
>
> The convergence properties of our algorithm is indeed an important
> question, and as we discussed above, it is directly linked to general
> challenges in the analysis of saddle-point algorithms (an active research
> area that is strongly motivated by the GAN minimax problem, among others).
> In this respect, we would like to highlight a possible opportunity that this connection brings.
> The problem of finding optimal representations from unlabeled
> features is very similar to the problem of optimal pre-training.
> As Reviewer "fmep" observed, this implies that the difficulty of
> min-max games is directly related to the difficulty of optimal pre-training.
> Thus any future advances in saddle-point algorithms may be utilized to improve pre-training.
> This is an important message for a wide audience since the pre-training
> approach is at the heart of recent major advances in AI.
>
> We thank the reviewer for the otherwise positive assessment, and hope
> that the new perspective we provide here may be reflected in the final
> score given to the paper.

---

### Official Review · Reviewer_fmep · 2023-07-07

**Soundness:** 3 good
**Presentation:** 2 fair
**Contribution:** 3 good
**Rating:** 6
**Confidence:** 3

**Summary:**

This paper presents a game theoretic framework for learning representations in a general setup where some knowledge/prior about the prediction task is available. The representation learning task is cast as a certain game --- which is worked out exactly in the fully linear setting under the squared loss (casting new light on PCA-type techniques), and for which a minimax solution-seeking iterative gradient-based method is provided for a more general case. Theory is accompanied by examples, both worked-out and experiments, that support the theory and offer new insights into well-known representation learning questions (such as the necessity of mixed rather than pure representations etc).

**Strengths:**

The main strength of this paper is that the game theoretic framework proposed by the authors appears to shed light onto many important questions about representations (including the necessity and type of required randomization in the representation, how to form universal representation-based initializations, how to use priors, etc.) in a very compact setup. Secondly, the theory in the linear case is worked out quite precisely and nicely, thus shedding new (to me at least) light on PCA-based representation technique. Nice worked-out examples are given throughout the paper to support the general theory. As well as, finally, not only is an interesting general representation-building algorithm given, but it is also, through a few experiments, confirmed to be implementable, reasonable, and overall sufficiently practical.

**Weaknesses:**

A weakness of the proposed game-theoretic framework (which could alternately be viewed as its strength, paradoxically) is that it doesn't seem to allow us to run away from the intrinsic hardness (and hardness of analysis) of finding good representations --- instead, this difficulty simply translates into the difficulty of procuring convergence analyses/rates for methods that solve for generally nonconvex/nonconcave saddle points that result out of this formulation. Now, why this could in the future prove to be a strength is that conceivably, it might allow for some future work on connecting hardness of finding good representations (in settings such as the one discussed in this paper), to the hardness of certain saddle point problems. Still, such future use is not fully clear at this point, which is why at this point in time I listed it here. (Of course, I would welcome arguments from the authors as to how this weakness could be turned into a strength.)

That said, given the hardness of the problem and the overall amount of interesting contributions in this paper, I will not consider the above to be of too much impact for the purposes of rating this paper.

**Questions:**

First off, the discussion about the information bottleneck approaches, which is in part done in the intro and in part --- in the appendix, left me still wondering about the mentioned existing two-player game theoretic formulation of this problem, and how it morally compares to the current proposed approach. Would the authors be able to provide me with some more detailed insight on this?

Additionally, and perhaps relatedly, I am wondering why the authors insist on branding their new setting as a three-player game rather than a two player game --- I am not sure if I am missing any crucial semantics behind this naming choice, and would appreciate any insights here. (In principle, the third player isn't even explicitly featuring in the general proposed Algorithm 1 for this setting --- currently it seems to just implicitly fit Q.)

I am also interested in hearing more about the connection between the stepwise fitting procedure, proposed by the authors, to boosting setups (which broadly speaking tend to also gradually add elements into the set to increase some notion of goodness). A sentence or two was mentioned to that extent in the appendix, but I would be interested in understanding any deeper existing underpinnings that led the authors to employ this particular setup here.

Finally, from what was described in the paper, I was still left to wonder what kind of useful initialization the authors imagine for the initial representation and initial function in Algorithm 1. A couple of concrete thought experiments would suffice and be much appreciated.

**Limitations:**

Yes.

---

> ### Author Rebuttal · Authors · 2023-08-06
>
> # Response to the weakness
>
> We highly appreciate this observation, and will add it to the conclusion
> of the revised paper (with a proper acknowledgment). To re-iterate
> this idea, our paper links between optimal unsupervised pre-training,
> to nonconvex/nonconcave min-max problems. This is an important message
> to researchers both in language models and optimization, since this relation implies any future improvement
> in iterative algorithms for min-max problems (an active research area,
> see Appendix B, line 677) will translate to improved pre-training/representation
> algorithms. This is a timely issue since such pre-training is
> at the core of recent breakthroughs in large language models.
>
> # Answers to the questions
>
> (1)+(2) The main distinction is between _supervised learning_ in [Dubois
> et al, Learning optimal representations with the decodable information
> bottleneck] and _unsupervised_ learning in our paper. In Dubois et
> al, the data is feature-response pairs $(X,Y)$ (in our notation $Y=f(X)$).
> The motivation in that paper comes from the general observation
> that practically good predictors (deep neural networks) in fact first learn an efficient
> representation $Z$ of the features, and then just use a simple predictor
> from $Z$ to $Y$. This is claimed to be quantified by the information
> bottleneck, where low mutual information $I(X;Z)$ indicates an efficient
> representation, and high mutual information $I(Z;Y)$ indicates that
> the representation allows for efficient prediction. Dubios et al formulated
> this idea as a two-player game, in which Alice chooses a prediction
> problem, i.e., a feature-response pair $(X,Y)$, and Bob chooses a
> representation $Z=R(X)$. In our notation, this means that the order
> of plays is $\max_{f}\min_{R}\min_{Q}$. The main idea of Dubios et
> al is to replace the standard mutual information with a different
> functional that has more desirable properties, but this is irrelevant
> to our paper. By contrast, we consider an unsupervised
> learning problem, so there are only features $X$, and no specific
> responses. We would like to optimize the representation $R$ when
> the response function $Y=f(X)$ is only known to be chosen adversarially
> from some class. This is formulated as a game of the order $\min_{R}\max_{f}\min_{Q}$,
> that is, first choosing a representation $R$, then a response function
> $f$, and then a predictor $Q$. To emphasize this order of plays,
> it was convenient for us to refer to these players as "representation
> player" ($R$), "response player" ($f$) and "predictor player"($Q$),
> where the $R$ and $Q$ players are different, yet cooperative, players,
> but one can indeed think of them as a single player. So, putting semantics
> aside (two/three player game), the technical distinction is between
> the order of plays, and the essential distinction is between a supervised
> vs. an unsupervised problem. Since this only lead to confusion, we
> will change this in the revised version of the paper.
>
> (3) The underlying idea here (representation), in GANs (generation),
> and in boosting (prediction) is twofold: (i) A powerful model can
> be obtained from a mixture of a few weak models (ii) Mixture models
> can be efficiently learned by a gradual addition of components to
> the mixture, letting the new component address the most challenging
> problem instance. In boosting, the final classifier is a mixture of
> simpler classifiers. Large weights are put on data points which are
> wrongly classified with the current mixture of classifiers, and the
> new component (classifier) is trained to cope with these samples.
> In GANs, the generated distribution is a mixture is of generative
> models. Large weights are put on examples which are easily discerned
> by the discriminator of the true and generated distributions, and
> the new component (generative distribution) optimizes the GAN objective
> on this weighted data. In our setting, the final representation is
> a mixture of representation rules. Weights are put on adversarial
> functions that cannot be accurately predicted with the current representation
> matrices. The new representation component aims to allow for accurate
> prediction of these functions. Overall, the common intuitive idea
> is very natural: The learning algorithm sees what is most lacking
> in the current mixture, and adds a new component that directly aims
> to minimize this shortage. The idea is nicely explained in
> of [Tolstikhin et al, AdaGAN:Boosting Generative Models].
>
> (4) The initial adversarial function can be simply initialized at
> random or using some prior knowledge (we mention this in Appendix
> G.1). This initialization is, however, of less importance compared
> to the initial representation, since it is typically
> of much larger dimension, and many gradient updates are required for
> its optimization. For the linear-MSE setting we propose a principled
> initialization method in Appendix H.1. Suppose that the
> required dimension is $r$. We iteratively construct the $r$ columns
> of the initial representation matrix $R^{(1)}$, as follows. First,
> we use the algorithm to optimize a representation for dimension $\tilde{r}=1$,
> initialized with an all-zero column. This optimization also outputs
> the adversarial function $f_{1}$. We set $f_{1}$ to be the first
> column of the sought initialization matrix $R^{(1)}$. Second, we
> optimize a representation for dimension $\tilde{r}=2$ initialized
> with $f_{1}$ as the first column and a zero second column. This optimization
> outputs the adversarial function $f_{2}$. We use $f_{2}$ as the
> second column of the sought initialization matrix $R^{(1)}$. We continue
> in this manner until there are $r$ columns, so that $R^{(1)}$ is
> initialized with the $r$ "most" adversarial functions, or the
> $r$ "principal adversarial functions". This also hints how initialization
> should be chosen in general.
>
> We thank the reviewer for the comments, and especially the one on
> pre-training. We hope that our clarifications can be reflected in an updated overall
> assessment of the paper.

---

> > ### Comment · Reviewer_fmep · 2023-08-21
> > **Acknowledgment**
> >
> > I thank the authors for their helpful responses to my questions and thoughts. As a result, I can now better locate many of the interrelations between this paper and some related concepts/works from the representation learning literature, the transfer learning literature, and the boosting/games literature (in particular, thank you for the boosting reference).
> >
> > I also much appreciated reading the authors' similarly helpful and detailed responses to other reviewers' questions and concerns. (i) Thank you for running the small synthetic experiment in response to another reviewer's request --- the plots, such as the gains compared to PCA, give a good sanity check for the proposed algorithmic framework. (ii) In terms of the semantics of the word "unsupervised", I agree with the reviewer who brought up this issue that this term is not accurate, but I can also see what has prompted the authors to use this term (esp. given that PCA is one of the recovered special cases).
> >
> > (iii) At the moment, I believe that the current manuscript offers a game-theoretic approach to generating representations that appears original and not developed in prior work --- its merits being exemplified by its potential ability to relate representations to appropriate saddle points, or, e.g. (to borrow from the other reviewers' points and the authors' responses below), being able to reason about the effects/necessity of randomization/mixing in representations. So, while I will further consult with the AC and other reviewers on their reservations, I currently believe that this paper successfully passes the novelty bar and would be useful to the community.
> >
> > I sincerely hope that the helpful nuggets of intuition from the authors' responses, as well as the reviewers' points, will make it into the updated version of the manuscript; I am sure they will help many a reader. My overall assessment of the paper being already positive, I would like to keep my current score, and will continue to support this paper.

---

> > > ### Author Response · Authors · 2023-08-21
> > > **Thank you**
> > >
> > > We thank you for the time and effort devoted to the evaluation of the paper, as well as for the feedback, which we will incorporate into revised version.

---

### Author Rebuttal · Authors · 2023-08-08

We responded to all the comments made by the 5 reviewers, as summarized below:

* __Reviewer fmep (score 6):__ The reviewer questioned the notion of a three-player game compared to a two-player game. We explain why this is just a semantic issue, and what actually matters is the order the players take actions. The order we specify is suitable for the unsupervised learning problem we study, whereas the order in previous papers is opposite and suitable for the supervised learning problems studied therein. We also credit the reviewer for an enlightening observation - that our paper links the difficulty of optimal pre-training to that of non-convex/non-concave games. We find this observation a valuable message since pre-training is a major ingredient in the recent success of large language models.

* __Reviewer rwLL (score 4):__ The reviewer is overall satisfied with the paper, but questions the convergence and complexity of the algorithm. As we explain, the analysis of iterative algorithms for finding saddle points is a rather delicate issue, even for the most basic games (i.e., convex-concave). Our algorithm aims to solve a non-convex/non-concave minimax game, and the SOTA in this problem domain is far from providing accurate convergence analysis for such games. As common, we thus have resorted to experimentation. In addition, we clarify the required computational complexity of the algorithm. We specifically detail the complexity of its inner iterations, since these only appear in the appendices. Given the difficulty of the problem, we find this complexity both reasonable and inevitable. We also highlight the main computational bottleneck of the algorithm (the full optimization of the predictor) and offer a remedy.

* __Reviewer xuhY (score 4):__ The reviewer questions the practicality of the mixed approach, and also points out missing related works. Regarding the mixed approach, we explain why the mixed minimax approach is viable when the goal is dimensionality reduction, and mention that we also have results for the optimal pure representation (Theorem 2). Regarding previous works, these papers follow the common aspect that "a representation should be thought for for future inference tasks". However, they are very different in the actual setting and in their results. Our paper considers an __unsupervised__ setting, and provides the optimal representation, either analytically or algorithmically. By contrast, the papers mentioned by the reviewer consider a realizable __supervised__ learning setting, and assume a two-step generic empirical risk minimization (ERM) algorithm, which can be impractical. Second, these papers derive generalization bounds (or "estimation error bounds"), whereas our bounds can thought of as an "approximation error" bounds. Thus, the results of our paper and those in previous works actually complement one another. We will add a citation to these papers in the revised paper, along with a discussion.

* __Reviewer 3ghR (score 6):__ The reviewer also mention the two-player vs. three player issue, and the viability of mixed representations in practice. We have addressed in detail both the comments, in a similar manner to our responses to previous comments on these aspects.

* __Reviewer WSXs (score 5):__ The reviewer asked to explicitly show the learning curve of the algorithm. Figure 1 in the additional PDF file explicitly shows how the regret decreases as the number of representation matrices increases. The reviewer also asked for an experiment that shows more explicitly the improvement of our algorithm compared to standard approaches, such as PCA. As detailed in our response, we have conducted an experiment, in which the feature vectors are pictures of simple shapes (shown in Figure 2 in the additional PDF file). The loss of the optimized representation is compared to the loss of standard PCA, and shows an impressive improvement (Figure 3 in the additional PDF file). This is still a synthetic-data experiment, but we will transform it to a real-world data experiment by replacing the shapes with photos of faces. Since the core principle is the same, we predict that the results will be of similar nature. That being said, our approach in this paper is theoretical and foundational, and we believe that the experiments we already conducted convincingly exemplify its merit.

We thank you for your effort in evaluating our paper.

---

### Decision · Program_Chairs · 2023-09-21

**Decision:**

Reject

**Comment:**

Three reviewers are in favor of acceptance while two reviewers are (weakly) not in favor. However, from my perspective, I believe a core criticism of one reviewer (of the linear case being easy) is without evidence, given the significant technical developments of the authors for the linear case. I hence emphasize this review less in the overall evaluation of this work.

On the positive side, the novel game framed by the authors is quite interesting. There are many insights, some highlighted in the discussion between Reviewer fmep and the authors. The connection to PCA is nice. The authors even managed to have some experiments, which helped convince one reviewer.

On the downside, the paper could do with a semi-major revision. The authors should better emphasize how their work is related to unsupervised learning as opposed to supervised learning. In particular, please see the comments given in the last post by Reviewer 3ghR. Also, I do believe the authors should look more closely at the work [3] and some discussion of this work in their paper feels warranted. One reviewer maintains a concern that the assumption of realizability made in [3] is not that critical for the comparison to the authors’ work, as the authors are in the infinite data regime and realizability vs non-realizability is mainly about estimation error rates of convergence (which are not relevant here). I suspect that in revising their work, the authors may even be able to better distinguish their work from supervised learning works by contrasting with [3].

Overall, this is a paper which, based on the ideas, I would like to accept. However, I am convinced that the authors can revise this work in several ways to make it stronger (including the addition of figures they produced during the discussion period). Pitching a new game like this one in a way that people "get" is important. So, as this time, I do not recommend this paper for acceptance, but I wish the authors luck revising and submitting at the next top venue.